# MMPD: Diverse Time Series Forecasting via Multi-Mode Patch Diffusion Loss

**Yunhao Zhang[1], Wenyao Hu[1], Jiale Zheng[2], Lujia Pan[2] & Junchi Yan[1]\***
[1]Shanghai Jiao Tong University, [2]Huawei Noah's Ark Lab
{zhangyunhao,hwy123456,yanjunchi}@sjtu.edu.cn
{zhengjiale2,panlujia}@huawei.com

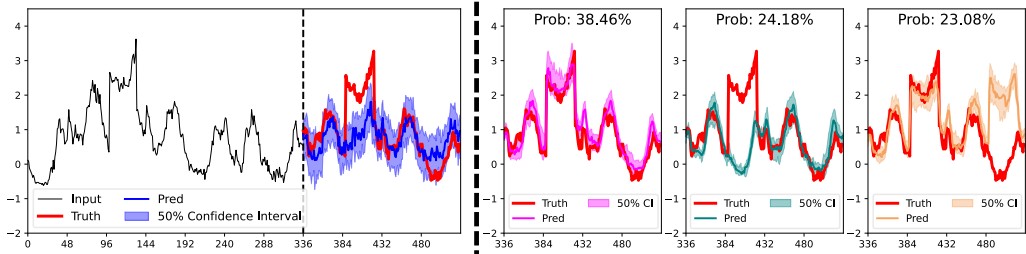

Figure 1: **MSE loss (Left)** *vs.* **our MMPD loss (Right)** using the same decoder-only Transformer backbone on dataset ETTm1, input 336-predict 192 task. MSE results in a single, ambiguous prediction with a symmetric, constant confidence interval, failing to capture sudden changes in the future. In contrast, our MMPD generates multiple sharp predictions with associated probabilities (only Top-3 predictions are shown), and the confidence intervals are asymmetric and vary over time. More visualizations are shown in Fig. 11 of Appendix J.

## Abstract

Despite the flourishing in time series (TS) forecasting backbones, the training mostly relies on regression losses like Mean Square Error (MSE). However, MSE assumes a one-mode Gaussian distribution, which struggles to capture complex patterns, especially for real-world scenarios where multiple diverse outcomes are possible. We propose the Multi-Mode Patch Diffusion (MMPD) loss, which can be applied to any patch-based backbone that outputs latent tokens for the future. Models trained with MMPD loss generate diverse predictions (modes) with the corresponding probabilities. Technically, MMPD loss models the future distribution with a diffusion model conditioned on latent tokens from the backbone. A lightweight Patch Consistent MLP is introduced as the denoising network to ensure consistency across denoised patches. Multi-mode predictions are generated by a multi-mode inference algorithm that fits an evolving variational Gaussian Mixture Model (GMM) during diffusion. Experiments on eight datasets show its superiority in diverse forecasting. Its deterministic and probabilistic capabilities also match the strong competitor losses, MSE and Student-T, respectively. The source code is publicly available at: `https://github.com/Thinklab-SJTU/MMPD`.

## 1 Introduction

Time series (TS) forecasting have made fast progress. Plenty of backbones have been proposed, incorporating various techniques like sparse attention (Li et al., 2019; Zhou et al., 2021), trend-season decomposition (Wu et al., 2021; Zeng et al., 2023), frequency enhancement (Zhou et al., 2022), patchify (Nie et al., 2023; Zhang & Yan, 2023) and cross-channel dependency (Liu et al., 2023).

Despite the rich works on backbone design, most works rely on regression losses like Mean Square Error (MSE) for training. However, using MSE essentially assumes that the future follows a Gaussian distribution with fixed variance (details in Sec. 3.1). Such a parametric distribution has several limitations, including its symmetric formulation and independent, constant uncertainty.

---

*Correspondence to: Junchi Yan. This work was partly supported by Scientific Research Innovation Capability Support Project for Young Faculty (U40) of the Ministry of Education of China, SRICSPYF-ZY2025019.

Most importantly, the single-mode Gaussian cannot support diverse forecasting, where the same past may lead to multiple possible futures. Diverse forecasting is necessary in the real world. On the data side, multi-mode pattern is a fundamental property: identical inputs can diverge into different futures due to unobserved background contexts (Bergmeir, 2024). On the application side, it is a natural requirement in downstream tasks: in domains like trading, an averaged forecast offers little actionable insight, whereas multi-mode predictions enable risk-aware decision (Tsay, 2005).

Therefore, even with carefully designed backbones, the model's capacity remains limited if the training loss cannot capture complex distributions. Several works have attempted to improve loss functions: Le Guen & Thome (2019; 2020) explore the Dynamic Time Warping (DTW). Due to high complexity, DTW-based losses are hard to scale to long-term forecasting. Salinas et al. (2020); Rasul et al. (2023) model the future with negative binomial and Student-T distributions. Woo et al. (2024) uses a mixture of parametric distributions. Although better than MSE, their formulations are still manually predefined, limiting the ability to model complex distributions.

To fill the gap, we propose Multi-Mode Patch Diffusion (MMPD) loss to model complex future distributions. MMPD is generally applicable to any patch-based backbone that divides input series into patches and outputs latent tokens for the future, now one of the most important categories of backbones in both supervised and foundation models. Given one input, models trained with MMPD loss can predict multiple diverse futures (modes), each with an associated probability, as illustrated in the right of Fig. 1. Meanwhile, MMPD loss also integrates with traditional deterministic forecasting, similar to MSE.

Technically, inspired by recent efforts of diffusion on visual tokens (Li et al., 2024a), MMPD constructs a diffusion process for future series conditioned on tokens from upstream forecasting backbones. In Sec. 3.2, we propose a lightweight Patch Consistent MLP as the denoising network in MMPD loss. When denoising a patch, it not only takes the corresponding token as the condition but also considers adjacent noisy patches, ensuring consistency across patches. The integration with deterministic forecasting is achieved by optimizing the diffusion objective at special anchor inputs. As diffusion samples exhibit multi-mode patterns, a multi-mode inference algorithm is devised in Sec. 3.3. It fits an evolving variational Gaussian Mixture Model (GMM) at each diffusion step alongside the reverse process. Priors from the forward process are injected via variational inference to guide the update of GMM. At the end of the reverse diffusion, the GMM outputs multi-mode predictions with corresponding probabilities. **The highlights are:**

**1)** Beyond the dominant MSE loss that assumes a simple Gaussian distribution, we propose the MMPD loss, leveraging the diffusion process to capture complex distributions. MMPD loss is backbone-agnostic and readily applicable to any patch-based backbone.

**2)** Observing multi-mode patterns in predictions, we devise a multi-mode inference algorithm that outputs diverse predictions with associated probabilities. Unlike pre-defined mixture distributions (Woo et al., 2024), the number and structure of modes are adaptively inferred, offering greater flexibility.

**3)** Experiments on eight datasets show the superiority of MMPD loss in diverse forecasting. Its deterministic and probabilistic capabilities also match the best-performing competitor losses, MSE and Student-T, respectively. Its generality is validated on four different backbones.

## 2 PRELIMINARIES ABOUT DIFFUSION MODELS

We leave the related works to Appendix A and briefly overview the preliminaries about diffusion models. Given training samples and corresponding conditions (e.g., images and captions): $\mathbf{y}^0, \mathbf{c} \sim q(\mathbf{y}^0, \mathbf{c})$, Diffusion models define a forward Markov process that gradually adds noise to the sample:

$$q(\mathbf{y}^{1:K}|\mathbf{y}^0, \mathbf{c}) = \prod_{k=1}^{K} q(\mathbf{y}^k|\mathbf{y}^{k-1}, \mathbf{c}) \quad q(\mathbf{y}^k|\mathbf{y}^{k-1}, \mathbf{c}) = \mathcal{N}(\mathbf{y}^k; \sqrt{1-\beta_k}\mathbf{y}^{k-1}, \beta_k\mathbf{I}) \quad (1)$$

where $\{\beta_k \in (0,1)\}_{k=1}^{K}$ is the variance schedule to control the added noise. With the forward process, a reverse Markov process for denoising is modeled by a neural network:

$$p_\phi(\mathbf{y}^{0:K-1}|\mathbf{y}^K, \mathbf{c}) = \prod_{k=1}^{K} p_\phi(\mathbf{y}^{k-1}|\mathbf{y}^k, \mathbf{c}) \quad p_\phi(\mathbf{y}^{k-1}|\mathbf{y}^k, \mathbf{c}) = \mathcal{N}\left(\mathbf{y}^{k-1}; \mu_\phi(\mathbf{y}^k, \mathbf{c}, k), \sigma_k^2\mathbf{I}\right) \quad (2)$$

where $\sigma_k$ is a step-dependent constant and $\mu_\phi$ represents the neural network that parameterizes the reverse process. The parameters of $\mu_\phi$ are learned by minimizing the negative log-likelihood $\mathbb{E}_{q(\mathbf{x}^0,\mathbf{c})}[-\log p_\phi(\mathbf{x}^0|\mathbf{c})]$. Through parameterization and simplification, the final objective is:

$$\mathcal{L} = \mathbb{E}_{\mathbf{y}^0,\mathbf{c},k,\epsilon}\left[\|\epsilon - \epsilon_\phi(\mathbf{y}^k,\mathbf{c},k)\|_2^2\right] \quad \mathbf{y}^k = \sqrt{\bar{\alpha}_k}\mathbf{y}^0 + \sqrt{1-\bar{\alpha}_k}\epsilon \quad \epsilon \sim \mathcal{N}(\epsilon;\mathbf{0},\mathbf{I}) \quad (3)$$

where $\alpha_k = 1 - \beta_k$ and $\bar{\alpha}_k = \prod_{s=1}^k \alpha_s$. $\epsilon_\phi$ is a network to parameterize $\mu_\phi$ in Eq. 2. It takes noisy sample $\mathbf{y}^k$, condition $\mathbf{c}$ and diffusion step $k$ as input and outputs the estimated noise in $\mathbf{y}^k$. Once well-trained, new samples can be generated from the reverse process, i.e., Eq. 2.

## 3 METHODOLOGY

### 3.1 BROADENING THE DEFINITION OF LOSS FROM A PROBABILISTIC VIEW

Given the past series $\mathbf{x} \in \mathbb{R}^T$, TS forecasting aims to predict values of the same series at the desired future horizon[1] $\mathbf{y} \in \mathbb{R}^\tau$. From a probabilistic view, the task is to model the conditional distribution of the future given its past: $p(\mathbf{y}|\mathbf{x})$. Assuming an independent Gaussian with predicted mean and fixed variance: $p_\theta(\mathbf{y}|\mathbf{x}) = \mathcal{N}(\mathbf{y}; f_\theta(\mathbf{x}), \sigma^2\mathbf{I})$, the objective of maximum likelihood estimation yields:

$$\max_\theta \mathbb{E}_{q(\mathbf{x},\mathbf{y})}\left[\log p_\theta(\mathbf{y}|\mathbf{x})\right] = \max_\theta \mathbb{E}_{q(\mathbf{x},\mathbf{y})}\left[-\frac{1}{2\sigma^2}\|\mathbf{y} - f_\theta(\mathbf{x})\|_2^2 + \text{Const}\right]$$
$$= \min_\theta \mathbb{E}_{q(\mathbf{x},\mathbf{y})}\left[\frac{\tau}{2\sigma^2}\text{MSE}(f_\theta(\mathbf{x}),\mathbf{y})\right] \quad (4)$$

where $f_\theta(\cdot) : \mathbb{R}^T \to \mathbb{R}^\tau$ is a neural network parameterized by $\theta$ to predict the mean. $\sigma \in \mathbb{R}$ is the constant standard deviation. $q(\mathbf{x},\mathbf{y})$ is the training dataset distribution.

Using gradient-based optimizers like Adam for training, the constant coefficient $\frac{\tau}{2\sigma^2}$ will be absorbed by the step size. Therefore, the following relationship holds (Bishop & Nasrabadi, 2006):

> **Using MSE loss implicitly assumes the future follows an independent Gaussian distribution, with predicted mean and fixed constant variance.**

This assumption is restrictive with limitations: 1) a single-mode Gaussian is unsuitable when multiple distinct futures are possible; 2) predicted steps are assumed independent, yet real-world steps are often correlated; 3) variance is fixed, whereas uncertainty typically evolves over time; 4) the Gaussian is symmetric, but real-world predictions may be asymmetric, e.g., rainfall is nonnegative.

More generally, regardless of how refined the network is, its expressiveness will be limited because MSE assumes a simple, parametric form for the future distribution. The same limitations apply to MAE loss, which assumes a Laplace distribution with predicted location and fixed, independent scale.

To move beyond restricted assumptions and model more complex distributions, we decouple the forecasting network into a ***backbone*** and a ***projector***:

$$f_\theta(\mathbf{x}) = g_\phi(\mathbf{H}), \mathbf{H} = h_\psi(\mathbf{x}), \theta = \{\phi,\psi\} \quad (5)$$

**1) The backbone** $h_\psi(\cdot)$ extracts latent representations and contains the majority of the parameters. Plenty of backbones with various techniques have been proposed in recent years (Wen et al., 2023).

**2) The projector** $g_\phi(\cdot)$ maps the representations to the output space, typically via a lightweight MLP with few parameters. Its design highly depends on the chosen distribution—for instance, using a Gaussian with predicted variance requires the projector to output both mean and variance.

Since the backbone is the core and dominates the parameter count, from the perspective of backbone optimization, the projector can be viewed as part of the loss, forming a composite, trainable loss:

$$\min_{\phi,\psi} \text{Loss}^\phi(\mathbf{H},\mathbf{y}), \mathbf{H} = h_\psi(\mathbf{x}) \quad (6)$$

In this broader definition, the projector $g_\phi$ acts as an auxiliary module that guides backbone optimization. This is conceptually related to the adversarial loss (Goodfellow et al., 2014), where a learnable

---

[1]We focus on univariate forecasting; for multivariate data, the loss is computed per channel and averaged.

discriminator is introduced to guide generator training. Adopting this view, we can design flexible losses that capture richer distributions beyond Gaussian forms. Moreover, traditional losses naturally fall into this framework: MSE can be expressed as: $\text{MSE}^{\phi}(\mathbf{H}, \mathbf{y}) = \frac{1}{\tau} \|\mathbf{y} - g_{\phi}(\mathbf{H})\|_2^2$

## 3.2 DIFFUSION TRAINING WITH PATCH CONSISTENT MLP

Unlocking the capacity of backbones requires losses corresponding to more flexible distribution families. Leveraging the strong ability to capture complex distributions, we propose a diffusion-based loss, $\text{MMPD}^{\phi}(\mathbf{H}, \mathbf{y})$. Unlike standalone TS diffusion models relying on specialized architectures (Tashiro et al., 2021), MMPD serves as a plug-and-play loss applicable across various backbones.

In this work, we focus on patch-based backbones, which divide past series into patches as input and output latent tokens for the future[2]. Specifically, in these backbones, past series $\mathbf{x}$ is divided into patches of length $P$ and then embedded into $T/P$ latent tokens. Backbones capture dependency among tokens and output latent tokens $\mathbf{H} = \{\mathbf{h}_j\}_{j=1}^{l}, l = \tau/P$, each corresponding to a future patch. As discussed in Appendix A, many recent supervised and pre-training models fall into this category.

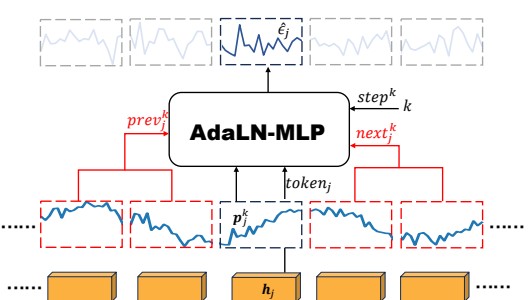

Figure 2: **Patch Consistent MLP** for diffusion. To predict the noise $\epsilon_j$ in patch $\mathbf{p}_j^k$, besides the corresponding token $\mathbf{h}_j$ and diffusion step $k$, adjacent noisy patches centered around $j$ ($\mathbf{p}_{j-r}^k, \ldots, \mathbf{p}_{j-1}^k$ and $\mathbf{p}_{j+1}^k, \ldots, \mathbf{p}_{j+r}^k$ colored in red) are also input to AdaLN-MLP as conditions. This ensures consistency across denoised patches.

The core idea of our MMPD loss is to enable flexible distribution modeling through conditional diffusion, where future latent tokens serve as the condition. To achieve this, a denoising network $\epsilon_{\phi}(\mathbf{y}^k, \{\mathbf{h}_j\}_{j=1}^{l}, k)$ is required. As an auxiliary module to guide backbone optimization, this denoiser should be lightweight. A straightforward strategy is to split $\mathbf{y}^k$ into patches $\{\mathbf{p}_1^k, \ldots, \mathbf{p}_l^k\}$ and use an MLP to denoise each patch $\mathbf{p}_j^k$ conditioned on token $\mathbf{h}_j$, as done in Li et al. (2024a) for visual tokens. However, such independent MLP models the marginal distribution of each patch $p(\mathbf{p}_j|\mathbf{x}), 1 \leq j \leq l$ rather than the joint distribution of all future patches $p(\mathbf{p}_1, \ldots, \mathbf{p}_l|\mathbf{x})$. This can lead to inconsistent samples during inference, resulting in discontinuous jumps between patches shown in Fig. 3(a).

To maintain consistency among patches while keeping denoiser lightweight, we extend Adaptive LayerNorm MLP (AdaLN-MLP) (Peebles & Xie, 2023) to construct Patch Consistent MLP:

$$\epsilon_{\phi}(\mathbf{y}^k, \{\mathbf{h}_j\}_{j=1}^{l}, k) = [\hat{\epsilon}_1 \bullet \ldots \bullet \hat{\epsilon}_l] \quad \hat{\epsilon}_j = \text{AdaLN-MLP}(\mathbf{p}_j^k, \mathbf{c}_j^k)$$
$$\mathbf{c}_j^k = \text{token}_j + \text{step}^k + \text{prev}_j^k + \text{next}_j^k$$
$$\text{token}_j = \mathbf{W}^{(token)}\mathbf{h}_j \quad \text{step}^k = \text{Emb}^{(step)}(k) \tag{7}$$
$$\text{prev}_j^k = \mathbf{W}^{(prev)}\left[\mathbf{p}_{j-r}^k \bullet \ldots \bullet \mathbf{p}_{j-1}^k\right] \quad \text{next}_j^k = \mathbf{W}^{(next)}\left[\mathbf{p}_{j+1}^k \bullet \ldots \bullet \mathbf{p}_{j+r}^k\right]$$

As illustrated in Fig. 2, the predicted noise $\hat{\epsilon} \in \mathbb{R}^{\tau}$ is the concatenation of predicted noise in each patch, i.e., $\hat{\epsilon}_j \in \mathbb{R}^P$. AdaLN-MLP is the denoising MLP from DiT block (Peebles & Xie, 2023), with details in Appendix B. To predict noise in patch $\mathbf{p}_j^k$, the conditioning vector integrates four components: latent tokens $\text{token}_j$, diffusion timestamp $\text{step}^k$, previous and next noisy patches $\text{prev}_j^k$ and $\text{next}_j^k$. Here, $\mathbf{W}^{(token)} \in \mathbb{R}^{d_{model} \times d_{model}}$; $\mathbf{W}^{(prev)}, \mathbf{W}^{(prev)} \in \mathbb{R}^{d_{model} \times rP}$ are learnable matrices. $\text{Emb}^{(step)}(\cdot)$ is the positional encoding. $r$ is a constant hyper-parameter that controls the adjacent range around patch $j$ that the MLP can access. Padding is used when $j \leq r$ or $j > l - r$.

The key distinction between Patch Consistent MLP and independent MLPs lies in its use of adjacent patches as conditions, i.e., the last line of Eq. 7. Fig. 3(b) shows that this ensures consistency across patches. Moreover, the additional parameters introduced (i.e., $\mathbf{W}^{(prev)}, \mathbf{W}^{(next)}$) are minimal.

**Integration with deterministic forecasting.** MMPD loss provides a flexible distribution. But in many applications, deterministic forecasting is still needed, which is the role typically served by MSE.

---

[2]MMPD can also be adapted to non-patch-based backbones with minor modification, as shown in Appendix H

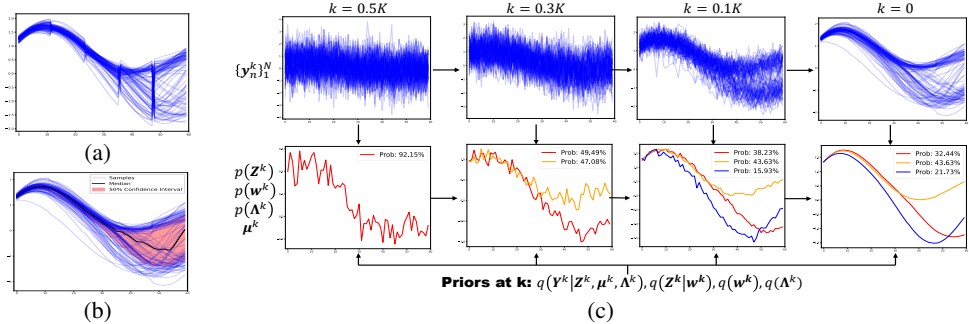

Figure 3: (a) Samples predicted by an independent denoising MLP on dataset Dynamic, showing the inconsistency between patches. (b) Samples predicted by our Patch Consistent MLP, displaying clear multi-mode patterns that are challenging to represent using simple statistics (black line: median, red area: 50% confidence interval). (c) The evolution of our multi-mode inference algorithm: at each step $k$, posteriors and estimations $p(\mathbf{Z}^k), p(\mathbf{w}^k), p(\mathbf{\Lambda}^k), \boldsymbol{\mu}^k$ are updated via variational EM steps, based on newly generated samples $\{\mathbf{y}_n^k\}_{n=1}^N$. The updates are also guided by priors at each step $k$.

A naive solution is to generate multiple samples and take the mean or median. However, diffusion iterations are costly. For efficiency, we integrate deterministic forecasting within the diffusion framework. Considering the diffusion objective (Eq. 3), if the noise cancels the sample at step $k^*$ such that $\mathbf{y}^{k^*} = \mathbf{0}$, the target reduces to a scaled negative ground truth $\epsilon = -\frac{\sqrt{\bar{\alpha}_{k^*}}}{\sqrt{1-\bar{\alpha}_{k^*}}}\mathbf{y}^0$. Thus, we treat $(\mathbf{0}, \{\mathbf{h}_j\}_{j=1}^l, k^*)$ as an anchor input for deterministic forecasting and define the joint objective:

$$\mathcal{L} = \lambda \left\| \epsilon - \epsilon_\phi(\mathbf{y}^k, \{\mathbf{h}_j\}_{j=1}^l, k) \right\|_2^2 + (1-\lambda) \left\| \frac{\sqrt{\bar{\alpha}_{k^*}}}{\sqrt{1-\bar{\alpha}_{k^*}}}\mathbf{y}^0 + \epsilon_\phi(\mathbf{0}, \{\mathbf{h}_j\}_{j=1}^l, k^*) \right\|_2^2 \tag{8}$$

where $\lambda = 0.99$ by default balances the probabilistic and deterministic objectives. $k^*$ is set to make $\bar{\alpha}_{k^*}$ close to 0.5 such that $\frac{\sqrt{\bar{\alpha}_{k^*}}}{\sqrt{1-\bar{\alpha}_{k^*}}} \approx 1$. Fig. 9 of Appendix F also shows that prediction accuracy is robust w.r.t $k^*$ across a broad range. After training, the deterministic prediction is directly obtained as $-\frac{\sqrt{1-\bar{\alpha}_{k^*}}}{\sqrt{\bar{\alpha}_{k^*}}}\epsilon_\phi(\mathbf{0}, \{\mathbf{h}_j\}_{j=1}^l, k^*)$, bypassing costly diffusion iterations. This integration introduces no new architectures, as it reuses the denoiser $\epsilon_\phi$. Importantly, the deterministic forecasting term does not conflict with the diffusion term - it is merely a special case of the diffusion objective at the anchor.

### 3.3 MULTI-MODE INFERENCE THROUGH EVOLVING VARIATIONAL GMM

Once trained, we get a flexible distribution $p_\theta(\mathbf{y}|\mathbf{x})$. However, unlike MSE loss that corresponds to a closed-form Gaussian distribution, $p_\theta(\mathbf{y}|\mathbf{x})$ modeled by a diffusion process is an implicit distribution that lacks an explicit analytic form. To summarize this distribution and extract interpretable information for downstream tasks, prior works typically draw samples from it and then compute statistics such as the median and confidence interval (Rasul et al., 2021a; Shen & Kwok, 2023) . However, as shown in Fig. 3(b), the samples exhibit multi-mode patterns, indicating that the same past can lead to multiple possible futures. Such diversity cannot be adequately captured by simple summary statistics.

To address this, we propose a multi-mode inference algorithm that explicitly summarizes distinct outcomes. Suppose the true distribution takes the following multi-mode form:

$$q(\mathbf{y}^0|\mathbf{x}) = \sum_{m=1}^M w_m \delta(\mathbf{y}^0 - \mathbf{y}_m^*), \sum_{m=1}^M w_m = 1 \tag{9}$$

where $\{\mathbf{y}_m^*\}_{m=1}^M$ denote the $M$ possible predictions and the probability to predict $\mathbf{y}_m^*$ is $w_m$. To estimate $\{w_m, \mathbf{y}_m^*\}_{m=1}^M$, we combine this multi-mode prior with the forward diffusion process (i.e., Eq. 1), yielding the forward distribution at step $k$:

$$q(\mathbf{y}^k|\mathbf{x}) = \sum_{m=1}^M w_m \mathcal{N}(\mathbf{y}^k; \sqrt{\bar{\alpha}_k}\mathbf{y}_m^*, (1-\bar{\alpha}_k)\mathbf{I}) \tag{10}$$

This is a Gaussian mixture distribution governed by two priors:

---

**Algorithm 1** Multi-Mode Inference Algorithm

---

**Input:** Future tokens $\{\mathbf{h}_j\}_{j=1}^l$, number of samples $N$ to draw via diffusion, maximum number of modes $M$, prior hyperparameters $\rho, u$ in Eq. 11.
**Initialize:** Estimation $\{\boldsymbol{\mu}_m^K\}_{m=1}^M$ and parameters in posteriors $\{\widetilde{u}_m^K, \widetilde{v}_m^K, \widetilde{\pi}_m^K\}_{m=1}^M$
**Ouput:** Statistics and probability of each mode
*# Generate initial samples at step $K$: $\{\mathbf{y}_n^K\}_{n=1}^N \sim \mathcal{N}(\mathbf{y}^K; \mathbf{0}, \mathbf{I})$*
**for** $k = K-1, \ldots, 0$ **do**
  *# Generate samples at step $k$: $\{\mathbf{y}_n^k\}_{n=1}^N \sim \text{Diffusion}\left(\{\mathbf{y}_n^{k+1}\}_{n=1}^N, \{\mathbf{h}_j\}_{j=1}^l, k+1\right)$*
  *# E-step: update posterior $p(\mathbf{Z}^k)$*

$$p(\mathbf{Z}^k) = \prod_{n=1}^N \prod_{m=1}^M (\widetilde{\gamma}_{nm}^k)^{z_{nm}^k}; \quad \widetilde{\gamma}_{nm}^k = \frac{\gamma_{nm}^k}{\sum_{s=1}^M \gamma_{ns}^k}$$

$$\ln \gamma_{nm}^k = -\frac{1}{2}\left[\frac{\widetilde{u}_m^{k+1}}{\widetilde{v}_m^{k+1}}\|\mathbf{y}_n^k - \boldsymbol{\mu}_m^{k+1}\|_2^2 + \tau \ln(2\pi)\right] + \frac{\tau}{2}\left[\psi(\widetilde{u}_m^{k+1}) - \ln(\widetilde{v}_m^{k+1})\right] + \psi(\widetilde{\pi}_m^{k+1}) - \psi(\sum_{s=1}^M \widetilde{\pi}_s^{k+1})$$

  *# M-step: update estimation $\boldsymbol{\mu}^k$, posterior $p(\mathbf{w}^k), p(\boldsymbol{\Lambda}^k)$*

$$\boldsymbol{\mu}_m^k = \frac{1}{\widetilde{N}_m^k}\sum_{n=1}^N \widetilde{\gamma}_{nm}^k \mathbf{y}_n^k; \quad \widetilde{N}_m^k = \sum_{n=1}^N \widetilde{\gamma}_{nm}^k$$

$$p(\mathbf{w}^k) = \text{Dirichlet}(\mathbf{w}^k; \widetilde{\boldsymbol{\pi}}^k); \quad \widetilde{\pi}_m^k = \pi_m + \widetilde{N}_m^k$$

$$p(\boldsymbol{\Lambda}^k) = \prod_{m=1}^M \text{Gamma}(\Lambda_m^k; \widetilde{u}_m^k, \widetilde{v}_m^k); \widetilde{u}_m^k = u_m^k + \frac{\tau}{2}\widetilde{N}_m^k; \widetilde{v}_m^k = v_m^k + \frac{1}{2}\sum_{n=1}^N \widetilde{\gamma}_{nm}^k \|\mathbf{y}_n^k - \boldsymbol{\mu}_m^k\|_2^2$$

**end for**
$\text{Mode}_m = \{\mathbf{y}_n^0 | \arg\max_s \widetilde{\gamma}_{ns}^0 = m\}$
$P(\text{Mode}_m) = |\text{Mode}_m|/N$ and get statistics of each mode

---

**1) Mixture weights.** The weights $w_m$ remain constant across steps $k$. In practice, the number of effective modes should be limited, i.e., $w_m \approx 0$ for most modes, to avoid over-fragmented predictions.
**2) Covariance matrix.** The covariance matrix evolves with $k$ and should be $(1 - \bar{\alpha}_k)\mathbf{I}$ at step $k$.

Consequently, when drawing $N$ samples via reverse diffusion, the collection at step $k$, $\mathbf{Y}^k = \{\mathbf{y}_n^k\}_{n=1}^N$, should follow the Gaussian mixture distribution in Eq. 10. This observation leads to the use of GMM over other clustering models. By fitting a GMM alongside the reverse process at each step, we can recover $\{w_m, \mathbf{y}_m^*\}_{m=1}^M$ from the final GMM at step 0. To better leverage the priors on mixture weights and covariance, we employ a variational GMM rather than the standard GMM, which enables explicit prior injection. The prior at step $k$ is set as:

$$q(\mathbf{Y}^k | \mathbf{Z}^k, \boldsymbol{\mu}^k, \boldsymbol{\Lambda}^k) = \prod_{n=1}^N \prod_{m=1}^M \mathcal{N}(\mathbf{y}_n^k; \boldsymbol{\mu}_m^k, (\Lambda_m^k)^{-1}\mathbf{I})^{z_{nm}^k} \quad q(\mathbf{Z}^k | \mathbf{w}^k) = \prod_{n=1}^N \prod_{m=1}^M (w_m^k)^{z_{nm}^k}$$

#Prior for mixture weights: $q(\mathbf{w}^k) = \text{Dirichlet}(\mathbf{w}^k; \boldsymbol{\pi}), \pi_m = \rho^{m-1}$      (11)

#Prior for variance: $q(\boldsymbol{\Lambda}^k) = \prod_{m=1}^M \text{Gamma}(\Lambda_m^k; u_m^k, v_m^k), u_m^k = u, v_m^k = u * (1 - \bar{\alpha}_k)$

We use $q(\cdot)$ to denote prior distributions and $\{\pi_m, u_m^k, v_m^k\}_{m=1}^M$ without tilde for parameters in priors. $\mathbf{Z}^k = \{\mathbf{z}_n^k\}_{n=1}^N$, where each $\mathbf{z}_n^k$ is a one-hot latent variable indicating the mode $\mathbf{y}_n^k$ belongs to. Line 1 defines the GMM with means $\boldsymbol{\mu}^k = \{\boldsymbol{\mu}_m^k\}_{m=1}^M$, inverse variances $\boldsymbol{\Lambda}^k = \{\Lambda_m^k\}_{m=1}^M$ and mixture weights $\mathbf{w}^k = \{w_m^k\}_{m=1}^M$. Line 2 assigns a constant Dirichlet prior w.r.t diffusion step $k$ for $\mathbf{w}^k$. Its parameter $\pi_m$ decays by $\rho$ when mode index $m$ increases, encouraging higher-indexed modes to vanish. Line 3 assigns a Gamma prior on $\boldsymbol{\Lambda}^k$ such that $(\mathbb{E}[\Lambda_m^k])^{-1} = v_m^k/u_m^k = 1 - \bar{\alpha}_k$, consistent with the covariance prior. No prior is set for $\boldsymbol{\mu}^k$ as it is related to the unknown $\mathbf{y}_m^*$. The only hyperparameters are $\rho$ and $u$, with $\rho$ controlling the decay of the Dirichlet prior over mixture weights and $u$ setting the shape of the Gamma prior on variances. Their effects are evaluated in Appenidx F.

With these priors, we get the multi-mode inference Algorithm 1, with detailed derivations in Appendix C. $p(\cdot)$ denote posterior distributions and $\{\widetilde{\pi}_m^k, \widetilde{u}_m^k, \widetilde{v}_m^k\}_{m=1}^M$ with tilde are for parameters in posteriors. As shown in Fig. 3(c), rather than using standard GMM as a post-processing method only applied to $\{\mathbf{y}_n^0\}_{n=1}^N$, we distribute GMM iterations across diffusion steps. This allows us to utilize the knowledge from Eq. 10 via evolving prior distributions, which mitigates the difficulty of GMM initialization and makes a more informed choice of the number of active modes.

Table 1: Top-3 MSE/MAE, MSE, CRPS evaluations of different losses. The forecasting horizon $\tau$ is {96, 192, 336, 720} for the first seven datasets and {60, 120, 180, 300} for Dynamic. Results are averaged over 4 horizons. Rank: average rank on 8 datasets. Bold/underline: best/second. Inf: infinity problem caused by outliers. See Table 7 in Appendix E for full results.

| Dataset | ETTh1 | ETTm1 | ETTh2 | ETTm2 | WTH | ECL | Traffic | Dynamic | Rank | ETTh1 | ETTm1 | ETTh2 | ETTm2 | WTH | ECL | Traffic | Dynamic | Rank |
|---|---|---|---|---|---|---|---|---|---|---|---|---|---|---|---|---|---|---|
| Metric | | | | Top-3 MSE | | | | | | | | | Top-3 MAE | | | | | |
| MSE | 0.430 | 0.348 | 0.364 | 0.264 | 0.224 | 0.176 | 0.433 | 0.336 | 3.5 | 0.440 | 0.381 | 0.398 | 0.320 | 0.262 | 0.278 | 0.326 | 0.311 | 4.875 |
| MAE | 0.441 | 0.364 | 0.368 | 0.276 | 0.235 | 0.179 | 0.449 | 0.426 | 5.25 | 0.437 | 0.375 | 0.392 | 0.321 | 0.263 | 0.272 | 0.310 | 0.295 | 4.125 |
| Gaussian | 0.439 | 0.361 | 0.379 | 0.280 | 0.255 | 0.165 | 0.419 | 0.343 | 4.75 | 0.437 | 0.386 | 0.411 | 0.337 | 0.292 | 0.256 | 0.282 | 0.309 | 5.125 |
| Student-T | 0.430 | 0.352 | 0.375 | 0.286 | 0.241 | 0.165 | 0.416 | 0.390 | 4.25 | 0.428 | 0.371 | 0.398 | 0.333 | 0.263 | 0.250 | 0.261 | 0.292 | 3.375 |
| Mix | 0.425 | 0.289 | 0.343 | 0.245 | 0.209 | 0.147 | 0.412 | 0.322 | 1.875 | 0.426 | 0.338 | 0.387 | 0.308 | 0.242 | 0.240 | 0.261 | 0.246 | 2 |
| MMPD | **0.396** | **0.269** | **0.299** | **0.214** | **0.193** | 0.147 | **0.389** | **0.301** | **1** | **0.412** | **0.331** | **0.357** | **0.285** | **0.221** | **0.238** | **0.254** | **0.207** | **1** |
| Metric | | | | MSE | | | | | | | | | CRPS | | | | | |
| MSE | 0.425 | 0.350 | 0.376 | 0.270 | 0.227 | **0.160** | **0.399** | 0.345 | 1.875 | 0.337 | 0.307 | 0.308 | 0.247 | 0.218 | 0.270 | 0.343 | 0.257 | 4.25 |
| MAE | 0.432 | 0.355 | 0.366 | 0.274 | 0.233 | 0.164 | 0.417 | 0.426 | 3.5 | 0.346 | 0.313 | **0.299** | 0.247 | 0.220 | 0.288 | 0.362 | 0.275 | 4.625 |
| Gaussian | 0.434 | 0.357 | 0.382 | 0.284 | 0.255 | 0.163 | 0.413 | 0.349 | 4 | 0.317 | 0.282 | 0.315 | 0.256 | 0.228 | 0.190 | 0.217 | 0.233 | 4.375 |
| Student-T | 0.426 | 0.349 | 0.372 | 0.283 | 0.241 | 0.164 | 0.418 | 0.392 | 3.5 | **0.310** | 0.271 | 0.300 | 0.250 | 0.201 | 0.187 | 0.204 | 0.224 | 2.25 |
| Mix | 0.446 | 0.358 | 0.390 | 0.285 | 0.259 | 0.167 | 0.426 | 0.482 | 6 | 0.316 | **0.269** | 0.310 | 0.247 | 0.209 | Inf | 0.205 | 0.224 | 3 |
| MMPD | **0.412** | **0.337** | **0.354** | **0.264** | 0.229 | 0.164 | 0.409 | 0.353 | 1.75 | 0.318 | 0.270 | 0.301 | **0.243** | **0.199** | 0.191 | **0.202** | **0.203** | 2 |

Table 2: MSE, Mix and MMPD losses over backbones: Crossformer, SegRNN and MaskAE. Horizons are consistent with those in Table 1. #1st: number of first ranks across 8 datasets. Inf: infinity problem caused by outliers. See Table 8&9 in Appendix E for the full results.

| Dataset | | ETTh1 | ETTm1 | ETTh2 | ETTm2 | WTH | ECL | Traffic | Dynamic | #1st | ETTh1 | ETTm1 | ETTh2 | ETTm2 | WTH | ECL | Traffic | Dynamic | #1st |
|---|---|---|---|---|---|---|---|---|---|---|---|---|---|---|---|---|---|---|---|
| Metric | | | | | Top-3 MSE | | | | | | | | | Top-3 MAE | | | | | |
| Crossformer | MSE | 0.443 | 0.378 | 0.372 | 0.266 | 0.223 | 0.184 | 0.451 | 0.331 | 0 | 0.452 | 0.397 | 0.412 | 0.323 | 0.268 | 0.288 | 0.342 | 0.304 | 0 |
| | Mix | 0.433 | 0.330 | 0.359 | 0.236 | 0.200 | 0.160 | 0.424 | 0.307 | 0 | 0.433 | 0.358 | 0.400 | 0.307 | 0.235 | 0.255 | 0.279 | 0.233 | 0 |
| | MMPD | **0.381** | **0.310** | **0.315** | **0.228** | **0.197** | **0.152** | **0.404** | **0.295** | **8** | **0.410** | **0.353** | **0.372** | **0.295** | **0.226** | **0.245** | **0.245** | **0.194** | **8** |
| SegRNN | MSE | 0.440 | 0.385 | 0.365 | 0.273 | 0.222 | 0.183 | 0.464 | 0.333 | 0 | 0.451 | 0.417 | 0.408 | 0.337 | 0.269 | 0.286 | 0.333 | 0.307 | 0 |
| | Mix | 0.435 | 0.335 | 0.341 | 0.248 | 0.214 | 0.155 | 0.439 | 0.330 | 0 | 0.432 | 0.378 | 0.389 | 0.311 | 0.246 | 0.245 | **0.253** | 0.247 | 1 |
| | MMPD | **0.402** | **0.321** | **0.321** | **0.233** | **0.201** | **0.150** | **0.418** | **0.295** | **8** | **0.426** | **0.375** | **0.377** | **0.305** | **0.234** | **0.242** | 0.242 | **0.210** | **7** |
| MaskAE | MSE | 0.438 | 0.354 | 0.355 | 0.280 | 0.226 | 0.178 | 0.436 | 0.339 | 0 | 0.447 | 0.389 | 0.392 | 0.337 | 0.263 | 0.280 | 0.329 | 0.312 | 0 |
| | Mix | 0.415 | 0.314 | 0.340 | 0.253 | 0.203 | 0.150 | 0.416 | 0.321 | 0 | 0.425 | 0.353 | 0.384 | 0.308 | 0.238 | 0.241 | 0.265 | 0.244 | 0 |
| | MMPD | **0.399** | **0.280** | **0.311** | **0.247** | **0.197** | **0.144** | **0.387** | **0.296** | **8** | **0.416** | **0.342** | **0.367** | **0.304** | **0.225** | **0.234** | **0.253** | **0.203** | **8** |
| Metric | | | | | MSE | | | | | | | | | CRPS | | | | | |
| Crossformer | MSE | 0.440 | **0.382** | 0.388 | 0.271 | **0.228** | **0.170** | 0.418 | **0.339** | 5 | 0.344 | 0.316 | 0.319 | 0.249 | 0.221 | 0.274 | 0.348 | 0.253 | 0 |
| | Mix | 0.460 | 0.399 | 0.403 | 0.273 | 0.241 | 0.182 | 0.459 | 0.455 | 0 | 0.323 | 0.281 | **0.248** | **0.202** | Inf | 0.217 | Inf | 0.194 | 3 |
| | MMPD | **0.416** | 0.388 | **0.374** | **0.270** | 0.232 | **0.170** | **0.420** | 0.349 | 4 | **0.314** | **0.287** | **0.316** | 0.249 | **0.204** | **0.197** | **0.208** | **0.194** | 5 |
| SegRNN | MSE | **0.433** | 0.383 | 0.376 | **0.279** | 0.226 | 0.168 | 0.428 | 0.342 | 3 | 0.341 | 0.321 | 0.314 | 0.259 | 0.221 | 0.273 | 0.345 | 0.255 | 0 |
| | Mix | 0.452 | **0.381** | **0.375** | 0.285 | 0.273 | 0.187 | 0.468 | 0.465 | 2 | Inf | Inf | Inf | Inf | Inf | Inf | Inf | Inf | 0 |
| | MMPD | 0.434 | 0.386 | 0.376 | 0.285 | 0.232 | **0.167** | **0.421** | **0.341** | 3 | **0.328** | **0.299** | **0.313** | **0.257** | **0.210** | **0.193** | **0.205** | **0.204** | **8** |
| MaskAE | MSE | 0.437 | 0.357 | 0.366 | 0.287 | **0.230** | 0.162 | **0.403** | 0.349 | 3 | 0.341 | 0.310 | **0.303** | 0.259 | 0.220 | 0.271 | 0.344 | 0.259 | 1 |
| | Mix | 0.456 | 0.371 | 0.394 | 0.291 | 0.270 | 0.170 | 0.441 | 0.476 | 0 | **0.319** | **0.277** | Inf | **0.250** | Inf | 0.194 | 0.208 | 0.222 | 3 |
| | MMPD | **0.421** | **0.342** | **0.362** | **0.281** | 0.231 | **0.161** | 0.404 | 0.350 | 5 | **0.319** | **0.277** | 0.305 | 0.258 | **0.201** | **0.188** | **0.201** | **0.202** | 6 |

## 4 EXPERIMENTS

We conduct experiments on: **ETTh1, ETTm1, ETTh2, ETTm2, WTH, ECL, Traffic, Dynamic**. The first seven are widely used datasets from previous works (Nie et al., 2023). The new Dynamic consists of 17 signals from a complex dynamical system without obvious periodic patterns. For each dataset, we fix the look-back window $T$ and make predictions on different horizons $\tau$.

Following Top-K accuracy for image classification (He et al., 2016), we use **Top-K MSE/MAE** ($K = 3$ in our setting) to evaluate multi-mode prediction: Top-K modes with the highest probabilities are selected and the minimum MSE/MAE among K modes is reported. Using small K and guided by the probability of each mode, Top-K MSE is more applicable than Best MSE (Le Guen & Thome, 2020), which computes the MSE of all $N$ samples and reports the best. We also report traditional metrics such as **MSE** and **Continuous Ranked Probability Score (CRPS)** to evaluate deterministic and probabilistic accuracy. Detailed setups, including datasets and metrics, are shown in Appendix D.

### 4.1 MAIN RESULTS

**MMPD *vs.* Baseline Losses.** We compare MMPD with the following losses: 1) deterministic losses **MSE** (Nie et al., 2023) and **MAE** (Liu et al., 2022a); 2) distribution-based losses **Gaussian** (Salinas et al., 2020) and **Student-T** (Rasul et al., 2023); 3) **Mix** (Woo et al., 2024) that mixes multiple parametric distributions for flexible modeling. We maintain the main backbone as a patch-based decoder-only Transformer (Goswami et al., 2024; Lin et al., 2024b) and change the losses.

Top-3 MSE and Top-3 MAE in Table 1 show that only Mix and MMPD can capture multi-mode patterns. Among them, our MMPD loss consistently outperforms Mix, as the number and form of mixture components in Mix are predefined, while in MMPD, they are learned directly from the data. Regarding deterministic forecasting performance measured by MSE, our MMPD loss is

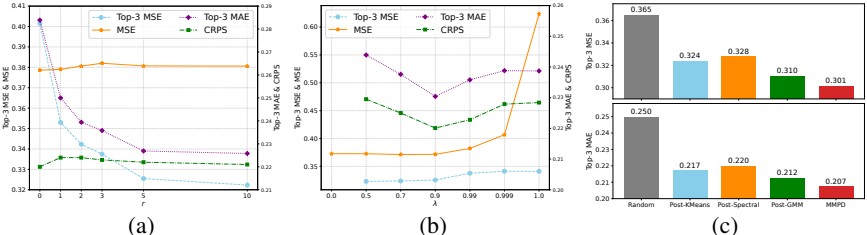

Figure 4: (a) Metrics for varying adjacent range $r$ in Patch Consistent MLP ($r = 0$: independent MLP) on Dynamic (prediction horizon $\tau = 180$). (b) Metrics for varying probabilistic/deterministic objective balancing weight $\lambda$, with all other settings identical to (a). (c) Top-3 MSE/MAE comparison between multi-mode inference and post-processing methods.

comparable to the best competitor, MSE loss, and even outperforms it on some datasets, showing MMPD effectively integrates deterministic forecasting. Similarly, MMPD performs on par with the best-performing baseline, Student-T, in terms of probabilistic forecasting measured by CRPS.

**Generality of MMPD Loss across Backbones.** Besides the decoder-only Transformer used in Table 1, we also compare MMPD loss with MSE and Mix across the following three backbones: 1) channel-mixing Transformer **Crossformer** (Zhang & Yan, 2023), 2) patch-based RNN **SegRNN** (Lin et al., 2023), 3) Masked AutoEncoder using pure Transformers **MaskAE** (Zhang et al., 2024b). Results in Table 2 demonstrate that the diverse forecasting capability of MMPD significantly outperforms MSE and Mix across all three backbones. The deterministic forecasting ability measured by MSE is comparable to MSE loss, which is consistent with Table 1. It is worth noting that, due to the log-normal component, Mix loss is likely to generate outliers, leading to the infinity problem in CRPS. This issue is particularly severe for the RNN-based SegRNN. In contrast, the CRPS of the MMPD loss remains stable, regardless of whether the upstream backbone is an RNN or a Transformer.

## 4.2 MODEL ANALYSIS

**Ablation of Patch Consistent MLP.** In Fig. 4(a), the independent MLP, which does not incorporate adjacent patches ($r = 0$), performs poorly in multi-mode prediction, with the Top-3 MSE even exceeding MSE. This occurs because the independent MLP only models the marginal distribution of each patch $p_\theta(\mathbf{p}_j|\mathbf{x})$ rather than the joint distribution of all patches, leading to inconsistent samples as shown in Fig. 3(a). In contrast, the Patch Consistent MLP, even with $r = 1$, significantly reduces both Top-3 MSE and Top-3 MAE. Further increasing $r$ slightly improves performance.

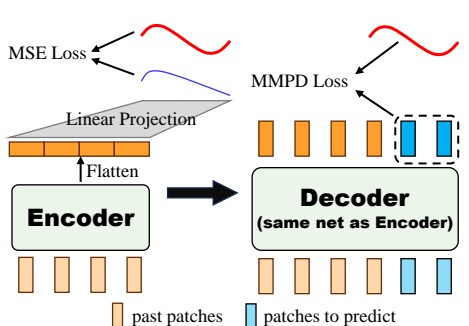

Figure 5: Approach for adapting encoder-only backbones to use MMPD loss: learnable tokens, indicating the patches to predict, are appended to the end of the past patch sequence. The padded sequence is fed to the same network as encoder, transforming the backbone into a decoder-only model. Only the output tokens corresponding to the future series are used for MMPD loss computation.

**Effect of Balancing Weight $\lambda$ in Eq. 8.** Fig. 4(b) shows that when training with only the diffusion objective ($\lambda = 1.0$), the deterministic prediction capability measured by MSE is poor. Decreasing it to 0.999, the MSE is greatly reduced, while other metrics are not harmed. As $\lambda$ gradually decreases, MSE improves and stabilizes. An interesting observation is that when $\lambda$ decreases from 1.0 to 0.9, besides MSE, other multi-mode and probabilistic metrics also get better. We suspect that this is the collaborative effect brought by joint training of the two objectives.

**Ablation of Multi-Mode Inference Algorithm.** Fig. 4(c) compares our multi-mode inference algorithm with various post-processing methods. Random assignment performs the worst, highlighting the necessity of multi-mode extraction. Our multi-mode inference algorithm significantly outperforms KMeans and spectral clustering. Furthermore, MMPD surpasses Post-GMM, which uses the same GMM formulation as MMPD but is applied directly to final samples $\{\mathbf{y}_n^0\}_{n=1}^N$ without the evolving priors. This stems from fitting an evolving GMM with dynamic priors on gradually denoised samples, which mitigates the

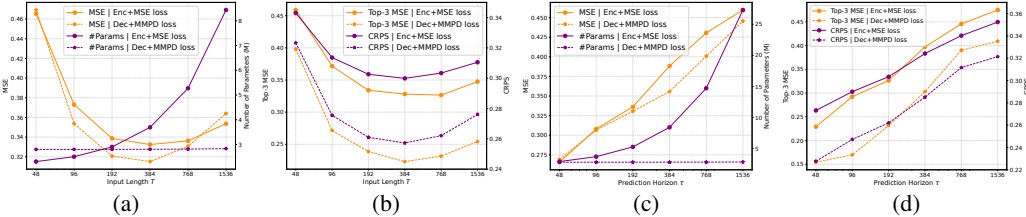

Figure 6: Comparison of adapted decoder-only PatchTST (Nie et al., 2023) with MMPD loss (Dec+MMPD loss) against the original encoder-only version with MSE loss (Enc+MSE loss). (a), (b): number of parameters and metrics (MSE, Top-3 MSE and CRPS) for varying input length $T$ on ETTm1, the prediction horizon is set to $\tau = 192$. (c), (d): number of parameters and metrics for varying prediction horizon $\tau$ on ETTm1, the input length is set to $T = 768$.

difficulty of parameter initialization in GMM and automatically selects a more appropriate number of activated modes. Due to page limit, other hyperparameter evaluations (e.g., noise schedule, diffusion steps, hyperparameters in Algorithm 1) are provided in Appendix F.

**Adapting Encoder-Only Backbones for MMPD Loss.** In the left of Fig. 5, besides backbones we have evaluated, there are encoder-only backbones that flatten the encoder outputs and linearly project them to predict the series for MSE loss (Nie et al., 2023; Luo & Wang, 2024). They do not generate future latent tokens, meaning MMPD loss cannot be directly applied. In the right of Fig. 5, we transform them into decoder-only ones for our MMPD loss by appending learnable tokens to the end of the input sequence.

We adapt the encoder-only PatchTST (Nie et al., 2023) into a decoder-only backbone to enable MMPD. In Fig. 6(a)&6(c), the scale of the projection layer in encoder-only version increases with input/output lengths, while remaining constant with our adaptation. Decoder-only PatchTST with MMPD gets lower MSE, indicating better scalability. Also, our adaptation enables multi-mode and probabilistic forecasting, evidenced by improved Top-3 MSE and CRPS in Fig. 6(b)&6(d).

**MMPD Loss *vs.* Standalone TS Diffusion Models.** We also compare MMPD loss with standalone TS diffusion models **1)CSDI** (Tashiro et al., 2021), **2)TSDiff** (Kollovieh et al., 2023), **3)MG-TSD** (Fan et al., 2024), **4)Diffusion-TS** (Yuan & Qiao, 2024), **5)D$^3$U** (Li et al., 2025). These models involve complex denoising networks, making long-term experiments conducted in Sec. 4.1 challenging. Following Tashiro et al. (2021), we forecast the next 24 steps using the past 96 on ETTh1. Results in Table 3 show that our MMPD significantly outperforms standalone diffusion models. This is because the decoupling from the backbone allows the MMPD loss to fully leverage the advanced backbone. Moreover, the lightweight denoising MLP in MMPD ensures faster inference compared to diffusion models with heavier networks and complex diffusion processes.

Table 3: Evaluation of MMPD loss against TS diffusion models on ETTh1, $T = 96, \tau = 24$. "Time" refers to average inference time per instance.

|  | Top-3 MSE | Top-3 MAE | MSE | CRPS | Time(s) |
|---|---|---|---|---|---|
| CSDI | 0.225 | 0.304 | 0.339 | 0.265 | 1.014 |
| TSDiff | 0.275 | 0.336 | 0.345 | 0.292 | 0.419 |
| MG-TSD | 0.287 | 0.331 | 0.340 | 0.306 | 0.217 |
| Diffusion-TS | 0.282 | 0.324 | 0.351 | 0.294 | 0.520 |
| D$^3$U | 0.244 | 0.344 | 0.338 | 0.285 | 0.453 |
| Decoder+MMPD | **0.186** | **0.280** | **0.298** | **0.254** | **0.075** |

**Efficiency Analysis.** As shown in Table 4, the Patch Consistent MLP for MMPD incurs marginally higher FLOPs than the conventional MLP for MSE loss. During training, MSE loss requires one MLP forward pass, while MMPD loss requires two: one for the diffusion objective and another for the deterministic in Eq. 8. However, since the backbone dominates training cost (i.e., $F_{bkb} >> F_{MLP}, F_{PC-MLP}$), the total training FLOPs of MMPD remain nearly identical to those of MSE. This equivalence also holds for deterministic inference, where both losses require a single MLP pass. For probabilistic and multi-mode inference, which MSE cannot perform, MMPD's overhead is significantly lower than that of standalone TS Diffusion models, as MMPD only requires a single heavy backbone pass followed by multiple lightweight MLP passes. The theoretical analysis is consistently supported by the experimental memory occupancy and speed measurements in Table 5.

Table 4: FLOPs across structures/stages for MSE Loss, TS Diffusion Models and MMPD Loss. $S$: number of blocks in MLPs, $d$ (short for $d_{model}$): hidden state dimension. $F_{bkb}, F_{MLP}, F_{PC-MLP}$: FLOPs of backbone, conventional MLP and Patch Consistent MLP. GMM FLOPs in MMPD are omitted as they are negligible vs. neural networks. See derivation in Appendix G.

| Structure/Stage | MSE Loss | TS Diffusion Models | MMPD Loss |
|---|---|---|---|
| MLP Projector | $F_{MLP} = O(\frac{\tau}{P}[(2S+1)d^2 + Pd])$ | N/A | $F_{PC-MLP} = O(\frac{\tau}{P}[(5S+3)d^2 + (2r+2)Pd])$ |
| Training (forward only) | $F_{bkb} + F_{MLP}$ | $F_{bkb}$ | $F_{bkb} + 2F_{PC-MLP}$ |
| Deterministic Inference | $F_{bkb} + F_{MLP}$ | N/A | $F_{bkb} + F_{PC-MLP}$ |
| Prob/Multi-Mode Inference | N/A | $NKF_{bkb}$ | $F_{bkb} + NKF_{PC-MLP}$ |

Table 5: Memory, training (per batch) and inference time (per instance) of MSE Loss, Diffusion-TS and MMPD Loss on dataset WTH, $T = 336, \tau = 192, \text{batch} = 32, N = 100, K = 20$.

| Stage | MSE Loss | | Diffusion-TS | | MMPD Loss | |
|---|---|---|---|---|---|---|
| | Memory (GB) | Time (ms) | Memory (GB) | Time (ms) | Memory (GB) | Time (ms) |
| Training | 2.599 | 89.9 | 4.358 | 676.4 | 2.930 | 106.3 |
| Deterministic Infer | 0.031 | 2.3 | N/A | | 0.034 | 3.1 |
| Prob/Multi-Mode Infer | N/A | | 11.245 | 28,495.1 | 0.505 | 415.8 |

## 5 FURTHER DISCUSSIONS AND CONCLUSION

In Appendix H, we extend MMPD beyond its basic setting to non-patch-based backbones by inserting a Transformer decoder layer between the backbone and MMPD loss. In Appendix I, beyond the single-dataset paradigm, we further apply MMPD to a multi-task model, UNITS (Gao et al., 2024), to perform multi-task, few-shot and zero-shot forecasting. In both cases, MMPD functions as a plug-and-play loss that can be incorporated with minimal changes. Compared with original models trained with MSE loss, MMPD preserves deterministic forecasting performance while enabling richer distribution modeling, supporting multi-mode and probabilistic forecasting.

In conclusion, we have proposed MMPD, a diffusion-based loss that goes beyond the dominant single-mode MSE loss in TS forecasting. By modeling complex distributions, MMPD equips models with multi-mode forecasting capabilities—an essential feature for many real-world applications, particularly those involving risk-aware decision making. Extensive benchmark experiments demonstrate the effectiveness and broad applicability of our approach.

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

## A    RELATED WORKS

**Backbones for TS Forecasting.** Early works employ RNNs (Flunkert et al., 2017) and CNNs (Lea et al., 2017) as backbones. Transformers for TS were introduced later, incorporating techniques such as sparse attention (Li et al., 2019; Zhou et al., 2021), trend-season decomposition (Wu et al., 2021), frequency enhancement (Zhou et al., 2022) and hierarchical structure (Liu et al., 2022b). Nie et al. (2023); Zhang & Yan (2023) proposed patch-based Transformers that divide time series into patches. This approach was later adopted widely, resulting in the development of various patch-based models (Vijay et al., 2023; Lin et al., 2023; Luo & Wang, 2024; Zhang et al., 2023; Yu et al., 2023; Zhang et al., 2024a). Furthermore, many recent pre-training models (Zhang et al., 2024b; Woo et al., 2024; Goswami et al., 2024; Liu et al., 2024b) and cross-modality models (Liu et al., 2024a; Zhou et al., 2023; Jin et al., 2024) also adopt patch-based backbones. Other studies have explored lightweight designs (Zeng et al., 2023; Lin et al., 2024a) and cross-channel dependency capture (Liu et al., 2023; Huang et al., 2023). Despite such advances in backbones, they mostly use regression loss functions. This greatly limits the backbones' capability, especially in diverse forecasting.

**Loss Functions for TS Forecasting.** Despite extensive research on backbones, limited focus was paid to forecasting specific losses. A line of works uses Dynamic Time Warping (DTW) (Müller, 2007), which computes the similarity between two series with dynamic programming. Cuturi & Blondel (2017) makes DTW differentiable. Le Guen & Thome (2019; 2020) extend DTW to evaluate shape and temporal distortions. However, it is hard to scale non-parallelizable DTW-based losses to long-term tasks. Another line predicts future distributions by estimating their parameters. Common distributions include Student-T (Rasul et al., 2023), Gaussian and negative binomial (Salinas et al., 2020). Additionally, Woo et al. (2024) mix multiple parametric distributions (e.g., Gaussian, Log-normal, etc.) via a Softmax layer. However, these methods rely on predefined formulations, failing to capture complex patterns.

**Deep Generative Models for TS.** Pioneering efforts on deep generative models for TS include GANs (Yoon et al., 2019), normalizing flows (Rasul et al., 2021b) and VAEs (Li et al., 2021). With the success of diffusion models (Ho et al., 2020; Peebles & Xie, 2023), many diffusion models for TS have also been proposed. Rasul et al. (2021a) propose an RNN-based diffusion model. Tashiro et al. (2021); Alcaraz & Strodthoff (2022) condition diffusion on observed data for imputation. Shen & Kwok (2023) and Li et al. (2024b) respectively use the prediction of autoregressive models and Transformers as the prior knowledge to guide diffusion. Yuan & Qiao (2024) introduces trend-season decomposition to enhance diffusion. Li et al. (2025) propose a two-stage framework that predicts the deterministic component via point forecasting and models the probabilistic residual with diffusion. These efforts primarily focus on refining denoising networks or optimizing the diffusion process. In contrast, our approach leverages diffusion models to develop a backbone-agnostic loss function.

## B    DETAILS OF ADALN-MLP

The AdaLN-MLP is an MLP that takes a noisy patch $\mathbf{p}$ and the condition vector $\mathbf{c}$ as input and predicts the noise in $\mathbf{p}$. It is originally a component of Diffusion Transformer (Peebles & Xie, 2023). The Diffusion Transformer was designed to replace the U-Net backbone in diffusion models, and it introduced Adaptive LayerNorm (AdaLN) blocks to inject conditions into diffusion models. One AdaLN block consists of an AdaLN-Attention block and an AdaLN-MLP block. For efficiency, we only use the AdaLN-MLP in our MMPD loss. The computation process of one AdaLN-MLP block is as follows:

$$\mathbf{z}^{(s)} = \mathbf{z}^{(s-1)} + \boldsymbol{\alpha}_{\text{gate}}^{(s)} \circ \text{MLP}\left(\text{AdaLN}(\mathbf{z}^{(s-1)}, \boldsymbol{\gamma}_{\text{scale}}^{(s)}, \boldsymbol{\beta}_{\text{shift}}^{(s)})\right)$$
$$\text{AdaLN}(\mathbf{z}, \boldsymbol{\gamma}, \boldsymbol{\beta}) = (1 + \boldsymbol{\gamma}) \circ \text{LayerNorm}(\mathbf{z}) + \boldsymbol{\beta} \qquad (12)$$
$$\boldsymbol{\alpha}_{\text{gate}}^{(s)} = \mathbf{W}_{\text{gate}}^{(s)} \phi(\mathbf{c}), \quad \boldsymbol{\gamma}_{\text{scale}}^{(s)} = \mathbf{W}_{\text{scale}}^{(s)} \phi(\mathbf{c}), \quad \boldsymbol{\beta}_{\text{shift}}^{(s)} = \mathbf{W}_{\text{shift}}^{(s)} \phi(\mathbf{c})$$

$\mathbf{z}^{(s)} \in \mathbb{R}^{d_{model}}$ is the output of $s$-th block (with $\mathbf{z}^{(0)}$ being the linearly embedded $\mathbf{p}$) and $\mathbf{c} \in \mathbb{R}^{d_{model}}$ is the condition. $\circ$ denotes the element-wise product and $\text{MLP}(\cdot)$ is the standard multilayer perceptron. $\boldsymbol{\alpha}_{\text{gate}}^{(s)}, \boldsymbol{\gamma}_{\text{scale}}^{(s)}, \boldsymbol{\beta}_{\text{shift}}^{(s)} \in \mathbb{R}^{d_{model}}$ are parameters to adjust the layer norm and they are obtained by projecting the condition $\phi(\mathbf{c})$ with $\mathbf{W}_{\text{gate}}^{(s)}, \mathbf{W}_{\text{scale}}^{(s)}, \mathbf{W}_{\text{shift}}^{(s)} \in \mathbb{R}^{d_{model} \times d_{model}}$ respectively, where $\phi(\cdot)$

is the activation function. Passing through $S$ AdaLN-MLP blocks, $\mathbf{z}^{(S)}$ is used to make the final prediction by:

$$
\begin{aligned}
\hat{\epsilon} &= \mathbf{W}^{(\text{final})}\text{AdaLN}(\mathbf{z}^{(S)}, \boldsymbol{\gamma}^{(\text{out})}_{\text{scale}}, \boldsymbol{\beta}^{(\text{out})}_{\text{shift}}) \\
\boldsymbol{\gamma}^{(\text{out})}_{\text{scale}} &= \mathbf{W}^{(\text{out})}_{\text{scale}}\phi(\mathbf{c}), \quad \boldsymbol{\beta}^{(\text{out})}_{\text{shift}} = \mathbf{W}^{(\text{out})}_{\text{shift}}\phi(\mathbf{c})
\end{aligned}
\tag{13}
$$

where $\mathbf{W}^{(\text{final})} \in \mathbb{R}^{P \times d_{model}}$ and $\mathbf{W}^{(\text{out})}_{\text{scale}}, \mathbf{W}^{(\text{out})}_{\text{shift}} \in \mathbb{R}^{d_{model} \times d_{model}}$.

## C  DERIVATION OF MULTI-MODE INFERENCE ALGORITHM

To leverage the prior knowledge from forward diffusion, we set the following prior at step $k$:

$$
\begin{aligned}
q(\mathbf{Y}^k|\mathbf{Z}^k, \boldsymbol{\mu}^k, \boldsymbol{\Lambda}^k) &= \prod_{n=1}^{N} \prod_{m=1}^{M} \mathcal{N}(\mathbf{y}_n^k; \boldsymbol{\mu}_m^k, (\Lambda_m^k)^{-1}\mathbf{I})^{z_{nm}^k} \\
q(\mathbf{Z}^k|\mathbf{w}^k) &= \prod_{n=1}^{N} \prod_{m=1}^{M} (w_m^k)^{z_{nm}^k} \\
q(\mathbf{w}^k) &= \text{Dirichlet}(\mathbf{w}^k; \boldsymbol{\pi}), \pi_m = \rho^{m-1} \\
q(\boldsymbol{\Lambda}^k) &= \prod_{m=1}^{M} \text{Gamma}(\Lambda_m^k; u_m^k, v_m^k) \\
u_m^k &= u, v_m^k = u * (1 - \bar{\alpha}_k)
\end{aligned}
\tag{14}
$$

Note that we use $q(\cdot)$ to denote prior distributions and $\{\pi_m^k, u_m^k, v_m^k\}_{m=1}^{M}$ without tilde for parameters in prior distributions. $p(\cdot)$ are for posterior distributions and $\{\widetilde{\pi}_m^k, \widetilde{u}_m^k, \widetilde{v}_m^k\}_{m=1}^{M}$ for parameters in posterior distributions.

With the above priors, we get the joint distribution $q(\mathbf{Y}^k, \mathbf{Z}^k, \mathbf{w}^k, \boldsymbol{\Lambda}^k|\boldsymbol{\mu}^k) = q(\mathbf{Y}^k|\mathbf{Z}^k, \boldsymbol{\mu}^k, \boldsymbol{\Lambda}^k)q(\mathbf{Z}^k|\mathbf{w}^k)q(\mathbf{w}^k)q(\boldsymbol{\Lambda}^k)$. Obtaining samples at step $k$, $\mathbf{Y}^k = \{\mathbf{y}_n^k\}_{n=1}^{N}$, our goal is to maximize the marginal log probability $\max_{\boldsymbol{\mu}^k} \ln q(\mathbf{Y}^k|\boldsymbol{\mu}^k)$ and get the posterior distribution $q(\mathbf{Z}^k, \mathbf{w}^k, \boldsymbol{\Lambda}^k|\mathbf{Y}^k, \boldsymbol{\mu}^k_{opt})$, where $\boldsymbol{\mu}^k_{opt}$ denotes optimal parameters. It is hard to directly optimize the marginal distribution and obtain the posterior as they both contain complex integral terms. Therefore, we introduce variational distribution $p(\mathbf{Z}^k, \mathbf{w}^k, \boldsymbol{\Lambda}^k)$ to approximate the posterior through variational inference (Bishop & Nasrabadi, 2006). To maximize $\max_{\boldsymbol{\mu}^k} \ln q(\mathbf{Y}^k|\boldsymbol{\mu}^k)$ and approximate posterior $q(\mathbf{Z}^k, \mathbf{w}^k, \boldsymbol{\Lambda}^k|\mathbf{Y}^k, \boldsymbol{\mu}^k)$ with $p(\mathbf{Z}^k, \mathbf{w}^k, \boldsymbol{\Lambda}^k)$, we get the following objective:

$$
\begin{aligned}
&\max_{\boldsymbol{\mu}^k, p^k} \quad \ln q(\mathbf{Y}^k|\boldsymbol{\mu}^k) - \text{KL}[p(\mathbf{Z}^k, \mathbf{w}^k, \boldsymbol{\Lambda}^k)||q(\mathbf{Z}^k, \mathbf{w}^k, \boldsymbol{\Lambda}^k|\mathbf{Y}^k, \boldsymbol{\mu}^k)] \\
&= \max_{\boldsymbol{\mu}^k, p^k} \quad \int p(\mathbf{Z}^k, \mathbf{w}^k, \boldsymbol{\Lambda}^k)\left[\ln q(\mathbf{Y}^k|\boldsymbol{\mu}^k) - \ln \frac{p(\mathbf{Z}^k, \mathbf{w}^k, \boldsymbol{\Lambda}^k)}{q(\mathbf{Z}^k, \mathbf{w}^k, \boldsymbol{\Lambda}^k|\mathbf{Y}^k, \boldsymbol{\mu}^k)}\right] d\mathbf{Z}^k d\mathbf{w}^k d\boldsymbol{\Lambda}^k \\
&= \max_{\boldsymbol{\mu}^k, p^k} \quad \int p(\mathbf{Z}^k, \mathbf{w}^k, \boldsymbol{\Lambda}^k) \ln \frac{q(\mathbf{Y}^k, \mathbf{Z}^k, \mathbf{w}^k, \boldsymbol{\Lambda}^k|\boldsymbol{\mu}^k)}{p(\mathbf{Z}^k, \mathbf{w}^k, \boldsymbol{\Lambda}^k)} d\mathbf{Z}^k d\mathbf{w}^k d\boldsymbol{\Lambda}^k \\
&= \max_{\boldsymbol{\mu}^k, p^k} \quad \mathcal{L}(\boldsymbol{\mu}^k, p(\mathbf{Z}^k, \mathbf{w}^k, \boldsymbol{\Lambda}^k))
\end{aligned}
\tag{15}
$$

where $p^k$ is short for $p(\mathbf{Z}^k, \mathbf{w}^k, \boldsymbol{\Lambda}^k)$ and $\mathcal{L}(\boldsymbol{\mu}^k, p(\mathbf{Z}^k, \mathbf{w}^k, \boldsymbol{\Lambda}^k))$ is often called Evidence Lower Bound (ELBO) in variational inference. Using mean field approximation, we decompose the variational distribution into $p(\mathbf{Z}^k, \mathbf{w}^k, \boldsymbol{\Lambda}^k) = p(\mathbf{Z}^k)p(\mathbf{w}^k)p(\boldsymbol{\Lambda}^k)$.

Given newly generated samples at step $k$, $\mathbf{Y}^k$, we maximize $\mathcal{L}(\boldsymbol{\mu}^k, p(\mathbf{Z}^k), p(\mathbf{w}^k), p(\boldsymbol{\Lambda}^k))$ w.r.t $p(\mathbf{Z}^k), \boldsymbol{\mu}^k, p(\mathbf{w}^k), p(\boldsymbol{\Lambda}^k)$ one by one. An advantage of fitting an evolving variational GMM alongside the diffusion process is that we can use well-fitted posteriors, i.e., $p(\mathbf{Z}^{k+1}), \boldsymbol{\mu}^{k+1}, p(\mathbf{w}^{k+1}), p(\boldsymbol{\Lambda}^{k+1})$ at step $k + 1$ as the initialization of step $k$. This is because in diffusion, the difference between samples generated in two adjacent steps $\mathbf{Y}^{k+1}$ and $\mathbf{Y}^k$ is small, and in our setting, the difference between the prior distributions of two adjacent steps is also minimal. As a result, we initialize

$p(\mathbf{Z}^k), \boldsymbol{\mu}^k, p(\mathbf{w}^k), p(\boldsymbol{\Lambda}^k)$ using optimized parameters from last step $k+1$ as the following:

$$
\begin{aligned}
p(\mathbf{Z}^k) &= \prod_{n=1}^{N} \prod_{m=1}^{M} (\widetilde{\gamma}_{nm}^{k+1})^{z_{nm}^k} \\
\boldsymbol{\mu}^k &= \boldsymbol{\mu}^{k+1} \\
p(\mathbf{w}^k) &= \text{Dirichlet}(\mathbf{w}^k; \widetilde{\boldsymbol{\pi}}^{k+1}) \\
p(\boldsymbol{\Lambda}^k) &= \prod_{m=1}^{M} \text{Gamma}(\Lambda_m^k; \widetilde{u}_m^{k+1}, \widetilde{v}_m^{k+1})
\end{aligned}
\tag{16}
$$

**E-step: Update** $p(\mathbf{Z}^k)$. Fixing $\boldsymbol{\mu}^k, p(\mathbf{w}^k), p(\boldsymbol{\Lambda}^k)$, the objective becomes:

$$
\begin{aligned}
&\max_{p(\mathbf{Z}^k)} \quad \mathcal{L}(\boldsymbol{\mu}^k, p(\mathbf{Z}^k), p(\mathbf{w}^k), p(\boldsymbol{\Lambda}^k)) \\
&= \max_{p(\mathbf{Z}^k)} \quad \int_{\mathbf{Z}^k, \boldsymbol{\Lambda}^k} p(\mathbf{Z}^k) p(\boldsymbol{\Lambda}^k) \ln q(\mathbf{Y}^k | \mathbf{Z}^k, \boldsymbol{\mu}^k, \boldsymbol{\Lambda}^k) d\mathbf{Z}^k d\boldsymbol{\Lambda}^k \\
&\quad + \int_{\mathbf{Z}^k, \mathbf{w}^k} p(\mathbf{Z}^k) p(\mathbf{w}^k) \ln q(\mathbf{Z}^k | \mathbf{w}) d\mathbf{Z}^k d\mathbf{w}^k - \int_{\mathbf{Z}^k} p(\mathbf{Z}^k) \ln p(\mathbf{Z}^k) d\mathbf{Z}^k + \text{Const} \\
&= \min_{p(\mathbf{Z}^k)} \quad \text{KL}[p(\mathbf{Z}^k) || \widetilde{p}(\mathbf{Z}^k)]
\end{aligned}
\tag{17}
$$

where Const denotes constant terms w.r.t $p(\mathbf{Z}^k)$, $\widetilde{p}(\mathbf{Z}^k)$ is a new distribution with:

$$
\ln \widetilde{p}(\mathbf{Z}^k) = \mathbb{E}_{p(\boldsymbol{\Lambda}^k)}[\ln q(\mathbf{Y}^k | \mathbf{Z}^k, \boldsymbol{\mu}^k, \boldsymbol{\Lambda}^k)] + \mathbb{E}_{p(\mathbf{w}^k)}[\ln q(\mathbf{Z}^k | \mathbf{w}^k)] + \text{Const}
\tag{18}
$$

Therefore, KL divergence is minimized when $p(\mathbf{Z}^k) = \widetilde{p}(\mathbf{Z}^k)$:

$$
\begin{aligned}
&\ln p(\mathbf{Z}^k) = \ln \widetilde{p}(\mathbf{Z}^k) \\
&= \mathbb{E}_{p(\boldsymbol{\Lambda}^k)}[\sum_{n=1}^{N} \sum_{m=1}^{M} z_{nm}^k \ln \mathcal{N}(\mathbf{y}_n^k; \boldsymbol{\mu}_m^k, (\Lambda_m^k)^{-1}\mathbf{I})] + \mathbb{E}_{p(\mathbf{w}^k)}[\sum_{n=1}^{N} \sum_{m=1}^{M} z_{nm}^k \ln w_m^k] + \text{Const} \\
&= \sum_{n=1}^{N} \sum_{m=1}^{M} z_{nm}^k \ln \gamma_{nm}^k + \text{Const}
\end{aligned}
\tag{19}
$$

where

$$
\begin{aligned}
\ln \gamma_{nm}^k = &-\frac{1}{2}\left[\frac{\widetilde{u}_m^{k+1}}{\widetilde{v}_m^{k+1}} \|\mathbf{y}_n^k - \boldsymbol{\mu}_m^{k+1}\|_2^2 + \tau \ln(2\pi)\right] + \frac{\tau}{2}\left[\psi(\widetilde{u}_m^{k+1}) - \ln(\widetilde{v}_m^{k+1})\right] \\
&+ \psi(\widetilde{\pi}_m^{k+1}) - \psi(\sum_{s=1}^{M} \widetilde{\pi}_s^{k+1})
\end{aligned}
\tag{20}
$$

Normalizing it, we get the formulation of $p(\mathbf{Z}^k)$ :

$$
p(\mathbf{Z}^k) = \prod_{n=1}^{N} \prod_{m=1}^{M} (\widetilde{\gamma}_{nm}^k)^{z_{nm}^k}, \quad \widetilde{\gamma}_{nm}^k = \frac{\gamma_{nm}^k}{\sum_{s=1}^{M} \gamma_{ns}^k}
\tag{21}
$$

which means that the posterior probability of $\mathbf{y}_n^k$ belonging to mode $m$ is $p(z_{nm}^k = 1) = \widetilde{\gamma}_{nm}^k$.

**M-step: Update** $\boldsymbol{\mu}^k$. Note that $p(\mathbf{Z}^k)$ has already be updated. Fixing $p(\mathbf{Z}^k), p(\mathbf{w}^k), p(\boldsymbol{\Lambda}^k)$, the objective becomes:

$$
\begin{aligned}
&\max_{\boldsymbol{\mu}^k} \quad \mathcal{L}(\boldsymbol{\mu}^k, p(\mathbf{Z}^k), p(\mathbf{w}^k), p(\boldsymbol{\Lambda}^k)) \\
&= \max_{\boldsymbol{\mu}^k} \quad \int_{\mathbf{Z}^k, \boldsymbol{\Lambda}^k} p(\mathbf{Z}^k) p(\boldsymbol{\Lambda}^k) \ln q(\mathbf{Y}^k | \mathbf{Z}^k, \boldsymbol{\mu}^k, \boldsymbol{\Lambda}^k) d\mathbf{Z}^k d\boldsymbol{\Lambda}^k + \text{Const} \\
&= \max_{\boldsymbol{\mu}^k} \quad \mathbb{E}_{p(\mathbf{Z}^k), p(\boldsymbol{\Lambda}^k)}[\sum_{n=1}^{N} \sum_{m=1}^{M} z_{nm}^k \ln \mathcal{N}(\mathbf{y}_n^k; \boldsymbol{\mu}_m^k, (\Lambda_m^k)^{-1}\mathbf{I})] \\
&= \min_{\boldsymbol{\mu}^k} \quad \sum_{m=1}^{M} \frac{\widetilde{u}_m^{k+1}}{\widetilde{v}_m^{k+1}} \sum_{n=1}^{N} \widetilde{\gamma}_{nm}^k \|\mathbf{y}_n^k - \boldsymbol{\mu}_m^k\|_2^2
\end{aligned}
\tag{22}
$$

Setting the gradient w.r.t $\boldsymbol{\mu}_m^k$ to zero, we get the updated $\boldsymbol{\mu}^k$:

$$\boldsymbol{\mu}_m^k = \frac{1}{\widetilde{N}_m^k} \sum_{n=1}^N \widetilde{\gamma}_{nm}^k \mathbf{y}_n^k, \quad \widetilde{N}_m^k = \sum_{n=1}^N \widetilde{\gamma}_{nm}^k \tag{23}$$

**M-step: Update** $p(\mathbf{w}^k)$**.** Note that $p(\mathbf{Z}^k), \boldsymbol{\mu}^k$ have already be updated. Similar to E-step, fixing $p(\mathbf{Z}^k), \boldsymbol{\mu}^k, p(\boldsymbol{\Lambda}^k)$, the objective becomes:

$$\begin{aligned}
&\max_{p(\mathbf{w}^k)} \quad \mathcal{L}(\boldsymbol{\mu}^k, p(\mathbf{Z}^k), p(\mathbf{w}^k), p(\boldsymbol{\Lambda}^k)) \\
&= \max_{p(\mathbf{w}^k)} \quad \int_{\mathbf{Z}^k, \mathbf{w}^k} p(\mathbf{Z}^k) p(\mathbf{w}^k) \ln q(\mathbf{Z}^k | \mathbf{w}^k) d\mathbf{Z}^k d\mathbf{w}^k \\
&\qquad + \int_{\mathbf{w}^k} p(\mathbf{w}^k) \ln q(\mathbf{w}^k) d\mathbf{w}^k - \int_{\mathbf{w}^k} p(\mathbf{w}^k) \ln p(\mathbf{w}^k) d\mathbf{w}^k + \text{Const}
\end{aligned} \tag{24}$$

And the log probability of the optimal $p(\mathbf{w}^k)$ should be:

$$\begin{aligned}
\ln p(\mathbf{w}^k) &= \mathbb{E}_{p(\mathbf{Z}^k)}[\ln q(\mathbf{Z}^k | \mathbf{w}^k)] + \ln q(\mathbf{w}^k) + \text{Const} \\
&= \sum_{m=1}^M (\pi_m + \widetilde{N}_m^k - 1) \ln w_m^k + \text{Const}
\end{aligned} \tag{25}$$

Therefore, the updated $p(\mathbf{w}^k)$ is the following Dirichlet distribution:

$$p(\mathbf{w}^k) = \text{Dirichlet}(\mathbf{w}^k; \widetilde{\boldsymbol{\pi}}^k), \quad \widetilde{\pi}_m^k = \pi_m + \widetilde{N}_m^k \tag{26}$$

**M-step: Update** $p(\boldsymbol{\Lambda}^k)$**.** Note that $p(\mathbf{Z}^k), \boldsymbol{\mu}^k, p(\mathbf{w}^k)$ have already be updated. Fixing $p(\mathbf{Z}^k), \boldsymbol{\mu}^k, p(\mathbf{w}^k)$, the objective becomes:

$$\begin{aligned}
&\max_{p(\boldsymbol{\Lambda}^k)} \quad \mathcal{L}(\boldsymbol{\mu}^k, p(\mathbf{Z}^k), p(\mathbf{w}^k), p(\boldsymbol{\Lambda}^k)) \\
&= \max_{p(\boldsymbol{\Lambda}^k)} \quad \int_{\mathbf{Z}^k, \boldsymbol{\Lambda}^k} p(\mathbf{Z}^k) p(\boldsymbol{\Lambda}^k) \ln q(\mathbf{Y}^k | \mathbf{Z}^k, \boldsymbol{\mu}^k, \boldsymbol{\Lambda}^k) d\mathbf{Z}^k d\boldsymbol{\Lambda}^k \\
&\qquad + \int_{\boldsymbol{\Lambda}^k} p(\boldsymbol{\Lambda}^k) \ln q(\boldsymbol{\Lambda}^k) d\boldsymbol{\Lambda}^k - \int_{\boldsymbol{\Lambda}^k} p(\boldsymbol{\Lambda}^k) \ln p(\boldsymbol{\Lambda}^k) d\boldsymbol{\Lambda}^k + \text{Const}
\end{aligned} \tag{27}$$

The log probability of the optimal $p(\boldsymbol{\Lambda}^k)$ should be:

$$\begin{aligned}
&\ln p(\boldsymbol{\Lambda}^k) \\
&= \mathbb{E}_{p(\mathbf{Z}^k)}[\ln q(\mathbf{Y}^k | \mathbf{Z}^k, \boldsymbol{\mu}^k, \boldsymbol{\Lambda}^k)] + \ln q(\boldsymbol{\Lambda}^k) + \text{Const} \\
&= \sum_{m=1}^M \left\{ \left( u_m^k + \frac{\tau}{2} \widetilde{N}_m^k - 1 \right) \ln \Lambda_m^k - \left( v_m^k + \frac{1}{2} \sum_{n=1}^N \widetilde{\gamma}_{nm}^k \| \mathbf{y}_n^k - \boldsymbol{\mu}_m^k \|_2^2 \right) \Lambda_m^k \right\} + \text{Const}
\end{aligned} \tag{28}$$

which means the updated $p(\boldsymbol{\Lambda}^k)$ is the following Gamma distribution:

$$p(\boldsymbol{\Lambda}^k) = \prod_{m=1}^M \text{Gamma}(\Lambda_m^k; \widetilde{u}_m^k, \widetilde{v}_m^k)$$

$$\widetilde{u}_m^k = u_m^k + \frac{\tau}{2} \widetilde{N}_m^k, \quad \widetilde{v}_m^k = v_m^k + \frac{1}{2} \sum_{n=1}^N \widetilde{\gamma}_{nm}^k \| \mathbf{y}_n^k - \boldsymbol{\mu}_m^k \|_2^2 \tag{29}$$

Eq. 21,23,26,29 are the resulting four steps in Algorithm 1. As each of the four steps raises $\mathcal{L}(\boldsymbol{\mu}^k, p(\mathbf{Z}^k), p(\mathbf{w}^k), p(\boldsymbol{\Lambda}^k))$, the objective gets greater after one iteration. It is also possible to perform the four-step iteration multiple times at each step $k$ to ensure convergence.

# D DETAILED SETUP OF EXPERIMENTS

## D.1 DATASETS

We conduct experiments on eight datasets following Nie et al. (2023); Wu et al. (2023). These datasets include:

- **1)ETTh1** and **2)ETTm1**. These two datasets record 7 key indicators of an electricity transformer, such as load and oil temperature. ETTh1 records data points every hour, and ETTm1 records every 15 minutes. The whole datasets cover a period of two years and we use data from the first 20 months and split it into train/validation/test sets with a ratio of 0.6:0.2:0.2.

- **3)ETTh2** and **4)ETTm2**. The contents, formats and data split of these two datasets are similar to those of ETTh1 and ETTm1, but the records are from another electricity transformer.

- **5)WTH.** This dataset contains 21 weather indicators in Beutenberg, including air temperature and dewpoint, with data points recorded every 10 minutes throughout the year 2020. The train/validation/test sets are split with a ratio of 0.7:0.1:0.2.

- **6)ECL.** This dataset records the hourly electricity consumption (in kW) of 321 clients from 2012 to 2014. The train/validation/test sets are split with a ratio of 0.7:0.1:0.2.

- **7)Traffic** This dataset includes road occupancy rates measured by 862 sensors on freeways in the San Francisco Bay area from July 2016 to June 2018, with data points recorded every hour. The train/validation/test sets are split with a ratio of 0.7:0.1:0.2.

- **8)Dynamic.** This dataset consists of 17 sensors reading and control signals of a simulated complex dynamical system, with data points recorded every second. The original dataset is from `https://www.kaggle.com/datasets/patrickfleith/dynamical-system-multivariate-time-series-forecast` and contains 5,000,000 timestamps. For training time concerns, we use the first 10% data, which corresponds to 500,000 timestamps, and split it into train/validation/test sets with a ratio of 0.7:0.1:0.2. It is worth noting that, despite using only 10% of the original data, the selected timestamps still far exceed those in the previous seven datasets.

The first seven datasets are widely used datasets for TS forecasting (Nie et al., 2023; Wu et al., 2023). In line with standard protocol, we use the last $T = 336$ steps to predict the next $\tau = \{96, 192, 336, 720\}$ steps. To evaluate model performance in more complex scenarios, we introduce a new dataset, Dynamic, which has more complex patterns with no obvious periodicity. For Dynamic, we use the last $T = 600$ steps (10 minutes) to predict the next $\tau = \{60, 120, 180, 300\}$ steps (corresponding to $\{1, 2, 3, 5\}$ minutes). Additionally, due to the large scale of Dynamic, we divide its test set into non-overlapping windows by setting the sliding window step equal to $\tau$. In contrast, for the other datasets, we generate overlapping windows with a sliding step of 1, following the common protocol. Detailed statistical characteristics of the used datasets are shown in Table 6.

Table 6: Statistical characteristics of datasets

| Dataset | #Channels | #Timestamps | Split | $T$ | $\tau$ | Field |
|---|---|---|---|---|---|---|
| ETTh1 | 7 | 14,400 | 0.6:0.2:0.2 | 336 | $\{96, 192, 336, 720\}$ | Electricity Transformer |
| ETTm1 | 7 | 57,600 | 0.6:0.2:0.2 | 336 | $\{96, 192, 336, 720\}$ | Electricity Transformer |
| ETTh2 | 7 | 14,400 | 0.6:0.2:0.2 | 336 | $\{96, 192, 336, 720\}$ | Electricity Transformer |
| ETTm2 | 7 | 57,600 | 0.6:0.2:0.2 | 336 | $\{96, 192, 336, 720\}$ | Electricity Transformer |
| WTH | 21 | 52,696 | 0.7:0.1:0.2 | 336 | $\{96, 192, 336, 720\}$ | Meteorological Indicators |
| ECL | 321 | 26,304 | 0.7:0.1:0.2 | 336 | $\{96, 192, 336, 720\}$ | Electricity Consumption |
| Traffic | 862 | 17,544 | 0.7:0.1:0.2 | 336 | $\{96, 192, 336, 720\}$ | Road Occupancy |
| Dynamic | 17 | 500,000 | 0.7:0.1:0.2 | 600 | $\{60, 120, 180, 300\}$ | Complex Dynamical System |

## D.2 METRICS

**Top-K MSE.** The ground truth target is $\mathbf{y} \in \mathbb{R}^\tau$. Multi-mode predictions are $\{\widetilde{\mathbf{y}}_m\}_{m=1}^M, \mathbf{y}_m \in \mathbb{R}^\tau$ and corresponding probabilities are $\{w_m\}_{m=1}^M, \sum_{m=1}^M w_m = 1$. Top-K MSE is computed as:

$$\mathcal{M} = \textbf{Top-K}(\{w_m\}_{m=1}^M)$$
$$\forall m \in \mathcal{M}, \text{MSE}_m = \frac{1}{\tau}\|\widetilde{\mathbf{y}}_m - \mathbf{y}\|_2^2 \tag{30}$$
$$\text{Top-K MSE} = \min_{m \in \mathcal{M}} \text{MSE}_m$$

We first pick the Top-K modes with the highest probabilities into $\mathcal{M}$. Then MSE of each mode in $\mathcal{M}$ is computed. Finally, the minimum MSE among the selected K mode is reported as Top-K MSE.

**Top-K MAE.** The computation of Top-K MAE is similar to Top-K MSE, with MSE changing to MAE:

$$\mathcal{M} = \textbf{Top-K}(\{w_m\}_{m=1}^M)$$
$$\forall m \in \mathcal{M}, \text{MAE}_m = \frac{1}{\tau}\|\widetilde{\mathbf{y}}_m - \mathbf{y}\|_1 \tag{31}$$
$$\text{Top-K MAE} = \min_{m \in \mathcal{M}} \text{MAE}_m$$

**MSE.** Given ground truth target $\mathbf{y} \in \mathbb{R}^\tau$ and deterministic prediction $\widetilde{\mathbf{y}} \in \mathbb{R}^\tau$, MSE is computed as:

$$\text{MSE} = \frac{1}{\tau}\|\widetilde{\mathbf{y}} - \mathbf{y}\|_2^2 \tag{32}$$

**CRPS.** CRPS is a frequently used metric for probabilistic prediction and it is originally defined for the scaler variable. Given the ground truth target $y \in \mathbb{R}$ and the cumulative distribution function (CDF) of the predicted distribution $\widetilde{F}(y)$, the CRPS is defined as:

$$\text{CRPS}(\widetilde{F}, y) = \int_{-\infty}^{+\infty} \left(\widetilde{F}(\widetilde{y}) - \mathbb{1}(\widetilde{y} \geq y)\right)^2 d\widetilde{y} \tag{33}$$

Gneiting & Raftery (2007) show that CRPS can also be computed by:

$$\text{CRPS}(\widetilde{F}, y) = \mathbb{E}_{\widetilde{y}}[|\widetilde{y} - y|] - \frac{1}{2}\mathbb{E}_{\widetilde{y},\widetilde{y}^*}[|\widetilde{y} - \widetilde{y}^*|] \tag{34}$$

where $\widetilde{y}, \widetilde{y}^*$ are random variables following the predicted distribution, i.e., distribution corresponding to $\widetilde{F}(y)$.

Following this formulation, given the ground truth target $\mathbf{y} \in \mathbb{R}^\tau$ and samples draw from the probabilistic model $\{\widetilde{\mathbf{y}}_i\}_{i=1}^N, \mathbf{y}_i \in \mathbb{R}^\tau$, we compute the CRPS by:

$$\text{CRPS}_t \approx \frac{1}{N}\sum_{i=1}^N |\widetilde{y}_{i,t} - y_t| - \frac{1}{2N^2}\sum_{i=1}^N\sum_{j=1}^N |\widetilde{y}_{i,t} - \widetilde{y}_{j,t}|$$
$$\text{CRPS} = \frac{1}{\tau}\sum_{t=1}^\tau \text{CRPS}_t \tag{35}$$

where $\widetilde{y}_{i,t} \in \mathbb{R}$ is the predicted value of timestamp $t$ in sample $\widetilde{\mathbf{y}}_i$. $y_t \in \mathbb{R}$ is the target at timestamp $t$. We approximate the CRPS at each step and use the average across steps as the CRPS of the whole series.

**Remark.** CRPS also has the following connection with quantile loss:

$$\text{CRPS}(\widetilde{F}, y) = 2\int_0^1 \rho_\alpha(y - \widetilde{F}^{-1}(\alpha))d\alpha$$
$$\rho_\alpha(u) = u(\alpha - \mathbb{1}(u < 0)) \tag{36}$$

As a result, some works (Woo et al., 2024) approximate $\text{CRPS}_t$ by:

$$\text{CRPS}_t \approx \frac{2}{K} \sum_{k=1}^{K} \rho_{\alpha_k} \left( y_t - Q(\{\widetilde{y}_{i,t}\}_{i=1}^{N}, \alpha_k) \right) \tag{37}$$

where $Q(\{\widetilde{y}_{i,t}\}_{i=1}^{N}, \alpha_k)$ computes the $\alpha_k$-quantile at step $t$. $\{\alpha_k\}_{k=1}^{K}$ are some pre-defined quantiles for integral approximation. We do not use this approximation because it will be affected by the selection of $\{\alpha_k\}_{k=1}^{K}$ and the quantile function $Q(\cdot)$.

We have described the metrics for a univariate instance. Since all the datasets we used are multivariate, we compute the metrics for each channel and then average them across channels to obtain the metrics for a multivariate instance. Finally, the metrics for a dataset are calculated by averaging the metrics across all instances in the test set.

### D.3 IMPLEMENTATIONS

The default patch size is set to $P = 12$. For the sake of saving runtime memory, we use $P = 24$ at $\tau = \{336, 720\}$ or when the dataset is ECL or Traffic. As for the Patch Consistent MLP, we use a one-block MLP with the dimension of hidden states $d_{model} = 256$, adjacent range hyper-parameter $r = 3$ on all datasets.

Regarding the training process, we use a linear noise schedule with $K_{train} = 1,000$ diffusion steps (Ho et al., 2020). The default diffusion-deterministic balancing weight, i.e., $\lambda$ in Eq. 8, is set to 0.99. When using SegRNN as the backbone, it is set to 0.9. Adam optimizer with a learning rate of 1e-4 is used for optimization. The maximum number of training epochs is set to 20, and if the validation loss does not decrease over 5 consecutive validations, the training process is terminated early. The commonly used instance normalization (Kim et al., 2021) is also applied to reduce the distribution shift.

As for inference, for each input, we generate $N = 100$ samples and set the maximum number of modes to $M = 10$. In our multi-mode inference algorithm, we resample the inference-time diffusion steps to $K_{infer} = 20$ and perform 10 EM iterations at each step. The hyper-parameters are set to $\rho = 0.5, u = 100$. For other baselines, we draw $N$ samples from their corresponding distributions and post-process them with a GMM in the same formulation as in MMPD to obtain multi-mode predictions.

All models are implemented in Pytorch and run on NVIDIA GeForce RTX 3090 GPUs with 24GB memory.

## E FULL RESULTS

Due to space limitations in the main text, we present the full results from Sec. 4.1 here. Table 7 shows the Top-3 MSE, Top-3 MAE, MSE and CRPS evaluations of different loss functions across various forecasting horizons $\tau$. Table 8 displays the Top-3 MSE and Top-3 MAE evaluations for MSE, Mix, and MMPD across three different backbones at different forecasting horizons. Finally, Table 9 presents the MSE and CRPS evaluations across three different backbones at different forecasting horizons.

Table 7: Full Top-3 MSE, Top-3 MAE, MSE and CRPS evaluations of different loss functions on different forecasting horizons $\tau$, which is set to {96, 192, 336, 720} for the first seven datasets and {60, 120, 180, 300} for Dynamic. "Gauss" is short for "Gaussian". "T" is short for "Student-T". Bold/underline indicates the best/second. Our method is marked in gray. Inf indicates the infinity problem caused by outliers.

| Metric | | Top-3 MSE | | | | | | Top-3 MAE | | | | | | MSE | | | | | | CRPS | | | | | |
|---|---|---|---|---|---|---|---|---|---|---|---|---|---|---|---|---|---|---|---|---|---|---|---|---|---|
| Loss | | MSE | MAE | Gauss | T | Mix | MMPD | MSE | MAE | Gauss | T | Mix | MMPD | MSE | MAE | Gauss | T | Mix | MMPD | MSE | MAE | Gauss | T | Mix | MMPD |
| ETTh1 | 96 | 0.365 | 0.390 | 0.381 | 0.375 | _0.371_ | **0.329** | 0.399 | 0.395 | 0.400 | 0.391 | _0.382_ | **0.371** | **0.374** | 0.385 | 0.383 | 0.377 | 0.399 | _0.375_ | 0.316 | 0.325 | 0.291 | **0.283** | _0.286_ | 0.289 |
| | 192 | 0.412 | 0.425 | 0.429 | 0.415 | _0.411_ | **0.396** | 0.426 | 0.422 | 0.427 | 0.415 | _0.419_ | **0.405** | _0.411_ | 0.415 | 0.426 | 0.413 | 0.426 | **0.406** | 0.331 | 0.338 | _0.311_ | **0.300** | 0.311 | 0.314 |
| | 336 | 0.446 | 0.450 | 0.447 | 0.458 | **0.410** | _0.415_ | 0.449 | 0.444 | 0.444 | 0.442 | **0.419** | _0.422_ | **0.436** | 0.439 | 0.440 | 0.454 | 0.449 | _0.422_ | 0.342 | 0.349 | _0.320_ | 0.322 | **0.314** | 0.326 |
| | 720 | 0.496 | 0.501 | 0.501 | _0.470_ | 0.510 | **0.445** | 0.488 | 0.486 | 0.476 | _0.463_ | 0.486 | **0.453** | _0.481_ | 0.488 | 0.488 | 0.462 | 0.511 | **0.444** | 0.361 | 0.371 | 0.345 | **0.333** | 0.353 | _0.340_ |
| | Avg | 0.430 | 0.441 | 0.439 | 0.430 | _0.425_ | **0.396** | 0.440 | 0.437 | 0.437 | 0.428 | _0.426_ | **0.412** | **0.425** | 0.432 | 0.434 | 0.426 | 0.446 | _0.412_ | 0.337 | 0.346 | 0.317 | _0.310_ | 0.316 | 0.318 |
| ETTm1 | 96 | 0.265 | 0.309 | 0.291 | 0.284 | _0.188_ | **0.180** | 0.333 | 0.336 | 0.344 | 0.327 | _0.277_ | **0.272** | _0.290_ | 0.307 | 0.297 | 0.287 | 0.303 | **0.280** | 0.281 | 0.291 | 0.253 | 0.242 | _0.239_ | **0.239** |
| | 192 | 0.317 | 0.344 | 0.340 | 0.333 | _0.253_ | **0.240** | 0.364 | 0.363 | 0.375 | 0.358 | _0.321_ | **0.314** | _0.323_ | 0.334 | 0.338 | 0.331 | 0.337 | **0.315** | 0.296 | 0.304 | 0.274 | 0.261 | **0.258** | _0.259_ |
| | 336 | 0.368 | 0.371 | 0.370 | 0.364 | _0.320_ | **0.291** | 0.393 | 0.385 | 0.393 | 0.381 | _0.354_ | **0.348** | _0.360_ | 0.359 | 0.364 | _0.359_ | 0.371 | **0.352** | 0.311 | 0.316 | 0.284 | _0.276_ | **0.276** | 0.280 |
| | 720 | 0.441 | 0.433 | 0.444 | 0.426 | _0.397_ | **0.367** | 0.435 | 0.419 | 0.432 | 0.418 | _0.398_ | **0.392** | 0.427 | 0.420 | 0.430 | _0.419_ | 0.423 | **0.401** | 0.338 | 0.339 | 0.317 | _0.304_ | **0.304** | 0.303 |
| | Avg | 0.348 | 0.364 | 0.361 | 0.352 | _0.289_ | **0.269** | 0.381 | 0.375 | 0.386 | 0.371 | _0.338_ | **0.331** | 0.350 | 0.355 | 0.357 | _0.349_ | 0.358 | **0.337** | 0.307 | 0.313 | 0.282 | 0.271 | **0.269** | _0.270_ |
| ETTh2 | 96 | 0.273 | 0.293 | 0.285 | 0.305 | _0.260_ | **0.241** | 0.334 | 0.336 | 0.347 | 0.343 | _0.328_ | **0.310** | _0.294_ | 0.296 | 0.305 | 0.304 | 0.322 | **0.293** | 0.264 | _0.263_ | 0.271 | **0.261** | 0.271 | 0.271 |
| | 192 | 0.345 | 0.360 | 0.363 | 0.364 | _0.320_ | **0.283** | 0.382 | 0.382 | 0.395 | 0.385 | _0.371_ | **0.344** | 0.367 | _0.358_ | 0.377 | 0.361 | 0.401 | **0.357** | 0.299 | **0.291** | 0.305 | _0.292_ | 0.309 | 0.294 |
| | 336 | 0.391 | 0.408 | 0.419 | 0.403 | _0.363_ | **0.310** | 0.414 | 0.416 | 0.433 | 0.416 | _0.393_ | **0.368** | 0.399 | 0.404 | 0.419 | 0.400 | 0.402 | **0.366** | 0.318 | _0.315_ | 0.329 | 0.316 | 0.318 | **0.312** |
| | 720 | 0.448 | _0.411_ | 0.450 | 0.430 | 0.430 | **0.360** | 0.463 | _0.433_ | 0.469 | 0.448 | 0.454 | **0.407** | 0.444 | _0.408_ | 0.430 | 0.423 | 0.437 | **0.400** | 0.350 | **0.326** | 0.355 | 0.332 | 0.343 | _0.329_ |
| | Avg | 0.364 | 0.368 | 0.379 | 0.375 | _0.343_ | **0.299** | 0.398 | 0.392 | 0.411 | 0.398 | _0.387_ | **0.357** | 0.376 | _0.366_ | 0.382 | 0.372 | 0.390 | **0.354** | 0.308 | _0.299_ | 0.315 | 0.300 | 0.310 | 0.301 |
| ETTm2 | 96 | _0.156_ | 0.174 | 0.176 | 0.180 | _0.156_ | **0.135** | 0.244 | 0.253 | 0.270 | 0.258 | _0.242_ | **0.224** | 0.169 | 0.174 | 0.194 | 0.180 | 0.179 | _0.173_ | 0.196 | 0.200 | 0.211 | 0.194 | _0.193_ | **0.192** |
| | 192 | 0.223 | 0.241 | 0.233 | 0.248 | _0.203_ | **0.185** | 0.293 | 0.298 | 0.307 | 0.305 | _0.279_ | **0.266** | _0.234_ | 0.238 | 0.243 | 0.246 | 0.244 | **0.227** | 0.229 | 0.230 | 0.235 | 0.230 | **0.226** | 0.225 |
| | 336 | 0.293 | 0.301 | 0.308 | 0.322 | _0.259_ | **0.225** | 0.342 | 0.338 | 0.356 | 0.364 | _0.319_ | **0.294** | _0.298_ | 0.298 | 0.306 | 0.317 | 0.318 | **0.289** | 0.261 | _0.258_ | 0.269 | 0.270 | 0.262 | **0.255** |
| | 720 | 0.383 | 0.387 | 0.403 | 0.395 | _0.362_ | **0.313** | 0.399 | 0.397 | 0.417 | 0.407 | _0.393_ | **0.357** | _0.379_ | 0.384 | 0.393 | 0.390 | 0.399 | **0.367** | 0.304 | _0.301_ | 0.310 | 0.305 | 0.309 | **0.298** |
| | Avg | 0.264 | 0.276 | 0.280 | 0.286 | _0.245_ | **0.214** | 0.320 | 0.321 | 0.337 | 0.333 | _0.308_ | **0.285** | **0.270** | 0.274 | 0.284 | 0.283 | 0.285 | _0.264_ | _0.247_ | 0.247 | 0.256 | 0.250 | 0.247 | **0.243** |
| WTH | 96 | _0.142_ | 0.153 | 0.156 | 0.157 | 0.144 | **0.121** | 0.192 | 0.193 | 0.207 | 0.193 | _0.183_ | **0.155** | 0.149 | _0.153_ | 0.166 | 0.160 | 0.167 | _0.153_ | 0.168 | 0.171 | 0.172 | **0.148** | 0.152 | _0.149_ |
| | 192 | 0.185 | 0.199 | 0.214 | 0.211 | _0.182_ | **0.157** | 0.234 | 0.238 | 0.263 | 0.243 | _0.220_ | **0.193** | **0.193** | 0.196 | 0.218 | 0.210 | 0.212 | **0.193** | 0.198 | 0.200 | 0.205 | 0.186 | _0.184_ | **0.177** |
| | 336 | 0.244 | 0.256 | 0.273 | 0.261 | _0.222_ | **0.206** | 0.284 | 0.283 | 0.308 | 0.281 | _0.256_ | **0.236** | **0.244** | 0.252 | 0.270 | 0.259 | 0.276 | _0.248_ | 0.231 | 0.233 | 0.240 | _0.214_ | 0.222 | **0.212** |
| | 720 | 0.325 | 0.333 | 0.375 | 0.336 | _0.288_ | **0.290** | 0.340 | 0.337 | 0.388 | 0.335 | _0.320_ | **0.301** | **0.321** | 0.330 | 0.367 | 0.333 | 0.380 | _0.323_ | 0.276 | 0.276 | 0.295 | _0.257_ | 0.277 | **0.256** |
| | Avg | 0.224 | 0.235 | 0.255 | 0.241 | _0.209_ | **0.193** | 0.262 | 0.263 | 0.292 | 0.263 | _0.242_ | **0.221** | **0.227** | 0.233 | 0.255 | 0.241 | 0.259 | _0.229_ | 0.218 | 0.220 | 0.228 | _0.201_ | 0.209 | **0.199** |
| ECL | 96 | 0.141 | 0.144 | 0.130 | 0.133 | _0.119_ | **0.119** | 0.247 | 0.242 | 0.224 | _0.220_ | 0.220 | **0.212** | **0.130** | 0.132 | _0.131_ | 0.133 | 0.136 | 0.133 | 0.258 | 0.275 | 0.167 | **0.165** | _0.165_ | 0.169 |
| | 192 | 0.162 | 0.165 | 0.149 | 0.152 | _0.136_ | **0.134** | 0.266 | 0.261 | 0.241 | 0.237 | _0.226_ | **0.225** | **0.147** | 0.151 | _0.148_ | 0.151 | 0.153 | 0.151 | 0.265 | 0.282 | 0.179 | _0.177_ | **0.176** | 0.181 |
| | 336 | 0.181 | 0.183 | 0.172 | 0.169 | _0.151_ | **0.147** | 0.284 | 0.278 | 0.266 | 0.255 | _0.242_ | **0.239** | **0.164** | 0.168 | 0.169 | _0.167_ | 0.170 | 0.167 | 0.272 | 0.289 | 0.197 | **0.190** | 0.640 | _0.195_ |
| | 720 | 0.219 | 0.222 | 0.207 | 0.207 | _0.182_ | **0.186** | 0.315 | 0.309 | 0.293 | 0.287 | **0.271** | 0.274 | **0.200** | 0.206 | _0.202_ | 0.205 | 0.210 | 0.205 | 0.287 | 0.304 | 0.217 | **0.214** | Inf | 0.221 |
| | Avg | 0.176 | 0.179 | 0.165 | 0.165 | _0.147_ | **0.147** | 0.278 | 0.272 | 0.256 | 0.250 | _0.240_ | **0.238** | **0.160** | 0.164 | _0.163_ | 0.164 | 0.167 | 0.164 | 0.270 | 0.288 | _0.190_ | **0.187** | Inf | 0.191 |
| Traffic | 96 | 0.382 | 0.408 | 0.382 | 0.380 | _0.373_ | **0.350** | 0.295 | 0.283 | 0.266 | 0.246 | _0.245_ | **0.237** | **0.368** | 0.389 | 0.388 | 0.392 | 0.398 | _0.383_ | 0.335 | 0.354 | 0.206 | _0.193_ | 0.194 | **0.190** |
| | 192 | 0.412 | 0.434 | 0.405 | 0.404 | _0.399_ | **0.374** | 0.309 | 0.297 | 0.274 | _0.254_ | 0.254 | **0.246** | **0.387** | 0.410 | 0.403 | 0.408 | 0.415 | _0.398_ | 0.339 | 0.359 | 0.212 | _0.199_ | 0.199 | **0.196** |
| | 336 | 0.441 | 0.446 | 0.421 | 0.419 | _0.414_ | **0.391** | 0.326 | 0.304 | 0.280 | _0.261_ | 0.261 | **0.253** | **0.411** | 0.420 | 0.414 | 0.419 | 0.426 | **0.411** | 0.345 | 0.361 | 0.216 | _0.204_ | 0.204 | **0.201** |
| | 720 | 0.495 | 0.508 | 0.468 | 0.462 | _0.460_ | **0.440** | 0.372 | 0.357 | 0.307 | _0.283_ | 0.284 | **0.280** | **0.432** | 0.450 | 0.447 | 0.453 | 0.465 | _0.444_ | 0.355 | 0.376 | 0.234 | _0.221_ | 0.222 | **0.219** |
| | Avg | 0.433 | 0.449 | 0.419 | 0.416 | _0.412_ | **0.389** | 0.326 | 0.310 | 0.282 | _0.261_ | 0.261 | **0.254** | **0.399** | 0.417 | 0.413 | 0.418 | 0.426 | _0.409_ | 0.343 | 0.362 | 0.217 | _0.204_ | 0.205 | **0.202** |
| Dynamic | 60 | 0.229 | 0.285 | 0.234 | 0.277 | _0.216_ | **0.179** | 0.214 | 0.191 | 0.205 | 0.183 | _0.157_ | **0.109** | **0.240** | 0.300 | 0.246 | 0.285 | 0.348 | _0.250_ | 0.203 | 0.220 | 0.165 | _0.152_ | 0.154 | **0.133** |
| | 120 | _0.302_ | 0.393 | 0.310 | 0.363 | 0.303 | **0.275** | 0.285 | 0.271 | 0.284 | 0.264 | _0.227_ | **0.184** | **0.314** | 0.393 | _0.320_ | 0.367 | 0.445 | 0.326 | 0.241 | 0.259 | 0.217 | 0.208 | _0.207_ | **0.187** |
| | 180 | 0.360 | 0.465 | 0.371 | 0.421 | _0.350_ | **0.338** | 0.335 | 0.323 | 0.337 | 0.325 | _0.270_ | **0.236** | **0.371** | 0.460 | 0.377 | 0.421 | 0.521 | 0.382 | 0.270 | 0.289 | 0.252 | 0.245 | _0.243_ | **0.223** |
| | 300 | 0.452 | 0.561 | 0.455 | 0.500 | _0.419_ | **0.415** | 0.409 | 0.394 | 0.409 | 0.398 | _0.330_ | **0.301** | **0.454** | 0.554 | **0.452** | 0.495 | 0.615 | 0.456 | 0.315 | 0.332 | 0.296 | _0.290_ | 0.290 | **0.270** |
| | Avg | 0.336 | 0.426 | 0.343 | 0.390 | _0.322_ | **0.301** | 0.311 | 0.295 | 0.309 | 0.292 | _0.246_ | **0.207** | **0.345** | 0.426 | _0.349_ | 0.392 | 0.482 | 0.353 | 0.257 | 0.275 | 0.233 | _0.224_ | 0.224 | **0.203** |

Table 8: Full Top-3 MSE and Top-3 MAE evaluations of MSE, Mix and MMPD losses across three different backbones on different forecasting horizons $\tau$, which is set to {96, 192, 336, 720} for the first seven datasets and {60, 120, 180, 300} for Dynamic. Bold indicates the best among three losses. Our method is marked in gray.

| Metric | | Top-3 MSE | | | | | | | | | Top-3 MAE | | | | | | | | |
|---|---|---|---|---|---|---|---|---|---|---|---|---|---|---|---|---|---|---|---|
| Backbone | | Crossformer | | | SegRNN | | | MaskAE | | | Crossformer | | | SegRNN | | | MaskAE | | |
| Loss | | MSE | Mix | MMPD | MSE | Mix | MMPD | MSE | Mix | MMPD | MSE | Mix | MMPD | MSE | Mix | MMPD | MSE | Mix | MMPD |
| ETTh1 | 96 | 0.365 | **0.329** | 0.336 | 0.375 | 0.354 | **0.334** | 0.374 | 0.351 | **0.341** | 0.406 | **0.372** | 0.377 | 0.413 | 0.389 | **0.385** | 0.407 | 0.390 | **0.376** |
| | 192 | 0.417 | 0.404 | **0.369** | 0.421 | 0.466 | **0.383** | 0.416 | 0.393 | **0.386** | 0.437 | 0.413 | **0.398** | 0.441 | 0.448 | **0.415** | 0.431 | 0.412 | **0.405** |
| | 336 | 0.462 | 0.466 | **0.399** | 0.464 | 0.423 | **0.408** | 0.461 | 0.426 | **0.413** | 0.464 | 0.454 | **0.420** | 0.459 | **0.428** | **0.428** | 0.458 | **0.425** | 0.426 |
| | 720 | 0.528 | 0.533 | **0.418** | 0.500 | 0.497 | **0.482** | 0.503 | 0.491 | **0.457** | 0.499 | 0.492 | **0.447** | 0.490 | **0.464** | 0.475 | 0.492 | 0.471 | **0.455** |
| | Avg | 0.443 | 0.433 | **0.381** | 0.440 | 0.435 | **0.402** | 0.438 | 0.415 | **0.399** | 0.452 | 0.433 | **0.410** | 0.451 | 0.432 | **0.426** | 0.447 | 0.425 | **0.416** |
| ETTm1 | 96 | 0.274 | **0.202** | 0.248 | 0.311 | 0.254 | **0.250** | 0.266 | 0.234 | **0.207** | 0.340 | **0.293** | 0.307 | 0.375 | 0.332 | **0.331** | 0.334 | 0.314 | **0.299** |
| | 192 | 0.347 | 0.284 | **0.269** | 0.364 | 0.313 | **0.301** | 0.326 | 0.278 | **0.240** | 0.384 | 0.335 | **0.333** | 0.406 | 0.366 | **0.365** | 0.374 | 0.334 | **0.314** |
| | 336 | 0.399 | 0.337 | **0.328** | 0.410 | 0.347 | **0.329** | 0.381 | 0.329 | **0.297** | 0.410 | 0.366 | **0.365** | 0.432 | 0.391 | **0.379** | 0.405 | 0.359 | **0.354** |
| | 720 | 0.492 | 0.497 | **0.395** | 0.454 | 0.428 | **0.404** | 0.442 | 0.417 | **0.375** | 0.455 | 0.437 | **0.407** | 0.455 | 0.424 | **0.424** | 0.443 | 0.406 | **0.401** |
| | Avg | 0.378 | 0.330 | **0.310** | 0.385 | 0.335 | **0.321** | 0.354 | 0.314 | **0.280** | 0.397 | 0.358 | **0.353** | 0.417 | 0.378 | **0.375** | 0.389 | 0.353 | **0.342** |
| ETTh2 | 96 | 0.301 | 0.294 | **0.250** | 0.288 | 0.275 | **0.273** | 0.270 | 0.279 | **0.236** | 0.359 | 0.351 | **0.326** | 0.352 | 0.341 | **0.338** | 0.331 | 0.342 | **0.310** |
| | 192 | 0.368 | 0.353 | **0.304** | 0.359 | 0.356 | **0.321** | 0.341 | 0.333 | **0.318** | 0.402 | 0.394 | **0.363** | 0.401 | 0.399 | **0.379** | 0.378 | 0.377 | **0.371** |
| | 336 | 0.386 | 0.362 | **0.327** | 0.383 | 0.355 | **0.314** | 0.387 | 0.358 | **0.322** | 0.420 | 0.401 | **0.377** | 0.419 | 0.397 | **0.373** | 0.414 | 0.391 | **0.374** |
| | 720 | 0.433 | 0.427 | **0.377** | 0.429 | 0.380 | **0.376** | 0.423 | 0.390 | **0.368** | 0.468 | 0.452 | **0.422** | 0.457 | 0.419 | **0.417** | 0.448 | 0.426 | **0.414** |
| | Avg | 0.372 | 0.359 | **0.315** | 0.365 | 0.341 | **0.321** | 0.355 | 0.340 | **0.311** | 0.412 | 0.400 | **0.372** | 0.408 | 0.389 | **0.377** | 0.392 | 0.384 | **0.367** |
| ETTm2 | 96 | 0.158 | 0.150 | **0.141** | 0.173 | 0.161 | **0.142** | 0.171 | 0.191 | **0.148** | 0.250 | 0.247 | **0.231** | 0.267 | 0.252 | **0.239** | 0.263 | 0.250 | **0.233** |
| | 192 | 0.220 | 0.200 | **0.196** | 0.236 | **0.208** | 0.215 | 0.235 | **0.214** | 0.223 | 0.291 | 0.281 | **0.277** | 0.314 | 0.288 | **0.288** | 0.308 | 0.285 | **0.285** |
| | 336 | 0.295 | 0.255 | **0.224** | 0.297 | 0.267 | **0.235** | 0.306 | 0.264 | **0.261** | 0.346 | 0.321 | **0.295** | 0.354 | 0.326 | **0.309** | 0.358 | 0.321 | **0.311** |
| | 720 | 0.391 | **0.338** | 0.350 | 0.387 | 0.355 | **0.340** | 0.409 | **0.342** | 0.355 | 0.406 | **0.378** | 0.379 | 0.413 | **0.378** | 0.384 | 0.420 | **0.377** | 0.386 |
| | Avg | 0.266 | 0.236 | **0.228** | 0.273 | 0.248 | **0.233** | 0.280 | 0.253 | **0.247** | 0.323 | 0.307 | **0.295** | 0.337 | 0.311 | **0.305** | 0.337 | 0.308 | **0.304** |
| weather | 96 | 0.134 | 0.130 | **0.120** | 0.140 | 0.140 | **0.125** | 0.140 | 0.132 | **0.118** | 0.189 | 0.174 | **0.157** | 0.198 | 0.179 | **0.165** | 0.190 | 0.171 | **0.153** |
| | 192 | 0.186 | 0.168 | **0.164** | 0.182 | 0.189 | **0.165** | 0.185 | 0.164 | **0.162** | 0.241 | 0.213 | **0.202** | 0.239 | 0.234 | **0.210** | 0.236 | 0.207 | **0.197** |
| | 336 | 0.247 | 0.217 | **0.210** | 0.240 | 0.233 | **0.217** | 0.246 | 0.229 | **0.210** | 0.296 | 0.252 | **0.241** | 0.288 | 0.266 | **0.252** | 0.284 | 0.264 | **0.241** |
| | 720 | 0.327 | **0.284** | 0.297 | 0.327 | **0.295** | 0.296 | 0.332 | **0.287** | 0.300 | 0.347 | **0.300** | 0.304 | 0.352 | **0.304** | 0.309 | 0.344 | **0.309** | 0.309 |
| | Avg | 0.223 | 0.200 | **0.197** | 0.222 | 0.214 | **0.201** | 0.226 | 0.203 | **0.197** | 0.268 | 0.235 | **0.226** | 0.269 | 0.246 | **0.234** | 0.263 | 0.238 | **0.225** |
| ECL | 96 | 0.146 | 0.130 | **0.120** | 0.145 | 0.130 | **0.122** | 0.142 | 0.122 | **0.118** | 0.253 | 0.223 | **0.214** | 0.252 | 0.218 | **0.217** | 0.247 | 0.213 | **0.209** |
| | 192 | 0.169 | 0.148 | **0.139** | 0.168 | 0.143 | **0.136** | 0.164 | 0.139 | **0.132** | 0.275 | 0.242 | **0.232** | 0.275 | 0.231 | **0.229** | 0.267 | 0.230 | **0.221** |
| | 336 | 0.192 | 0.169 | **0.164** | 0.188 | 0.160 | **0.151** | 0.184 | 0.156 | **0.145** | 0.295 | 0.266 | **0.257** | 0.292 | 0.251 | **0.244** | 0.286 | 0.245 | **0.235** |
| | 720 | 0.229 | 0.193 | **0.186** | 0.230 | 0.190 | **0.190** | 0.224 | 0.186 | **0.182** | 0.327 | 0.288 | **0.279** | 0.326 | 0.280 | **0.278** | 0.319 | 0.275 | **0.269** |
| | Avg | 0.184 | 0.160 | **0.152** | 0.183 | 0.155 | **0.150** | 0.178 | 0.150 | **0.144** | 0.288 | 0.255 | **0.245** | 0.286 | 0.245 | **0.242** | 0.280 | 0.241 | **0.234** |
| Traffic | 96 | 0.397 | 0.375 | **0.367** | 0.409 | 0.388 | **0.365** | 0.384 | 0.372 | **0.345** | 0.310 | 0.259 | **0.248** | 0.305 | **0.232** | 0.242 | 0.299 | 0.245 | **0.235** |
| | 192 | 0.436 | 0.409 | **0.389** | 0.440 | 0.419 | **0.400** | 0.418 | 0.400 | **0.373** | 0.326 | 0.272 | **0.253** | 0.318 | **0.245** | 0.251 | 0.316 | 0.257 | **0.244** |
| | 336 | 0.453 | 0.431 | **0.405** | 0.463 | 0.432 | **0.428** | 0.432 | 0.418 | **0.392** | 0.341 | 0.280 | **0.260** | 0.328 | **0.253** | 0.263 | 0.321 | 0.265 | **0.253** |
| | 720 | 0.518 | 0.483 | **0.456** | 0.543 | 0.516 | **0.481** | 0.509 | 0.476 | **0.440** | 0.391 | 0.304 | **0.283** | 0.380 | **0.280** | 0.285 | 0.381 | 0.291 | **0.278** |
| | Avg | 0.451 | 0.424 | **0.404** | 0.464 | 0.439 | **0.418** | 0.436 | 0.416 | **0.387** | 0.342 | 0.279 | **0.261** | 0.333 | 0.253 | **0.260** | 0.329 | 0.265 | **0.253** |
| Dynamic | 60 | 0.227 | 0.194 | **0.162** | 0.228 | 0.248 | **0.176** | 0.230 | 0.205 | **0.172** | 0.207 | 0.131 | **0.089** | 0.212 | 0.182 | **0.108** | 0.213 | 0.149 | **0.105** |
| | 120 | 0.300 | 0.297 | **0.258** | 0.304 | 0.332 | **0.266** | 0.304 | 0.301 | **0.263** | 0.281 | 0.218 | **0.163** | 0.285 | 0.252 | **0.185** | 0.286 | 0.224 | **0.177** |
| | 180 | 0.357 | 0.335 | **0.332** | 0.356 | 0.344 | **0.328** | 0.363 | 0.342 | **0.334** | 0.329 | 0.263 | **0.224** | 0.329 | 0.258 | **0.239** | 0.338 | 0.267 | **0.229** |
| | 300 | 0.440 | **0.401** | 0.429 | 0.445 | **0.395** | 0.408 | 0.458 | 0.437 | **0.414** | 0.399 | 0.319 | **0.300** | 0.403 | 0.299 | **0.308** | 0.410 | 0.338 | **0.299** |
| | Avg | 0.331 | 0.307 | **0.295** | 0.333 | 0.330 | **0.295** | 0.339 | 0.321 | **0.296** | 0.304 | 0.233 | **0.194** | 0.307 | 0.247 | **0.210** | 0.312 | 0.244 | **0.203** |

Table 9: Full MSE and CRPS evaluations of MSE, Mix and MMPD losses across three different backbones on different forecasting horizons $\tau$, which is set to {96, 192, 336, 720} for the first seven datasets and {60, 120, 180, 300} for Dynamic. Bold indicates the best among three losses. Our method is marked in gray. Inf indicates the infinity problem caused by outliers.

| Metric | | MSE | | | | | | | | | CRPS | | | | | | | | |
|---|---|---|---|---|---|---|---|---|---|---|---|---|---|---|---|---|---|---|---|
| Backbone | | Crossformer | | | SegRNN | | | MaskAE | | | Crossformer | | | SegRNN | | | MaskAE | | |
| Loss | | MSE | Mix | MMPD | MSE | Mix | MMPD | MSE | Mix | MMPD | MSE | Mix | MMPD | MSE | Mix | MMPD | MSE | Mix | MMPD |
| ETTh1 | 96 | **0.373** | 0.418 | 0.381 | 0.377 | 0.396 | **0.374** | 0.386 | 0.397 | **0.383** | 0.320 | 0.294 | **0.293** | 0.320 | 0.294 | 0.296 | 0.320 | **0.293** | 0.294 |
| | 192 | 0.419 | 0.459 | **0.403** | 0.419 | 0.480 | 0.424 | 0.422 | 0.453 | **0.414** | 0.337 | 0.315 | **0.305** | 0.336 | 0.330 | **0.316** | 0.334 | 0.316 | **0.312** |
| | 336 | 0.455 | 0.468 | **0.422** | 0.454 | 0.454 | 0.460 | 0.454 | 0.464 | **0.428** | 0.349 | 0.332 | **0.324** | 0.346 | Inf | **0.331** | 0.347 | **0.320** | 0.328 |
| | 720 | 0.512 | 0.495 | **0.456** | 0.484 | 0.480 | **0.479** | 0.487 | 0.508 | **0.457** | 0.370 | 0.352 | **0.335** | **0.363** | Inf | 0.369 | 0.363 | 0.345 | **0.340** |
| | Avg | 0.440 | 0.460 | **0.416** | **0.433** | 0.452 | 0.434 | 0.437 | 0.456 | **0.421** | 0.344 | 0.323 | **0.314** | 0.341 | Inf | **0.328** | 0.341 | **0.319** | 0.319 |
| ETTm1 | 96 | **0.293** | 0.309 | 0.345 | 0.324 | **0.317** | 0.332 | **0.294** | 0.323 | 0.298 | 0.284 | **0.244** | 0.266 | 0.297 | **0.261** | 0.273 | 0.282 | 0.258 | **0.255** |
| | 192 | 0.360 | **0.351** | 0.355 | 0.368 | **0.359** | 0.389 | 0.333 | 0.339 | **0.316** | 0.307 | **0.260** | 0.272 | 0.315 | Inf | **0.292** | 0.300 | 0.262 | **0.259** |
| | 336 | 0.398 | **0.378** | 0.409 | 0.401 | 0.390 | **0.380** | 0.375 | 0.373 | **0.350** | 0.323 | **0.280** | 0.295 | 0.328 | Inf | **0.301** | 0.317 | **0.278** | 0.285 |
| | 720 | 0.478 | 0.559 | **0.445** | 0.439 | 0.460 | 0.445 | 0.428 | 0.448 | **0.405** | 0.351 | 0.340 | **0.316** | 0.346 | **0.321** | 0.328 | 0.339 | 0.312 | **0.310** |
| | Avg | **0.382** | 0.399 | 0.388 | 0.383 | **0.381** | 0.386 | 0.357 | 0.371 | **0.342** | 0.316 | **0.281** | 0.287 | 0.321 | Inf | **0.299** | 0.310 | 0.277 | **0.277** |
| ETTh2 | 96 | 0.332 | **0.326** | 0.329 | **0.312** | 0.320 | 0.319 | **0.288** | 0.338 | 0.296 | 0.285 | **0.286** | 0.288 | 0.278 | **0.275** | 0.281 | **0.260** | 0.281 | 0.265 |
| | 192 | 0.399 | 0.394 | **0.373** | 0.379 | 0.388 | **0.373** | **0.364** | 0.394 | 0.372 | 0.316 | 0.320 | **0.311** | 0.313 | **0.312** | 0.314 | **0.296** | 0.310 | 0.306 |
| | 336 | 0.393 | 0.421 | **0.373** | 0.389 | **0.374** | 0.384 | 0.393 | 0.408 | **0.371** | 0.324 | 0.330 | **0.319** | 0.321 | Inf | **0.317** | **0.317** | 0.320 | 0.318 |
| | 720 | 0.428 | 0.472 | **0.419** | 0.425 | **0.417** | 0.430 | 0.418 | 0.435 | **0.408** | 0.351 | 0.356 | **0.346** | 0.345 | **0.329** | 0.340 | 0.338 | Inf | **0.331** |
| | Avg | 0.388 | 0.403 | **0.374** | 0.376 | **0.375** | 0.376 | 0.366 | 0.394 | **0.362** | 0.319 | 0.323 | **0.316** | 0.314 | Inf | **0.313** | **0.303** | Inf | 0.305 |
| ETTm2 | 96 | **0.166** | 0.177 | 0.174 | 0.185 | 0.182 | **0.173** | 0.184 | **0.182** | 0.182 | 0.197 | 0.200 | **0.195** | 0.211 | **0.197** | 0.206 | 0.207 | **0.196** | 0.200 |
| | 192 | 0.234 | **0.226** | 0.241 | 0.247 | **0.241** | 0.265 | **0.248** | 0.257 | 0.256 | 0.228 | **0.225** | 0.231 | **0.243** | Inf | 0.244 | 0.239 | **0.231** | 0.244 |
| | 336 | 0.299 | 0.297 | **0.272** | **0.300** | 0.324 | 0.307 | 0.310 | 0.319 | **0.303** | 0.264 | 0.260 | **0.251** | 0.270 | 0.303 | **0.261** | 0.273 | **0.262** | 0.270 |
| | 720 | **0.387** | 0.391 | 0.391 | **0.383** | 0.391 | 0.397 | 0.405 | 0.406 | **0.382** | 0.309 | **0.308** | 0.318 | **0.313** | Inf | 0.317 | 0.318 | **0.309** | 0.318 |
| | Avg | 0.271 | 0.273 | **0.270** | **0.279** | 0.285 | 0.285 | 0.287 | 0.291 | **0.281** | 0.249 | **0.248** | 0.249 | 0.259 | Inf | **0.257** | 0.259 | **0.250** | 0.258 |
| WTH | 96 | **0.144** | 0.156 | 0.148 | 0.151 | 0.170 | **0.149** | **0.149** | 0.162 | 0.150 | 0.166 | **0.147** | 0.150 | 0.172 | Inf | **0.156** | 0.168 | 0.148 | **0.147** |
| | 192 | **0.196** | 0.209 | 0.199 | **0.188** | 0.249 | 0.191 | 0.194 | 0.211 | **0.192** | 0.202 | 0.186 | **0.185** | 0.198 | 0.211 | **0.192** | 0.199 | Inf | **0.181** |
| | 336 | **0.250** | 0.263 | 0.253 | **0.241** | 0.305 | 0.248 | **0.247** | 0.330 | 0.248 | 0.238 | 0.219 | **0.217** | 0.232 | Inf | **0.227** | 0.233 | 0.251 | **0.216** |
| | 720 | **0.322** | 0.336 | 0.326 | **0.323** | 0.368 | 0.340 | 0.328 | 0.377 | **0.333** | 0.279 | **0.255** | 0.263 | 0.281 | Inf | **0.267** | 0.279 | 0.276 | **0.261** |
| | Avg | **0.228** | 0.241 | 0.232 | **0.226** | 0.273 | 0.232 | **0.230** | 0.270 | 0.231 | 0.221 | **0.202** | 0.204 | 0.221 | Inf | **0.210** | 0.220 | Inf | **0.201** |
| ECL | 96 | 0.138 | 0.147 | **0.133** | **0.134** | 0.152 | 0.135 | **0.130** | 0.137 | **0.130** | 0.261 | Inf | **0.170** | 0.260 | Inf | **0.170** | 0.258 | **0.166** | 0.166 |
| | 192 | 0.156 | 0.168 | **0.154** | 0.155 | 0.171 | **0.154** | 0.149 | 0.156 | **0.147** | 0.268 | Inf | **0.186** | 0.267 | Inf | **0.183** | 0.265 | 0.179 | **0.178** |
| | 336 | **0.176** | 0.193 | 0.187 | **0.171** | 0.194 | 0.172 | 0.166 | 0.173 | **0.164** | 0.276 | Inf | **0.209** | 0.275 | Inf | **0.196** | 0.273 | 0.192 | **0.191** |
| | 720 | 0.210 | 0.222 | **0.208** | 0.210 | 0.231 | **0.206** | 0.204 | 0.214 | **0.202** | 0.291 | Inf | **0.224** | 0.291 | Inf | **0.222** | 0.289 | 0.238 | **0.216** |
| | Avg | **0.170** | 0.182 | **0.170** | 0.168 | 0.187 | **0.167** | 0.162 | 0.170 | **0.161** | 0.274 | Inf | **0.197** | 0.273 | Inf | **0.193** | 0.271 | 0.194 | **0.188** |
| Traffic | 96 | 0.386 | 0.427 | **0.398** | 0.390 | 0.425 | **0.383** | **0.371** | 0.408 | 0.375 | 0.339 | 0.205 | **0.198** | 0.337 | Inf | **0.192** | 0.335 | 0.195 | **0.189** |
| | 192 | 0.412 | 0.443 | **0.409** | 0.412 | 0.446 | **0.409** | 0.395 | 0.427 | **0.394** | 0.344 | 0.211 | **0.202** | 0.341 | Inf | **0.199** | 0.341 | 0.203 | **0.195** |
| | 336 | 0.422 | 0.460 | **0.419** | 0.431 | 0.463 | **0.429** | **0.402** | 0.442 | 0.406 | 0.348 | 0.216 | **0.207** | 0.344 | Inf | **0.207** | 0.342 | 0.207 | **0.201** |
| | 720 | **0.454** | 0.507 | **0.454** | 0.477 | 0.536 | **0.464** | 0.446 | 0.488 | **0.440** | 0.362 | 0.234 | **0.223** | 0.358 | Inf | **0.222** | 0.358 | 0.226 | **0.218** |
| | Avg | **0.418** | 0.459 | 0.420 | 0.428 | 0.468 | **0.421** | **0.403** | 0.441 | 0.404 | 0.348 | 0.217 | **0.208** | 0.345 | Inf | **0.205** | 0.344 | 0.208 | **0.201** |
| Dynamic | 60 | 0.238 | 0.326 | **0.237** | **0.238** | 0.367 | **0.238** | **0.241** | 0.335 | 0.245 | 0.201 | Inf | **0.119** | 0.202 | 0.172 | **0.130** | 0.204 | 0.149 | **0.132** |
| | 120 | 0.311 | 0.432 | **0.308** | 0.313 | 0.454 | **0.311** | **0.316** | 0.441 | **0.316** | 0.237 | Inf | **0.171** | 0.239 | Inf | **0.187** | 0.242 | 0.204 | **0.183** |
| | 180 | **0.366** | 0.493 | 0.380 | 0.369 | 0.474 | **0.368** | **0.376** | 0.511 | 0.379 | 0.266 | Inf | **0.215** | 0.268 | Inf | **0.225** | 0.273 | 0.241 | **0.222** |
| | 300 | **0.441** | 0.570 | 0.472 | **0.446** | 0.565 | **0.446** | **0.462** | 0.615 | 0.458 | 0.307 | Inf | **0.271** | 0.310 | Inf | **0.274** | 0.317 | 0.293 | **0.270** |
| | Avg | **0.339** | 0.455 | 0.349 | 0.342 | 0.465 | **0.341** | **0.349** | 0.476 | 0.350 | 0.253 | Inf | **0.194** | 0.255 | Inf | **0.204** | 0.259 | 0.222 | **0.202** |

## F  ADDITIONAL EXPERIMENTS ON HYPER-PARAMETERS

Table 10: Metrics comparison of different noise schedules on ETTh1 ($T = 336, \tau = 96$). The default schedule used in main experiments is marked in gray.

|  | Top-3 MSE | Top-3 MAE | MSE | CRPS |
|---|---|---|---|---|
| Quadratic | **0.314** | **0.362** | 0.379 | **0.286** |
| Cosine | 0.317 | 0.364 | 0.382 | 0.289 |
| Linear | 0.329 | 0.371 | **0.375** | 0.289 |

Table 11: Top-3 MSE/MAE evaluations versus varying maximum number of modes $M$ in Algorithm 1 on ETTh1 ($T = 336, \tau = 96$). The default setting in main experiments is marked in gray.

| $M$ | 5 | 10 | 15 | 20 | 25 |
|---|---|---|---|---|---|
| Top-3 MSE | 0.3200 | 0.3197 | 0.3200 | 0.3199 | 0.3203 |
| Top-3 MAE | 0.3667 | 0.3667 | 0.3668 | 0.3667 | 0.3668 |

Table 12: Top-3 MSE/MAE evaluations versus varying mixture weights prior hyperparameter $\rho$ in Algorithm 1 on ETTh1 ($T = 336, \tau = 96$).

| $\rho$ | 0.1 | 0.3 | 0.5 | 0.7 | 0.9 |
|---|---|---|---|---|---|
| Top-3 MSE | 0.329 | 0.322 | 0.320 | 0.314 | 0.307 |
| Top-3 MAE | 0.377 | 0.378 | 0.367 | 0.358 | 0.354 |

Table 13: Top-3 MSE/MAE evaluations versus varying variance prior hyperparameter $u$ in Algorithm 1 on ETTh1 ($T = 336, \tau = 96$).

| $u$ | 0.001 | 0.01 | 0.1 | 1 | 10 | 100 | 1000 |
|---|---|---|---|---|---|---|---|
| Top-3 MSE | 0.322 | 0.321 | 0.323 | 0.323 | 0.318 | 0.320 | 0.314 |
| Top-3 MAE | 0.372 | 0.371 | 0.372 | 0.372 | 0.361 | 0.367 | 0.364 |

**Noise Schedule.** In our main experiments, we use the linear schedule as default. Table 10 further evaluates two advanced schedules: Quadratic (Kong & Ping, 2021) and Cosine (Nichol & Dhariwal, 2021). Results show that MMPD benefits from these advanced schedules. This indicates that developing time-series-specific schedules is a promising direction, since time series and image data have fundamentally different characteristics.

**Diffusion Steps in Training.** Fig. 7 demonstrates that increasing the number of training diffusion steps $K_{train}$ significantly improves Top-3 MSE, Top-3 MAE and MSE. Notably, this enhancement comes without computational overhead during inference, as we employ resampling with fixed inference steps $K_{infer}$. For our experiments, we adopt $K_{train} = 1,000$ as the default setting.

**Diffusion Steps in Inference.** Fig. 8 shows that increasing the number of inference diffusion steps $K_{infer}$ consistently improves prediction accuracy. However, as revealed by our efficiency analysis in Table 4, this improvement comes with increased computational overhead. To achieve a balance between accuracy and efficiency, we set $K_{infer} = 20$ as our default configuration.

**Anchor step $k^*$ in Eq. 8.** Fig. 9 shows that overly large $\bar{\alpha}_{k^*}$ harms multi-mode and probabilistic prediction, while small ones degrade deterministic prediction. The performance is robust across a broad range around $\bar{\alpha}_{k^*} = 0.5$. To maintain a simple formulation of Eq. 8, we choose $k^*$ to make $\bar{\alpha}_{k^*}$ close to 0.5.

**Maximum number of modes $M$ in Algorithm 1.** Table 11 shows that the multi-mode accuracy remains robust w.r.t $M$. This arises from using a variational GMM rather than a standard one. In this formulation, only the maximum number of modes needs to be specified, while the inference algorithm automatically determines the appropriate number of active modes. This behavior is illustrated in Fig. 3(c), where we set $M = 10$, but only 3 modes are activated after inference.

**Mixture weights prior hyperparameter $\rho$ in Algorithm 1.** $\rho$ controls the prior over the number of activated modes-a larger $\rho$ encourages utilizing more modes. Table 12 shows that larger $\rho$ leads to slightly lower Top-3 MSE/MAE, but activating too many modes may confuse downstream users.

**Variance prior hyperparameter $u$ in Algorithm 1.** $u$ influences the prior over the variance, where a larger value reflects greater confidence in the variance estimated from forward diffusion. As shown in the Table 13 below, performance remains robust.

## G  DETAILED FLOPS ANALYSIS

**FLOPs of MSE Loss.** MSE loss uses a conventional MLP to project latent tokens into future series. A conventional $S$-block MLP consists of three components:

- One linear input layer with $O(d^2)$ FLOPs;

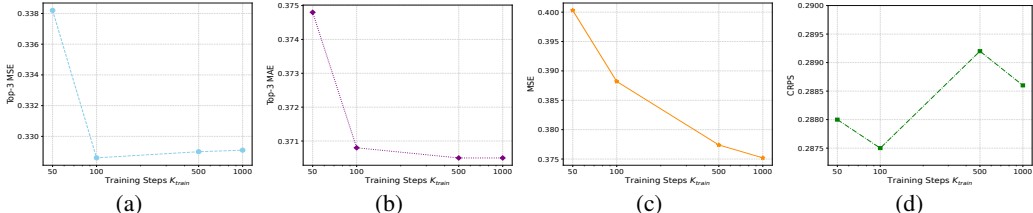

(a)    (b)    (c)    (d)

Figure 7: (a) Top-3 MSE, (b) Top-3 MAE, (c) MSE, (d) CRPS evaluations versus varying diffusion steps in training ($K_{train}$) on ETTh1 ($T = 336, \tau = 96$). The Linear noise schedule is used and the diffusion steps in inference are set to $K_{infer} = 20$.

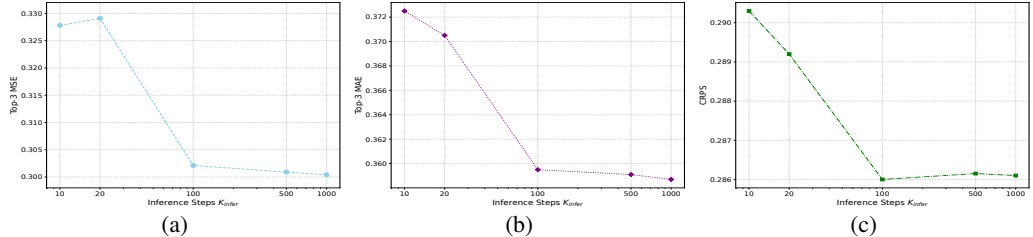

(a)      (b)      (c)

Figure 8: (a) Top-3 MSE, (b) Top-3 MAE, (c) CRPS evaluations versus varying diffusion steps in inference ($K_{infer}$). $K_{train} = 1,000$ and other settings are same as Fig. 7. MSE is omitted as $K_{infer}$ does not affect deterministic prediction.

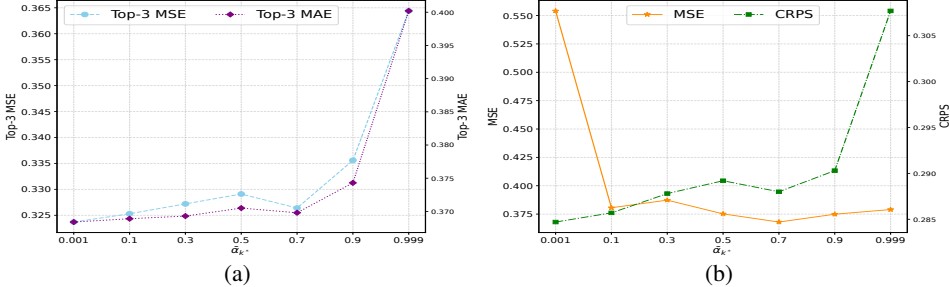

(a)        (b)

Figure 9: (a) Top-3 MSE & Top-3 MAE, (b) MSE & CRPS evaluations versus varying $\bar{\alpha}_{k*}$ in Eq. 8 on ETTh1 ($T = 336, \tau = 96$).

- $S$ hidden blocks, each has two linear layers and one activation layer between them. The FLOPs are $O(2d^2)$;
- One linear output layer with $O(Pd)$ FLOPs;

Therefore, the FLOPs for a conventional MLP applied to one token is $O\left((2S + 1)d^2 + Pd\right)$. To predict a series of length $\tau$, this MLP is simultaneously applied to $\tau/P$ tokens, making the total FLOPs $F_{MLP} = O(\frac{\tau}{P}[(2S + 1)d^2 + Pd])$.

For both training and deterministic inference, MSE loss requires one backbone forward pass and one MLP forward pass, so the FLOPs are $F_{bkb} + F_{MLP}$. It should be noted that MSE loss cannot perform probabilistic or multi-mode prediction, marked with "N/A" in Tables 4.

**FLOPs of MMPD Loss.** MMPD loss uses Patch Consistent MLP as the denoising network, which is slightly more complex than the conventional MLP above. A $S$-block Patch Consistent MLP consists of three components:

- Input operations in Eq. 7, requiring $O(d^2 + (2r + 1)Pd)$ FLOPs;
- $S$ AdaLN-MLP blocks in Eq. 12, each requires $O(5d^2)$ FLOPs;
- One AdaLN output block in Eq. 13, requiring $O(2d^2 + Pd)$ FLOPs;

Same as MSE, the Patch Consistent MLP is simultaneously applied to $\tau/P$ tokens, so the total FLOPs are $F_{PC-MLP} = O(\frac{\tau}{P}[(5S + 3)d^2 + (2r + 2)Pd])$.

As for EM steps in Algorithm 1, three vector operations contribute most FLOPs and other scalar operations are negligible. The three vector operations are: 1) Computation of $\ln \gamma_{nm}^k$ requires $O(3\tau MN)$ ; 2) $\boldsymbol{\mu}_m^k$ requires $O(2\tau MN)$; 3) $\widetilde{v}_m^k$ requires $O(3\tau MN)$ . So one EM step has the FLOPs of $F_{EM} = O(8\tau MN)$.

For training, MMPD Loss requires one backbone forward pass and two Patch Consistent MLP passes: one for the diffusion objective and another for the deterministic objective in Eq. 8. So the training FLOPs are $F_{bkb} + 2F_{PC-MLP}$. Considering the backbone dominates training cost (i.e., $F_{bkb} >> F_{MLP}, F_{PC-MLP}$), training FLOPs of MMPD remain nearly identical to those of MSE.

Similar to MSE, deterministic inference of MMPD loss requires one backbone forward pass and one Patch Consistent MLP forward pass, so the FLOPs are $F_{bkb} + F_{PC-MLP}$, differences only lie in MLP architectures. For probabilistic and multi-mode predictions, MMPD needs to generate $N$ samples, each through $K$ diffusion-EM iterations. So the FLOPs are $F_{bkb} + K(NF_{PC-MLP} + F_{EM})$. As $F_{EM} << NF_{PC-MLP}$, we omit it in Table 4 for simplicity. To generate one instance, MMPD loss only requires a single backbone pass, followed by $K$ lightweight MLP passes, making its cost significantly lower than that of TS Diffusion models requiring $K$ backbone passes.

## H  FURTHER DISCUSSION: EXTENDING MMPD TO NON-PATCH-BASED BACKBONES

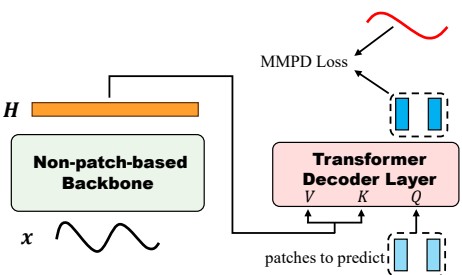

Figure 10: Adapting non-patch-based backbones to use MMPD loss: a single Transformer decoder layer is inserted between the backbone and MMPD. In this layer, backbone output $\mathbf{H}$ serves as key and value, while learnable tokens indicating patches to predict act as the queries. The decoder output is then used for MMPD loss computation.

The latent representations $\mathbf{H}$ extracted by non-patch-based backbones are not naturally expressed as $\{\mathbf{h}_j\}_{j=1}^l$, thus MMPD loss cannot be directly applied. As illustrated in Fig. 10, we address this by inserting a single Transformer decoder layer: $\mathbf{H}$ serves as key and value, while learnable tokens indicating prediction patches act as query. This decoder layer transforms $\mathbf{H}$ into tokens suitable for MMPD loss.

Using this adaptation, we apply MMPD to following non-patch-based backbones: **1) TSMixer** (Chen et al., 2023), a fully MLP model without patching; **2) iTransformer** (Li et al., 2024b), a non-patch-based Transformer designed to model inter-channel dependencies. Table 14 shows that our adaptation with MMPD loss performs on par with the original architectures using MSE loss in deterministic forecasting, while significantly outperforming them in multi-mode and probabilistic forecasting. This demonstrates the effectiveness of our adaptation and highlights the potential of applying MMPD to a broader range of backbones.

Table 14: Comparison of non-patch-based backbones (TSMixer and iTransformer) trained using MSE loss and adapted MMPD loss on datasets ETTh1/ETTm1/WTH, $T = 336, \tau = 192$.

| Dataset | | ETTh1 | ETTm1 | WTH | ETTh1 | ETTm1 | WTH | ETTh1 | ETTm1 | WTH | ETTh1 | ETTm1 | WTH |
|---|---|---|---|---|---|---|---|---|---|---|---|---|---|
| Metric | | Top-3 MSE | | | Top-3 MAE | | | MSE | | | CRPS | | |
| TSMixer | MSE | 0.384 | 0.285 | 0.140 | 0.415 | 0.353 | 0.198 | 0.390 | 0.306 | 0.149 | 0.323 | 0.289 | 0.169 |
| | MMPD | **0.367** | **0.230** | **0.118** | **0.396** | **0.309** | **0.161** | **0.378** | 0.306 | **0.148** | **0.304** | **0.260** | **0.150** |
| iTransformer | MSE | 0.390 | 0.298 | 0.147 | 0.414 | 0.361 | 0.200 | 0.400 | **0.317** | **0.158** | 0.324 | 0.292 | 0.173 |
| | MMPD | **0.361** | **0.279** | **0.132** | **0.395** | **0.326** | **0.180** | **0.386** | 0.334 | 0.170 | **0.304** | **0.278** | **0.165** |

## I  FURTHER DISCUSSION: EXTENDING MMPD TO MULTI-TASK LEARNING

Beyond the traditional single-dataset paradigm, we further extend the MMPD loss to multi-task learning, where a unified model is trained across multiple datasets and settings. Such a model can directly perform multi-task forecasting and be adapted to few-shot or zero-shot forecasting on new datasets.

Table 15: Multi-task forecasting comparison of UNITS pretrained with MSE and MMPD loss on 20 forecasting tasks.

| Metric | MSE | | MAE | | Top-3 MSE | | Top-3 MAE | | CRPS | |
|---|---|---|---|---|---|---|---|---|---|---|
| Pretraining Loss | MSE | MMPD | MSE | MMPD | MSE | MMPD | MSE | MMPD | MSE | MMPD |
| NN5 | 0.618 | **0.595** | 0.551 | **0.530** | 0.623 | **0.572** | 0.559 | **0.512** | 0.453 | **0.390** |
| ECL-96 | 0.168 | **0.161** | 0.270 | **0.260** | 0.178 | **0.146** | 0.285 | **0.241** | 0.271 | **0.189** |
| EC-192 | 0.184 | **0.175** | 0.283 | **0.272** | 0.195 | **0.163** | 0.301 | **0.256** | 0.280 | **0.201** |
| ECL-336 | 0.203 | **0.192** | 0.300 | **0.289** | 0.203 | **0.174** | 0.313 | **0.270** | 0.285 | **0.211** |
| ECL-720 | 0.242 | **0.229** | 0.331 | **0.318** | 0.234 | **0.215** | 0.338 | **0.302** | 0.300 | **0.234** |
| ETTh1-96 | 0.397 | **0.367** | 0.420 | **0.400** | 0.377 | **0.325** | 0.413 | **0.367** | 0.334 | **0.286** |
| ETTh1-192 | 0.438 | **0.408** | 0.448 | **0.427** | 0.433 | **0.386** | 0.443 | **0.404** | 0.353 | **0.312** |
| ETTh1-336 | 0.465 | **0.442** | 0.465 | **0.448** | 0.456 | **0.436** | 0.458 | **0.425** | 0.365 | **0.325** |
| ETTh1-720 | 0.507 | **0.472** | 0.500 | **0.478** | 0.513 | **0.474** | 0.496 | **0.464** | 0.394 | **0.347** |
| Exchange-192 | 0.261 | **0.225** | 0.364 | **0.343** | 0.223 | **0.160** | 0.338 | **0.279** | 0.318 | **0.274** |
| Exchange-336 | 0.464 | **0.415** | 0.494 | **0.469** | 0.349 | **0.306** | 0.429 | **0.393** | 0.439 | **0.372** |
| ILI | **2.073** | 2.345 | **0.895** | 0.967 | 2.045 | **1.981** | 0.894 | **0.855** | **0.739** | 0.785 |
| Traffic-96 | 0.475 | **0.446** | 0.314 | **0.288** | 0.483 | **0.414** | 0.343 | **0.265** | 0.352 | **0.211** |
| Traffic-192 | 0.484 | **0.460** | 0.314 | **0.292** | 0.477 | **0.426** | 0.336 | **0.268** | 0.350 | **0.211** |
| Traffic-336 | 0.498 | **0.477** | 0.319 | **0.299** | 0.519 | **0.465** | 0.349 | **0.280** | 0.357 | **0.219** |
| Traffic-720 | 0.532 | **0.510** | 0.336 | **0.315** | 0.545 | **0.507** | 0.359 | **0.299** | 0.361 | **0.233** |
| Weather-96 | **0.163** | 0.166 | 0.214 | **0.213** | 0.149 | **0.136** | 0.203 | **0.173** | 0.176 | **0.166** |
| Weather-192 | **0.212** | 0.213 | 0.257 | **0.254** | **0.178** | 0.180 | 0.236 | **0.220** | 0.212 | **0.202** |
| Weather-336 | **0.267** | 0.270 | 0.297 | **0.294** | 0.269 | **0.245** | 0.268 | **0.267** | 0.251 | **0.241** |
| Weather-720 | **0.344** | 0.353 | 0.347 | **0.345** | **0.323** | 0.328 | **0.317** | 0.327 | 0.300 | **0.287** |
| Winning Counts | 5/20 | **15/20** | 1/20 | **19/20** | 2/20 | **18/20** | 1/20 | **19/20** | 1/20 | **19/20** |

Table 16: Few-shot forecasting comparison of UNITS tuned with MSE and MMPD loss on new datasets. For each setting, only 5% of the training set is used for prompt-based tuning.

| Metric | MSE | | MAE | | Top-3 MSE | | Top-3 MAE | | CRPS | |
|---|---|---|---|---|---|---|---|---|---|---|
| Prompt Tuning Loss | MSE | MMPD | MSE | MMPD | MSE | MMPD | MSE | MMPD | MSE | MMPD |
| ETTh2-96 | 0.409 | **0.377** | 0.415 | **0.403** | 0.382 | **0.341** | 0.409 | **0.374** | 0.326 | **0.292** |
| ETTh2-192 | **0.380** | 0.381 | **0.398** | 0.403 | 0.365 | **0.335** | 0.389 | **0.373** | 0.343 | **0.318** |
| ETTh2-336 | **0.436** | 0.447 | **0.437** | 0.445 | 0.415 | **0.397** | 0.431 | **0.413** | 0.361 | **0.347** |
| ETTh2-720 | 0.449 | **0.445** | 0.454 | **0.453** | 0.431 | **0.420** | 0.458 | **0.447** | 0.382 | **0.361** |
| SaugeenRiverFlow | 1.270 | **1.248** | 0.576 | **0.569** | 1.128 | **1.088** | 0.510 | **0.485** | 0.543 | **0.480** |
| Winning Counts | 2/5 | **3/5** | 2/5 | **3/5** | 0/5 | **5/5** | 0/5 | **5/5** | 0/5 | **5/5** |

Table 17: Zero-shot forecasting comparison on new datasets of pretrained UNITS using MSE and MMPD.

| Metric | MSE | | MAE | | Top-3 MSE | | Top-3 MAE | | CRPS | |
|---|---|---|---|---|---|---|---|---|---|---|
| Pretraining Loss | MSE | MMPD | MSE | MMPD | MSE | MMPD | MSE | MMPD | MSE | MMPD |
| Solar | 0.202 | **0.169** | 0.320 | **0.299** | 0.207 | **0.095** | 0.331 | **0.193** | 0.272 | **0.187** |
| River | 2.336 | **2.294** | **0.735** | 0.746 | 2.298 | **1.722** | 0.733 | **0.722** | 0.643 | **0.624** |
| Hospital | **1.115** | 1.175 | **0.818** | 0.834 | 0.998 | **0.860** | 0.772 | **0.705** | 0.690 | **0.622** |
| Winning Counts | 1/3 | **2/3** | 2/3 | 1/3 | 0/3 | **3/3** | 0/3 | **3/3** | 0/3 | **3/3** |

We evaluate this idea using UNITS (Gao et al., 2024), a unified multi-task model that integrates multiple tasks within a single framework. Since our focus is solely on forecasting, we adopt the supervised variant (UNITS-SUP) without incorporating classification techniques. As a patch-based model originally trained and tuned with MSE loss, MMPD loss can be integrated into it with minimal changes to enhance its ability to capture complex distributions.

**Multi-task Forecasting.** Following Gao et al. (2024), we first conduct multi-task supervised pretraining on 20 forecasting tasks using either MSE or MMPD loss. Results in Table 15 show that for deterministic forecasting measured by MSE/MAE, MMPD loss matches or improves performance over MSE. A possible reason is that MMPD provides a more challenging objective that encourages richer representations, which are especially beneficial in multi-task scenarios. For multi-mode and probabilistic forecasting measured by Top-3 MSE/MAE and CRPS, UNITS trained with MMPD significantly outperforms its MSE-trained counterpart. This demonstrates that MMPD integrates well with UNITS and effectively enables the modeling of complex distributions.

**Few-shot Forecasting.** We then tune the pretrained models with MSE and MMPD loss to perform few-shot prediction on new datasets. In each setting, only 5% of the training data is utilized and the prompt-based tuning from Gao et al. (2024) is used. As shown in Table 16, MMPD matches MSE in deterministic accuracy while consistently outperforming it in probabilistic and multi-mode forecasting.

**Zero-shot Forecasting.** Finally, Table 17 evaluates zero-shot forecasting capabilities of models pretrained with MSE and MMPD. Results are consistent with the few-shot setting: MMPD maintains comparable deterministic accuracy while offering superior probabilistic and multi-mode performance.

# J   VISUALIZATIONS

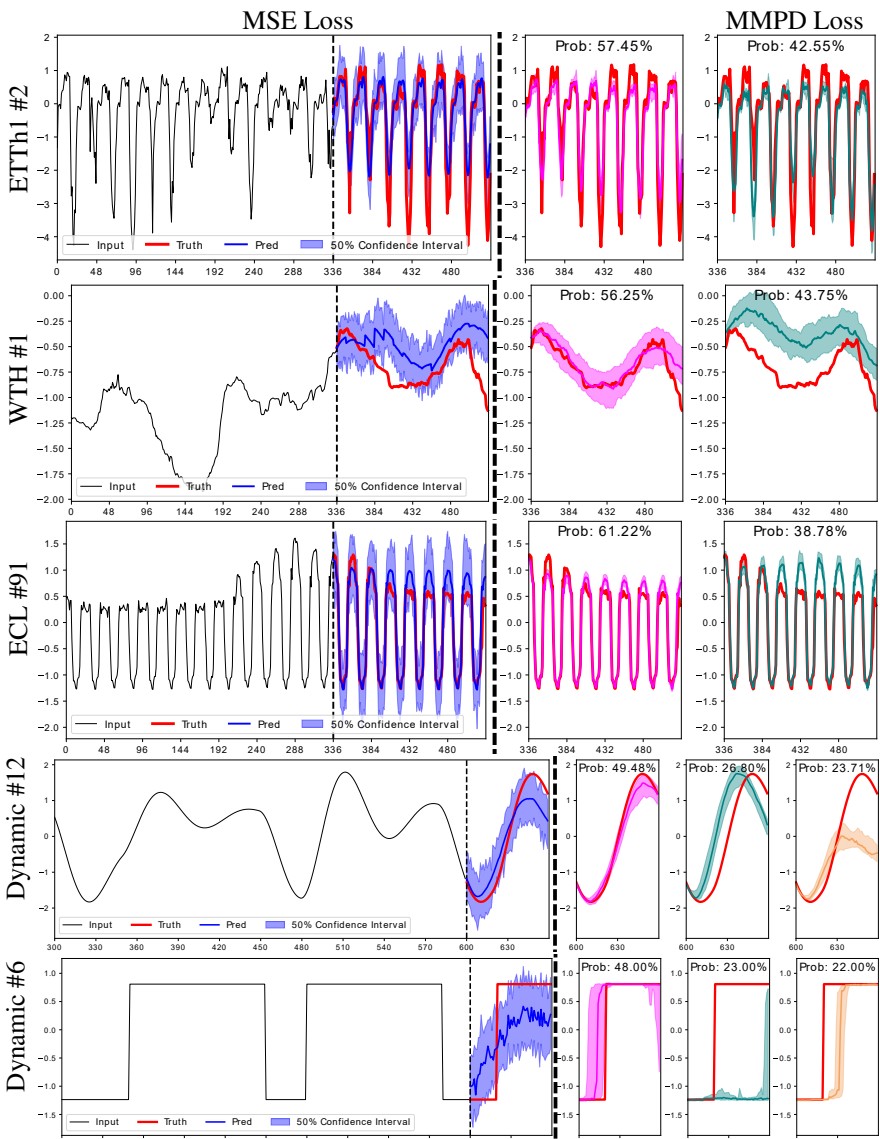

Figure 11: Forecasting cases of MSE loss *vs.* MMPD loss. Each row represents one instance, with the dataset name and channel number indicated on the left. Predictions from MSE and MMPD losses are separated by a thick vertical dashed line. On the left, one-mode predictions generated by the MSE loss are shown, along with the corresponding input series. On the right, multi-mode predictions generated by the MMPD loss are displayed, with their corresponding probabilities shown at the top.

# K   DECLARATION OF LARGE LANGUAGE MODELS USAGE

In this work, large language models were used solely for word choice and language polishing. They did not contribute to research ideation, methodology or analysis.

