# OpenReview forum: "MMPD: Diverse Time Series Forecasting via Multi-Mode Patch Diffusion Loss"
_ICLR.cc/2026/Conference — ICLR 2026 Poster_

### Official Review · Reviewer_kPh3 · 2025-10-28

**Soundness:** 4
**Presentation:** 4
**Contribution:** 4
**Rating:** 6
**Confidence:** 4

**Summary:**

The paper proposes MMPD as a training objective: it turns the loss into a conditional diffusion process. Given future latent tokens produced by an upstream patch-based backbone, a lightweight Patch-Consistent MLP performs denoising to model complex, multi-modal future distributions; at inference, a progressively-evolving variational GMM extracts several plausible futures along with their probabilities.

**Strengths:**

- **S1** Solid work with complete design and experiments: from motivation and method (loss = diffusion + Patch-Consistent MLP) to inference (evolving variational GMM) and ablations/efficiency analysis, the pipeline is very clear and well executed.
- **S2** Motivation is important and well argued: the paper clearly articulates that MSE-centric training implicitly assumes a unimodal Gaussian with fixed variance, which struggles to cover real-world multiple possible futures and time-varying/asymmetric uncertainty. The authors cleverly interpret MSE through the lens of a distributional assumption and then extend it to a higher-level, more expressive formulation.

**Weaknesses:**

- **W1** Baseline coverage is incomplete: while comparisons include point-estimation and certain probabilistic/mixed-distribution schemes, the work does not compare against D3U[1], which follows a related paradigm of decoupling deterministic and uncertain components.
- **W2** Patch-based pipelines typically introduces padding. For these non-real-value patches, it is unclear whether the Patch-Consistent MLP applies an explicit distinction been made. Moreover, padding patches lack a valid $\text{next}_{k_j}$ to model. The paper should provide further implementation details and experiments clarifying how padding patches are treated.

[1]. Li, Q., Zhang, Z., Yao, L., Li, Z., Zhong, T., & Zhang, Y. (2025). Diffusion-based decoupled deterministic and uncertain framework for probabilistic multivariate time series forecasting. In The Thirteenth International Conference on Learning Representations.

**Questions:**

See W1 and W2. If the author can address my concerns, I will consider increasing my score.

---

> ### Author Response · Authors · 2025-11-19
> **Response to Reviewer kPh3**
>
> Thank you for your time and effort in reviewing our work. Your valuable comments, especially the related work you pointed out, will help us further improve the paper. Below are our responses to your specific questions.
>
> # 1. Comparison with D3U (Weakness 1)
>
> D3U is a closely related approach, as it also decomposes forecasting into deterministic and uncertain (probabilistic) components. Unlike our joint training paradigm, it adopts a two-stage hierarchical pipeline: 1)pretraining a point forecasting model to predict the deterministic component; 2)training a conditional diffusion model to predict the residual uncertainty, the condition is obtained from the pretrained deterministic model.
>
> We compare a decoder-only Transformer trained with our MMPD against D3U. The results in the following table show that our approach consistently outperforms D3U across datasets. We observed two advantages of MMPD over D3U: 1)MMPD is an end-to-end solution while D3U requires separate deterministic pretraining followed by diffusion training; 2)our patch-consistent MLP denoiser is more lightweight than the denoising Transformer used in D3U.
>
> *Evaluation of MMPD against D3U on ETTh1, ETTm1 and Weather. The prediction horizon is set to $\tau=192$.*
> |Dataset|Method|Top-3 MSE|Top-3 MAE|MSE|CRPS|
> |-|-|-|-|-|-|
> |ETTh1|D3U|0.407|0.424|0.472|0.348|
> ||Decoder+MMPD(ours)|**0.396**|**0.405**|**0.406**|**0.314**|
> |ETTm1|D3U|0.272|0.350|0.363|0.298|
> ||Decoder+MMPD(ours)|**0.240**|**0.314**|**0.315**|**0.259**|
> |Weather|D3U|0.163|0.226|0.214|0.210|
> ||Decoder+MMPD(ours)|**0.157**|**0.193**|**0.193**|**0.177**|
>
> We emphasize that our work is complementary to refined diffusion-based forecasting models such as D3U. MMPD leverages the diffusion process to formulate a flexible loss for complex output distributions, whereas D3U focuses on designing advanced time-series-specific diffusion processes and denoisers. In this sense, MMPD can integrate and benefit from such improvements, rather than just competing with them.
>
> # 2. Padding operation for non-divisible input and output lengths (Weakness 2)
>
> For cases where the input length ($T$) or output length ($\tau$) is not divisible by the patch size ($P$), i.e., $T \bmod P \neq 0$ or $\tau \bmod P \neq 0$, we apply nearest-value padding to make both lengths divisible by $P$. Specifically, for the input $x = [x_1, \dots, x_T]$, we pad:
>
> $$x' = [\underbrace{x_1, \dots, x_1}_{P - (T \bmod P)\ \text{times}},\ x_1, \dots, x_T]$$
>
> For the target series $y = [y_1, \dots, y_\tau]$, we pad:
>
> $$y' = [y_1, \dots, y_\tau,\ \underbrace{y_\tau, \dots, y_\tau}_{P - (\tau \bmod P)\ \text{times}}]$$
>
> Thus, the new input $x'$ and target $y'$ are divisible by $P$. During inference, we first output series with padded length and then remove the padded area. **In this way, the network only processes divisible series, so no modification to patch consistent MLP has to be made.** Moreover, we can further use a mask to filter out the padded positions when computing the objective in Equation (8) for finer control.
>
> Evaluation in the following table shows that padded points introduce no significant negative impact on accuracy, with or without masking. The main reason is that, as the atomic element for patch-based models, the patch size is relatively small compared with series lengths, i.e., $P<<T,\tau$. In the padding operation, at most $P-1$ values are padded, resulting in a limited influence on the overall series.
>
> *Evaluation of MMPD with divisible and non-divisible patch sizes on ETTm1 ($T=336, \tau=192$). $\checkmark$: divisible patch size; $\times$: non-divisible. Mask indicates whether padded points are excluded in the objective calculation by masking.*
> |Patch Size|Mask|Top-3 MSE|Top-3 MAE|MSE|CRPS|
> |-|-|-|-|-|-|
> |10$\times$|False|0.241|0.314|0.317|0.259|
> |10$\times$|True|0.241|0.315|0.317|0.260|
> |12$\checkmark$|N/A|0.240|0.314|0.315|0.259|
> |15$\times$|False|0.243|0.318|0.317|0.261|
> |15$\times$|True|0.243|0.317|0.317|0.261|
>
> Additionally, whether padded or not, when predicting patches near the end of the target, the adjacent range may exceed the valid boundary, i.e., $j > l - r$ ($j$: position of the patch, $l$: number of patches in target, $r$: adjacent range). In this case, **we fill the missing patches with zeros** by defining:
>
> $$p_{j+s}^k = 0, \text{ when } j+s>l$$
>
> A similar operation is used near the beginning of the target when previous patches are unavailable. We will clarify these details for completeness in the final version.
>
> We hope these responses have adequately addressed your concerns. If you have any further questions or suggestions, please feel free to let us know — we would be happy to provide additional clarification.

---

> ### Comment · Reviewer_kPh3 · 2025-11-26
>
> You have addressed most of my concerns, good work. I have increased my score. Please include the corresponding results and analysis in the revised version of the manuscript.
>
> I would further encourage the authors to provide results in long-input/long-output settings (i.e., beyond the commonly used 96 input and {96, 192, 336, 720} output horizons) to examine whether the effectiveness of MMDP paradigm can be consistently maintained.
>
> If author can address this concern and give some useful insights, I will consider increasing my score to 10.

---

> > ### Author Response · Authors · 2025-11-26
> >
> > We sincerely appreciate your timely response and positive feedback.
> >
> > Regarding input length, **instead of the limited 96, we follow PatchTST and set the default input length to 336 in our experiments** (both original submission and rebuttal) to ensure sufficient historical context for the model.
> >
> > As you suggested, we will adjust both the input and output lengths and include the comparison with D3U. We will get back to you once the results are available.

---

> > ### Author Response · Authors · 2025-11-28
> > **Experiments on Varying Input and Output Lengths**
> >
> > Following your suggestion, we conducted a comprehensive study on performance of MMPD for different input and output lengths. Specifically, both the input length ($T$) and output length ($\tau$) are varied in $\{48, 96, 192, 336, 720\}$, resulting in $5 \times 5$ settings. We evaluate MMPD under all 25 settings and compare against D3U.
> >
> > As shown in the table below, when the input length is short ($T \le 96$), D3U performs better. However, as the input window increases ($T \ge 192$), MMPD is able to leverage the extended historical context more effectively and consistently outperforms D3U. In contrast, D3U’s performance saturates around $T=192$ and even degrades when further increasing the input length. We hypothesize that this limitation stems from D3U’s condition-injection design, which uses fully connected layers to compress the encoded patch sequence into a fixed-size condition vector. This projection disrupts the temporal structure in the representation, making it harder for the model to benefit from a longer lookback window.
> >
> >
> > *Top-3 MSE comparsion of D3U v.s. MMPD on ETTm1 with different input and output lengths. Results are shown in form of **D3U|MMPD** in each cell. Better result is marked in bold.*
> > | output\input | 48 | 96 | 192 | 336 | 720 |
> > |---|---|---|---|---|---|
> > |48|**0.329**\|0.358|0.181\|**0.175**|0.157\|**0.151**|0.170\|**0.140**|0.175\|**0.138**|
> > |96|**0.330**\|0.346|**0.220**\|0.222|0.203\|**0.187**|0.206\|**0.179**|0.256\|**0.171**|
> > |192|**0.383**\|0.410|**0.275**\|0.283|0.263\|**0.252**|0.272\|**0.238**|0.298\|**0.232**|
> > |336|**0.417**\|0.494|**0.317**\|0.326|0.304\|**0.294**|0.334\|**0.292**|0.316\|**0.290**|
> > |720|**0.474**\|0.553|**0.388**\|0.414|0.385\|**0.384**|0.399\|**0.381**|0.409\|**0.359**|
> >
> > Top-3 MAE results are consistent with the Top-3 MSE observations. The trends of MSE and CRPS with respect to input length are also similar. Notably, MMPD consistently outperforms D3U on these two metrics across most settings, as shown in the following tables.
> >
> >
> > *MSE comparsion of  **D3U|MMPD**. Better result is marked in bold.*
> > | output\input | 48 | 96 | 192 | 336 | 720 |
> > |---|---|---|---|---|---|
> > | 48 | 0.540\|**0.507** | 0.284\|**0.274** | 0.271\|**0.252** | 0.274\|**0.245** | 0.285\|**0.253** |
> > | 96 | 0.448\|**0.427** | 0.321\|**0.317** | 0.305\|**0.284** | 0.308\|**0.286** | 0.318\|**0.283** |
> > | 192 | 0.487\|**0.471** | 0.368\|**0.354** | 0.339\|**0.320** | 0.363\|**0.315** | 0.365\|**0.316** |
> > | 336 | 0.540\|**0.523** | 0.403\|**0.381** | 0.380\|**0.350** | 0.381\|**0.344** | 0.386\|**0.345** |
> > | 720 | 0.589\|**0.565** | 0.461\|**0.439** | 0.448\|**0.407** | 0.445\|**0.394** | 0.451\|**0.388** |
> >
> > *CRPS comparsion of  **D3U|MMPD**. Better result is marked in bold.*
> > | output\input | 48 | 96 | 192 | 336 | 720 |
> > |---|---|---|---|---|---|
> > | 48 | 0.343\|**0.335** | 0.251\|**0.236** | 0.247\|**0.224** | 0.249\|**0.221** | 0.262\|**0.221** |
> > | 96 | 0.315\|**0.306** | 0.271\|**0.256** | 0.263\|**0.240** | 0.267\|**0.240** | 0.287\|**0.240** |
> > | 192 | 0.333\|**0.324** | 0.289\|**0.274** | 0.278\|**0.261** | 0.298\|**0.259** | 0.306\|**0.257** |
> > | 336 | **0.350**\|0.355 | 0.303\|**0.289** | 0.295\|**0.275** | 0.304\|**0.273** | 0.300\|**0.273** |
> > | 720 | 0.372\|**0.371** | 0.352\|**0.319** | 0.320\|**0.309** | 0.331\|**0.301** | 0.334\|**0.294** |

---

### Official Review · Reviewer_hVy7 · 2025-10-28

**Soundness:** 4
**Presentation:** 4
**Contribution:** 4
**Rating:** 8
**Confidence:** 2

**Summary:**

This paper reframes time-series forecasting by making the training objective itself a conditional diffusion model. A patchified forecaster produces future latent tokens, and a lightweight, cross-patch Patch-Consistent MLP learns to denoise them, capturing inherently multi-modal futures. At inference, the method does not rely on ad-hoc sampling; instead, it tracks modes through the reverse process with a stepwise variational GMM, yielding a small set of trajectories with calibrated probabilities. The approach is plug-and-play for patch-based backbones and includes a direct deterministic path without sacrificing efficiency. Across eight benchmarks, it consistently improves Top-K diversity metrics while remaining competitive—or better—on standard MSE (point forecasting) and CRPS (probabilistic accuracy).

**Strengths:**

- **S1**: The paper distills a clear limitation of point-estimation training and turns the objective into a flexible distribution model, which is highly instructive for time-series forecasting research. Importantly, the authors make an effort to generalize beyond patch-based designs, discussing (and partially validating) how the idea can extend to non-patch backbones—thus increasing the method’s relevance to a wider set of architectures.

- **S2**: The empirical study spans diverse datasets, multiple backbones, and both deterministic and probabilistic metrics, complemented by ablations and efficiency analyses. This breadth and depth make the conclusions credible.

**Weaknesses:**

- **W1**: Please consider releasing the training/evaluation code and experiment scripts (including configs, seeds, data preprocessing, and checkpointing details). This would enable exact reproduction of tables/figures and facilitate fair downstream comparisons.

- **W2**: Recent LLM-style time-series methods also employ patching. Adding representative LLM-based patch baselines would strengthen the empirical positioning and clarify where MMPD stands relative to these emerging approaches.

- **W3**: The paper states “We focus on univariate forecasting; for multivariate data, the loss is computed per channel and averaged.” Can author discuss how MMPD handles inter-channel correlations.

**Questions:**

See W1 to W3.

---

> ### Author Response · Authors · 2025-11-19
> **Response to Reviewer hVy7**
>
> We are grateful for your valuable comments, which would help us improve the paper. The following are our responses to your specific concerns.
>
> # 1. Source code release (Weakness 1)
>
> We will release the source code in the final version, including hyperparameter settings and one-click runnable scripts, to facilitate reproduction of our results. Additionally, we will provide clear guidance to help users train their own backbones with the MMPD loss.
>
> # 2. Comparison with LLM-based foundation models (Weakness 2)
>
> As suggested, we additionally compare MMPD with the following LLM-style foundation time-series forecasting models:
>
> 1. Chronos[1]: a foundation TS forecasting model that quantizes series into discrete tokens and pretrains via LLM-style autoregression. The LLM architecture T5 is used as the backbone.
>
> 2. Moirai [2]: a foundation TS forecasting model that splits input into multi-scale patches and uses mixture distributions for complex output modeling. It employs a Transformer with a carefully designed any-variate attention mechanism as the backbone.
>
> *[1] Chronos: Learning the language of time series. TMLR, 2024.*
>
> *[2] Unified Training of Universal Time Series Forecasting Transformers. In ICML, 2024.*
>
> Both foundational models are pre-trained on large-scale datasets to perform zero-shot forecasting on unseen data. Results in the following Table show that the decoder trained with our MMPD loss achieves better performance. This is expected, since they are zero-shot foundation models, while our supervised approach can leverage training data. The comparison is not to compete with zero-shot foundation models, but to clarify MMPD’s role within the broader probabilistic forecasting landscape.
>
> *Evaluation of MMPD loss against LLM-style foundation TS forecasting models on ETTh1, $T=96, \tau=24$*.
> ||Top-3 MSE|Top-3 MAE|MSE|CRPS|
> |-|-|-|-|-|
> |Chronos|0.321|0.331|0.413|0.296|
> |Moirai|0.365|0.370|0.395|0.282|
> |Decoder+MMPD (ours)|**0.186**|**0.280**|**0.298**|**0.254**|
>
>
> # 3. Extension of MMPD to explicitly model multivariate dependency (Weakness 3)
>
> The standard approach for multivariate loss is to compute losses per channel and then average across channels. Our original solution also follows this protocol.
>
> For multivariate extension, we further clarify that modeling multivariate dependency can be decomposed into two aspects:
>
> 1. **Extracting cross-channel dependency into latent tokens.** This can be handled by cross-channel backbones and our MMPD integrates well with these backbones. Actually, the Crossformer we used in our experiments is a typical backbone of this category.
> 2. **Modeling the joint distribution of all future patches across steps and channels**. This requires extending the Patch-Consistent MLP so that each channel’s prediction incorporates information from other channels. We achieve this by adding a channel-aware conditioning term into the current MLP. Specifically, for an $S$-dimensional dataset, at denosing step $k$, we use $p^k_{j, s}$ to denote the $j$-th patch in channel $s$. The modified condition is:
>
> $$c_{j,s}^k= token_{j,s}+step^k+prev_{j,s}^k+next_{j,s}^k+channel_{j,s}^k$$
>
> $$channel_{j,s}^k=\sum_{v \ne s} W_vp^k_{j, v}$$
>
> Here, the first four terms remain identical to the univariate case. The new term $channel_{j,s}^k$ aggregates patches from all other channels at step $j$. $W_v \in \mathbb{R}^{d_{model}\times P}$ is the projection matrix for channel $v$. As a result, the extended MMPD jointly conditions on: 1) adjacent patches within the same channel; 2) patches from other channels at the same step.
>
> We implemented the above multivariate Patch Consistent MLP and compared it against the original univariate version. Results are shown in the table below.
>
> *Comparison of multivariate v.s. univariate Patch Consistent MLP on Weather ($\tau=192$). $\checkmark$ denotes the original version used in the paper.*
> |MLP|Top-3 MSE|Top-3 MAE|MSE|CRPS|
> |-|-|-|-|-|
> |Univariate$\checkmark$|0.161|0.200|0.195|0.185|
> |Multivariate|0.160|0.198|0.195|0.182|
>
> Although the multivariate version incorporates patches from other channels, the performance improvement remains marginal. The possible reason is that unlike temporal adjacency, it is hard to define "adjacent" channels, so the denosing network must simultaneously consider all other channels. Our current linear combination may not be expressive enough to capture such complex inter-channel interactions. Additionally, the receptive field should not be limited to step $j$ and incorporating temporally adjacent steps from other channels may also be necessary.
>
> Therefore, a more powerful and explicitly multivariate denoising network would be needed to fully exploit cross-channel dependencies, which would conflict with our design goal of keeping MMPD lightweight and plug-and-play. We consider this an interesting direction for future work.
>
> Thank you again for your thoughtful review. Please let us know if you have any additional questions or suggestions.

---

> ### Comment · Reviewer_hVy7 · 2025-11-27
>
> Thank for the authors' efforts. Regarding W2, I am more curious about how MMPD enhances LLM-style TSFM. Firstly, most LLM-style TSFMs adopt a patching approach to construct tokens, which aligns with the models that MMPD focuses on. Moreover, the multi-modal generation proposed by MMPD intuitively resembles the autoregressive generation of text tokens in LLM within the language space. This makes me particularly intrigued by whether MMPD could serve as a future paradigm for integrating temporal tokens with text tokens. As this represents a widely explored direction in the recent community, if MMPD is proven to have specific insight for LLM-based models, this work would be of significant value.
>
> I understand that conducting such experiments within the remaining rebuttal period can be challenging. The authors are welcome to engage solely in an exploratory discussion with me. Regardless of whether new experimental results can be provided, I am willing to recommend this excellent work to the community. Based on the authors' response, I have increased my confidence in this paper.

---

> ### Author Response · Authors · 2025-11-30
> **Integrating tempoal tokens with text tokens using MMPD**
>
> We sincerely appreciate your encouraging feedback and your increased confidence score. Your observation that MMPD could serve as a future paradigm for integrating temporal tokens with text tokens is highly insightful, and we are excited to discuss this direction further.
>
> # Multi-mode next patch prediction with MMPD
>
> A key property of MMPD is its ability to generate multiple plausible future predictions along with their corresponding probabilities. This multi-mode prediction aligns naturally with how LLMs model a probability distribution over vocabulary tokens, providing a promising foundation for connecting time-series modeling with language modeling.
>
> To align with the next-token-prediction paradigm of LLMs, we extend MMPD to autoregressive next-patch-prediction. Given temporal pathes $[p_1, p_2, \dots, p_T]$, a causal Transformer produces latent states $[h_1, h_2, \dots, h_T]$. Following the paradigm of LLM, $h_t$ is used to predict the next patch $p_{t+1}$. Since clean adjacent patches(i.e., $p_t, p_{t-1}, \dots$) are already encoded into $h_t$, consistency across patches is naturally maintained. Thus, the patch-consistent MLP can be simplified to an AdaLN-MLP conditioned only on $h_t$, forming the objective $MMPD^\phi(h_t, p_{t+1})$ for next-patch-prediction training.
>
> During inference, we autoregressively predict patches: at each step $t$, our multi-mode inference algorithm produces a set of candidate patches with associated probabilities $\{(p_{t+1}^m, w_m)\}_{m=1}^M$. This representation is analogous to a token-level probability distribution in LLMs, enabling the direct adoption of standard sampling strategies from language modeling for temporal patch generation.
>
> # Joint pretraining of time series and text in a unified model
>
> With this autoregressive multi-mode formulation, MMPD enables a single model to learn from time-series and text data jointly. The two modalities can be concatenated in a shared tokenized space (with designated boundary tokens), and the model is pretrained using a unified objective:
>
> - Cross-entropy for next-text-token prediction
> - MMPD for next-temporal-patch prediction
>
> This yields a joint model that natively supports both data types—analogous in spirit to Transfusion[1], which unifies image-text learning via a combination of image diffusion and next-token-prediction objectives.
>
> During inference, temporal patches and text tokens are generated autoregressively using a shared sampling strategy, thanks to the multi-mode formulation of MMPD loss. Such a unified model would naturally support multi-task capabilities, including text-prompted time-series generation, cross-modal reasoning and understanding.
>
> We agree that this represents a highly promising research direction for the community. Thank you again for your valuable guidance and strong support for our work. We believe this line of development will significantly enhance the applicability of LLM-style TSFMs.
>
> *[1] Zhou, C., Yu, L., Babu, A., Tirumala, K., Yasunaga, M., Shamis, L., ... & Levy, O. (2024). Transfusion: Predict the next token and diffuse images with one multi-modal model. arXiv preprint arXiv:2408.11039.*

---

### Official Review · Reviewer_XDPi · 2025-10-29

**Soundness:** 3
**Presentation:** 2
**Contribution:** 3
**Rating:** 6
**Confidence:** 3

**Summary:**

This paper addresses the limitation of MSE-based training in time series forecasting, which assumes unimodal Gaussian distributions and fails to capture multiple plausible future outcomes. The authors propose to use a diffusion model conditioned on backbone latent tokens to model multi-modal future distributions. A Patch Consistent MLP serves as the denoising network, and a variational GMM-based inference algorithm generates diverse predictions with probabilities.

**Strengths:**

1. The paper tackles the restrictive unimodal Gaussian assumption inherent in MSE loss, which is a genuine problem for real-world time series where identical inputs can lead to multiple plausible futures due to unobserved contextual factors.

2. The work provides clear motivation from both data perspective and application perspective, making the contribution practically valuable.

3. The proposed MMPD loss can be applied to any patch-based forecasting backbone without architectural changes, and the generality is validated across four different backbones, demonstrating broad applicability.

**Weaknesses:**

1. While the paper compares against MSE and Student-T, it lacks comparison with more recent and sophisticated probabilistic forecasting methods (e.g., recent normalizing flows, autoregressive diffusion models, or transformer-based probabilistic forecasters), making it difficult to assess where this stands in the current landscape.

2. Despite claims of being "lightweight," the method requires iterative diffusion steps K during training, generating N samples and fitting GMM at each reverse step during inference, and storing and processing adjacent patches. No concrete analysis of training time overhead, inference latency, or memory usage compared to MSE baseline is provided.

3. The used GMM assumptions may be too restrictive. While GMM is more flexible than single Gaussian, it still assumes modes follow Gaussian distributions with spherical covariance. Real multi-modal distributions may have asymmetric modes, correlated dimensions, or non-Gaussian shapes, limiting the method's ability to capture truly complex distributions.

**Questions:**

1.  During training, you optimize diffusion loss on random (y^k, k) pairs. During inference, you fit GMM by assuming samples follow Eq. 10. Is there a theoretical or empirical guarantee that the learned diffusion model produces samples consistent with this GMM structure?

2. The diffusion variance schedule is critical. Do you use standard schedules from image diffusion, or is adaptation needed for time series?

---

> ### Author Response · Authors · 2025-11-19
> **Response to Reviewer XDPi (1/2)**
>
> Thank you for your time and effort in reviewing our work. Your valuable comments will help us further improve our work. Our responses to your specific concerns are below:
>
> # 1. Comparison with more recent and sophisticated probabilistic methods (Weakness 1)
>
> We have compared with refined standalone TS diffusion models in $\underline{\text{Table 3 of the main text}}$. As suggested, we now extend the comparison to a broader set of probabilistic models:
>
> 1. TempFlow [1]: a step-wise autoregressive model based on normalizing flow;
> 2. D3U [2]: a diffusion model that decouples deterministic and uncertain components, with a point-forecasting model predicting the deterministic term and a diffusion model predicting the uncertain.
> 3. Chronos[3]: a foundation time-series model using quantized series and advanced LLM architectures for zero-shot forecasting.
> 4. Moirai [4]: a patch-based foundation model employing mixture distributions for complex outputs, enabling zero-shot forecasting.
>
> Results in the following Table show that the decoder-only Transformer trained with our MMPD loss consistently outperforms these models. Among them, TempFlow uses an outdated channel-mixing embedding and step-wise normalizing flow optimization, leading to weaker performance. D3U is a carefully designed diffusion model. Compared with its two-stage pipeline (point prediction + residual diffusion), our end-to-end approach is more straightforward. It should be mentioned that Chronos and Moirai are zero-shot foundation models. It is expected that supervised models outperform them when training data is available. The comparison is not to "defeat" these zero-shot models, but rather to better position MMPD within the broader probabilistic forecasting landscape.
>
> *Evaluation of MMPD loss against refined probabilistic forecasting models on ETTh1, $T=96, \tau=24$*.
> ||Top-3 MSE|Top-3 MAE|MSE|CRPS|
> |-|-|-|-|-|
> |TempFlow|0.794|0.621|1.016|0.558|
> |D3U|0.244|0.344|0.338|0.285|
> |Chronos (zero-shot)|0.321|0.331|0.413|0.296|
> |Moirai (zero-shot)|0.365|0.370|0.395|0.282|
> |Decoder+MMPD (ours)|**0.186**|**0.280**|**0.298**|**0.254**|
>
> *[1] KRasul, K., Sheikh, A.-S., Schuster, I., Bergmann, U. M., & Vollgraf, R. (2021). Multivariate probabilistic time series forecasting via conditioned normalizing flows. In ICLR.*
>
> *[2] Li, Q., Zhang, Z., Yao, L., Li, Z., Zhong, T., & Zhang, Y. (2025). Diffusion-based decoupled deterministic and uncertain framework for probabilistic multivariate time series forecasting. In ICLR.*
>
> *[3] Ansari, A. F., Stella, L., Turkmen, C., Zhang, X., Mercado, P., Shen, H., ... & Wang, Y. (2024). Chronos: Learning the language of time series. TMLR.*
>
> *[4] Woo, G., Liu, C., Kumar, A., Xiong, C., Savarese, S., & Sahoo, D. (2024). Unified Training of Universal Time Series Forecasting Transformers. In ICML.*
>
> # 2. Complexity analysis of MMPD loss (Weakness 2)
>
> **We have provided detailed theoretical and experimental complexity analysis of MMPD compared with MSE in $\underline{\text{Efficiency Analysis at Page 9 and Appendix G}}$. It appears that this part of the content might not have been sufficiently highlighted due to the formatting issue. We will adjust it in the final version.**
>
> Below is a summary of the analysis:
> 1. In implementation, the processing of adjacent patches is achieved via 1D convolution and then shifting, which is highly efficient and does not require explicit storage.
> 2. For training, the cost of MMPD is slightly higher than MSE because: 1)MMPD requires two MLP passes while MSE only requires one; 2)the patch consistent MLP is more complex than the conventional one. **However, since the backbone dominates training computation, the training cost remains nearly identical to MSE.**
> 3. For inference:
>
>    3.1. **If only deterministic prediction is required, the cost is very close to MSE** as only one patch consistent MLP pass is needed;
>
>    3..2. **To make probabilistic prediction that MSE cannot offer, MMPD’s overhead is much lower than that of standalone TS Diffusion models**, since: 1)Only one backbone pass is required to get the conditional tokens; 2)All denoising iterations are performed by a lightweight MLP. In contrast, standalone Diffusion models use heavy networks to perform denosing iterations.
>
>    3.3. **The cost of GMM is negligible compared with neural network computation** as it only involves vector operations. In short, one GMM iteration requires $O(\tau N M)$ FLOPs. while one patch consistent MLP pass requires $O(\frac{\tau}{P} N d^2)$. ($\tau$: prediction length, $N$: number of samples to generate, $M$: maximum number of modes in GMM, $d$: hidden dimension of MLP, $P$: patch size)
>
> In summary, the training and deterministic inference costs of MMPD are very close to MSE. For probabilistic/multi-mode prediction—which MSE cannot provide—MMPD incurs far less overhead than standalone diffusion models.

---

> > ### Author Response · Authors · 2025-11-19
> > **Response to Reviewer XDPi (2/2)**
> >
> > # 3. Rationality of GMM prior (Weakness 3 & Question 1)
> >
> > We do not directly assume the samples follow a GMM distribution. **The GMM prior is theoretically induced from the multi-mode assumption and forward diffusion process.** The derivation is summarized below:
> >
> > 1. Observing the multi-mode pattern in denosing samples, we assume that the true distribution follows the discrete multi-mode form in Equation (9):
> >
> > $$q(y^0|x)=\sum_{m=1}^Mw_m\delta(y^0-y^*_m)$$
> >
> > which states that there exist at most $M$ possible future outcomes, each with probability $w_m$.
> >
> > 2. A well-known property of the diffusion process is that its forward distribution $q(y^k|y^0)$ has a closed form expression (see Equation (4) of DDPM[5] for details):
> >
> > $$q(y^k|y^0)=\mathcal{N}(y^k;\sqrt{\bar{\alpha}_k}y^0,(1-\bar{\alpha}_k)I)$$
> >
> > *[5] Ho, J., Jain, A., & Abbeel, P. (2020). Denoising diffusion probabilistic models. In NeurIPS.*
> >
> > 3. Marginalizing the joint distribution $q(y^k,y^0|x)$ over $y^0$, we can get the GMM prior in Equation (10):
> >
> > $$q(y^k|x)=\int q(y^k,y^0|x) dy^0=\int q(y^0|x)q(y^k|y^0) dy^0$$
> >
> > $$=\sum_{m=1}^Mw_m\int\delta(y^0-y^*_m)\mathcal{N}(y^k;\sqrt{\bar{\alpha}_k}y^0,(1-\bar{\alpha}_k)I)dy^0$$
> >
> > $$=\sum_{m=1}^Mw_m\mathcal{N}(y^k;\sqrt{\bar{\alpha}_k}y^*_m,(1-\bar{\alpha}_k)I)$$
> >
> > Moreover, with this GMM prior distribution, our inference algorithm is entirely derived in closed form. The detailed derivation is in $\underline{\text{Appendix C}}$.
> >
> > We acknowledge that the true future multi-mode distribution may be more complex than the simplified discrete assumption above, making $q(y^k|x)$ deviate from an exact GMM. To address this potential mismatch, **we do not directly output the estimated parameters of GMM as predictions. Instead, we use GMM clustering as a mode-separation tool** and compute statistics of each cluster to provide the final predictions ($\underline{\text{Last two lines of Algorithm 1}}$). This yields a flexible distribution for each mode other than Gaussian with spherical covariance. An example is shown in $\underline{\text{Figure 1 of main text}}$, where different modes exhibit varying and asymmetric confidence intervals.
> >
> > # 4. Concerns regarding the variance schedule (Question 2)
> >
> > **We adopt the standard linear variance schedule commonly used in image diffusion models, without any domain-specific modification.** Additionally, we also evaluate some advanced schedules in $\underline{\text{Table 10 of Appendix F}}$, where results show further improvements. This suggests that designing time-series specific schedules is a promising direction for future research.
> >
> > Thank you again for your thoughtful comments. We hope our responses have addressed your concerns, and we would be happy to clarify anything further if needed.

---

### Official Review · Reviewer_7ixF · 2025-11-01

**Soundness:** 3
**Presentation:** 3
**Contribution:** 3
**Rating:** 6
**Confidence:** 3

**Summary:**

This paper addresses the challenge of diverse forecasting in time series analysis, where multiple plausible future outcomes exist rather than a single deterministic forecast. It introduces MMPD loss, a diffusion model-based probabilistic loss function that can be integrated with various patch-based forecasting backbones. The experiments on eight benchmark datasets and comparisons with several baselines demonstrate MMPD’s good performance in diverse, deterministic, and probabilistic forecasting.

**Strengths:**

1. Introduces a diffusion-based loss capturing multi-modal future distributions, addressing fundamental limitations of traditional single-mode losses.
2. Can be applied to any patch-based forecasting backbone, making it flexible for different architectures.
3. Effectively generates multiple sharp predictions with corresponding probabilities, enabling risk-aware decision-making.
4. The proposed AdaLN-MLP accounts for adjacent patches, reducing inconsistency and unrealistic discontinuities in forecasts.

**Weaknesses:**

1. Multi-mode inference requires multiple samples and iterative GMM fitting, which may introduce latency for real-time applications.
2. The diffusion-based loss and multi-mode inference algorithm add complexity to the training and prediction pipeline.
3. How robust is MMPD to different patch sizes and ranges of adjacent patch conditions in the Patch Consistent MLP?
4. Can MMPD be extended to explicitly handle multivariate forecasting jointly rather than per-channel averaging?
5. How does MMPD perform under non-stationary or concept drift conditions in time series data?

**Questions:**

See above

---

> ### Author Response · Authors · 2025-11-19
> **Response to Reviewer 7ixF (1/2)**
>
> Thank you for your thorough review and thoughtful comments. Below, we address the concerns regarding complexity, robustness to patch settings, multivariate dependency modeling, and performance under non-stationarity.
>
> # 1. Additional complexity of MMPD Loss compared with MSE (Weakness 1&2)
>
> To model complex distributions beyond simple Gaussian assumptions, MMPD indeed introduces more computation than MSE. However, we have carefully controlled this overhead so that the method remains practical in real applications. We provide detailed theoretical and experimental complexity analysis in $\underline{\text{Efficiency Analysis at Page 9 and Appendix G}}$. Below is a brief summary:
>
> 1. For training, the cost of MMPD is slightly higher than MSE because: 1)MMPD requires two MLP passes while MSE only requires one; 2)the patch consistent MLP is more complex than the conventional one. **However, since the backbone dominates training computation, the training cost remains nearly identical to MSE.**
> 2. For inference:
>
>     2.1. **If only deterministic prediction is required, the cost is very close to MSE** as only one patch consistent MLP pass is needed;
>
>      2.2. **To make probabilistic predictions that MSE cannot offer, MMPD’s overhead is much lower than that of standalone TS Diffusion models**, since: 1)Only one backbone pass is required to get the conditional tokens; 2)All denoising iterations are performed by a lightweight MLP. In contrast, standalone Diffusion models use heavy networks to perform denoising iterations.
>
>     2.3. **The cost of GMM is negligible compared with neural network computations** as it only involves vector operations. In short, one GMM iteration requires $O(\tau N M)$ FLOPs. while one patch consistent MLP pass requires $O(\frac{\tau}{P} N d^2)$. ($\tau$: prediction length, $N$: number of samples to generate, $M$: maximum number of modes in GMM, $d$: hidden dimension of MLP, $P$: patch size)
>
> In summary, the training and deterministic inference costs of MMPD are very close to MSE. For probabilistic/multi-mode prediction—which MSE cannot provide—MMPD incurs far less overhead than standalone diffusion models.
>
> # 2. Robustness to different patch sizes and ranges of adjacent patches (Weakness 3)
>
> **Patch size $P$**. We adjust the patch size and evaluate its effect in the following table. Results show that MMPD is robust to patch size within a large region. It should be noticed that, as an important hyperparameter for patch-based models, patch size not only affects MMPD but also affects the upstream backbone.
>
> *Top-3 MSE comparison of MMPD and MSE for varying patch size on Weather (prediction horizon $\tau=192$). Other metrics act similarly to Top-3 MSE and are omitted. The default value used in our main experiments is marked with $\checkmark$.*
> |Patch Size|6|8|12$\checkmark$|24|48|
> |-|-|-|-|-|-|
> |MSE|0.190|0.184|0.186|0.186|0.186|
> |MMPD|0.170|0.164|0.161|0.157|0.161|
>
> **Range of adjacent patches $r$**. We evaluated the effect of $r$ on dataset Dynamic in $\underline{\text{Figure 4(a) of main text}}$. Here we evaluate it on more datasets, and the results are shown in the table below. Across all datasets, **The key transition occurs from $r=0$ to $r=1$**, enabling MMPD to maintain consistency among patches, leading to a sharp performance improvement. Further increasing $r$ leads to performance improvements, but the gains gradually diminish.
>
> *Top-3 MSE for varying adjacent range $r$ in Patch Consistent MLP on different datasets (prediction horizon $\tau=192$ for ETTh1, ETTm1, Weather and $\tau=180$ for Dynamic). Top-3 MAE exhibits similar trends to Top-3 MSE. MSE and CRPS remain unchanged w.r.t $r$.The default value is marked with $\checkmark$.*
> |$r$|0|1|2|3$\checkmark$|5|10|
> |-|-|-|-|-|-|-|
> |ETTh1|0.404|0.387|0.383|0.377|0.372|0.365|
> |ETTm1|0.283|0.244|0.239|0.239|0.241|0.237|
> |Weather|0.192|0.168|0.165|0.158|0.159|0.157|
> |Dynamic|0.402|0.353|0.342|0.338|0.326|0.332|

---

> > ### Author Response · Authors · 2025-11-19
> > **Response to Reviewer 7ixF (2/2)**
> >
> > # 3. Extension of MMPD to explicitly model multivariate dependency (Weakness 4)
> >
> > The standard approach for multivariate loss is to compute losses per channel and then average across channels. Our original solution also follows this protocol.
> >
> > For multivariate extension, we further clarify that modeling multivariate dependency can be decomposed into two aspects:
> >
> > 1. **Extracting cross-channel dependency into latent tokens.** This can be handled by cross-channel backbones and our MMPD integrates well with these backbones. Actually, the Crossformer we used in our experiments is a typical backbone of this category.
> > 2. **Modeling the joint distribution of all future patches across steps and channels**. This requires extending the Patch-Consistent MLP so that each channel’s prediction incorporates information from other channels. We achieve this by adding a channel-aware conditioning term into the current MLP. Specifically, for an $S$-dimensional dataset, at denosing step $k$, we use $p^k_{j, s}$ to denote the $j$-th patch in channel $s$. The modified condition is:
> >
> > $$c_{j,s}^k= token_{j,s}+step^k+prev_{j,s}^k+next_{j,s}^k+channel_{j,s}^k$$
> >
> > $$channel_{j,s}^k=\sum_{v \ne s} W_vp^k_{j, v}$$
> >
> > Here, the first four terms remain identical to the univariate case. The new term $channel_{j,s}^k$ aggregates patches from all other channels at step $j$. $W_v \in \mathbb{R}^{d_{model}\times P}$ is the projection matrix for channel $v$. As a result, the extended MMPD jointly conditions on: 1) adjacent patches within the same channel; 2) patches from other channels at the same step.
> >
> > We implemented the above multivariate Patch Consistent MLP and compared it against the original univariate version. Results are shown in the table below.
> >
> > *Comparison of multivariate v.s. univariate Patch Consistent MLP on Weather ($\tau=192$). $\checkmark$ denotes the original version used in the paper.*
> > |MLP|Top-3 MSE|Top-3 MAE|MSE|CRPS|
> > |-|-|-|-|-|
> > |Univariate$\checkmark$|0.161|0.200|0.195|0.185|
> > |Multivariate|0.160|0.198|0.195|0.182|
> >
> > Although the multivariate version incorporates patches from other channels, the performance improvement remains marginal. The possible reason is that unlike temporal adjacency, it is hard to define "adjacent" channels, so the denosing network must simultaneously consider all other channels. Our current linear combination may not be expressive enough to capture such complex inter-channel interactions. Additionally, the receptive field should not be limited to step $j$ and incorporating temporally adjacent steps from other channels may also be necessary.
> >
> > Therefore, a more powerful and explicitly multivariate denoising network would be needed to fully exploit cross-channel dependencies, which would conflict with our design goal of keeping MMPD lightweight and plug-and-play. We consider this an interesting direction for future work.
> >
> > # 4. Performance under non-stationary or concept drift conditions (Weakness 5)
> >
> > To evaluate performance under non-stationarity, we generate synthetic datasets with explicit drift. The generation process includes: 1) generate a base stationary time series; 2) add a drift component.
> >
> > We consider the following two drifts:
> >
> > - Mean-Drift: mean gradually increases over time;
> > - Variance-Drift: variance gradually increases over time.
> >
> > These synthetic datasets explicitly contain non-stationary patterns that cause out-of-distribution issues in train/valid/test splits. We compare MMPD with MSE loss on these datasets. Results in the following table show that MMPD maintains competitive deterministic accuracy while enabling richer distribution modeling, supporting multi-mode and probabilistic forecasting.
> >
> > *Comparison of MMPD against MSE loss on different synthetic non-stationary datasets.*
> > |Drift|Loss|Top-3 MSE|Top-3 MAE|MSE|CRPS|
> > |-|-|-|-|-|-|
> > |Mean-Drift|MSE|0.0158|0.1003|**0.0072**|0.1539|
> > ||MMPD|**0.0068**|**0.0657**|0.0075|**0.0473**|
> > |Variance-Drift|MSE|0.0918|0.2418|**0.0690**|0.2703|
> > ||MMPD|**0.0671**|**0.2071**|0.0717|**0.1590**|
> >
> > We hope our responses have fully addressed your concerns. If there are any additional questions or clarifications needed, we would be happy to provide them.

---

### Author Response · Authors · 2025-11-30
**Summary of discussion during the rebuttal period**

Traditional MSE-centric training implicitly assumes a unimodal Gaussian distribution with fixed variance. We propose the MMPD loss for patch-based backbones, enabling the modeling of complex future distributions and the generation of multiple diverse predictions with corresponding probabilities.

**Before the discussion stage, our work received positive evaluations from all four reviewers (scores: 8, 6, 6, 6).** Reviewers agreed that the work is clearly motivated (XDPi, kPh3), addresses a fundamental limitation of MSE-centric training (7iXF, XDPi, hVy7), applies to any patch-based backbone (7iXF, XDPi, hVy7), and includes comprehensive experiments (hVy7, kPh3).

Reviewers raised several technical concerns and suggestions, which we addressed thoroughly. The common concerns are:

- **7ixF and XDPi questioned about MMPD's efficiency vs. MSE.** We referred them to detailed analysis on Page 9 and Appendix G, and clarified that MMPD’s training and deterministic inference costs are close to MSE. The probabilistic and multi-mode inference--beyond MSE’s capability--is far more efficient than standalone diffusion models.

- **7ixF and hVy7 suggested extending MMPD to explicitly capture cross-channel dependency.** We experimented with this extension and found that the performance gain is marginal despite having higher complexity.

- **XDPi, hVy7, and kPh3 requested comparisons with broader probabilistic models.** We compare with advanced normalizing flows, diffusion models, and zero-shot foundation models, demonstrating that MMPD consistently outperforms them.

Other specific concerns are about hyper-parameter effects (7ixF), performance on non-stationary data (7ixF), GMM prior justification (XDPi), and padding for non-divisible lengths (kPh3). We resolved all of them with additional experiments or theoretical explanations.

After the first-round discussion by **Nov 26 EST (before the reported leakage on Nov 27)**, Reviewer hVy7 and kPh3 confirmed that major concerns were addressed. Reviewer hVy7 increased the confidence score from 2 to 4; Reviewer kPh3 increased the overall rating from 6 to 8, as evidenced by their comments. **Thus, the post-discussion scores were 8, 8, 6, 6, with confidences 4, 4, 3, 3 before reverting.**

Reviewer hVy7 and kPh3 also gave forward-looking suggestions. Although no further feedback could be obtained, we still addressed them as follows:

- **hVy7 suggested that MMPD can serve as a future paradigm for integrating temporal tokens with text tokens.** We proposed an approach to adapt MMPD for multi-mode next patch prediction, enabling unified modeling of text and time-series modalities within a shared autoregressive paradigm.
- **kPh3 request evaluations on long input/output settings.** We evaluated MMPD on varying input/output lengths, showing that MMPD benefits more from extended lookback windows and achieves better performance than the baseline.

---

### Public Comment · ~Saleh_GHOLAM_ZADEH1 · 2026-03-24
**There are some typos in the appendix**

Congratulations! Very interesting and nice work. I enjoyed reading this paper a lot.

I found these minor typos in the appendix. \
In page 16, M-step, "be updated" should be replaced with "been updated".
In page 17, below equation 23 and another one below equation 26, "be updated" should be replaced with "been updated".

Additionally inside Algorithm 1, you may want to mention that, E-step and M-step  are done until convergence (at every fixed k). In the current version it sounds like you do only one E-step and one M step, although appendix C and the released code say otherwise.

---

> ### Public Comment · ~Yunhao_Zhang1 · 2026-04-03
>
> Thanks for your attention!
>
> According to the EM-steps, we found that iterating until convergence at each fixed k is not computationally efficient; most of the time, a single iteration is enough. Therefore, we use only one iteration in Algorithm 1. For implementation, we make it a hyperparameter and set the default value to 10, ensuring a sufficient margin for usage in unforeseen scenarios.

---

### Meta-Review · Area_Chair_nzBF · 2026-01-05

**Summary:**

The paper addresses a fundamental limitation of MSE-based training by modeling multi-modal future distributions through a diffusion-based loss, which is well motivated from both data and application perspectives. The proposed approach is flexible and architecture-agnostic, as it can be integrated with various forecasting backbones without modification. Extensive experiments across multiple datasets and models demonstrate strong performance, producing sharp, diverse predictions with calibrated probabilities for risk-aware decision making. All reviewers gave highly positive and consistent evaluations of this work, with one reviewer even considering increasing the score to 10. In my view, there is little hesitation in recommending acceptance of this paper.

**Reviewer Concerns:**

All concerns are addressed.

**Reviewer Scores:**

I think Reviewer kPh3 would increase the score if participate fully in discussion.

---

### Decision · Program_Chairs · 2026-01-26

Accept (Poster)